# On the Role of Initialization on the Implicit Bias in Deep Linear Networks

## Abstract

Despite Deep Learning's (DL) empirical success, our theoretical understanding of its efficacy remains limited. One notable paradox is that while conventional wisdom discourages perfect data fitting, deep neural networks are designed to do just that, yet they generalize effectively. This study focuses on exploring this phenomenon attributed to the implicit bias at play. Various sources of implicit bias have been identified, such as step size, weight initialization, optimization algorithm, and number of parameters. In this work, we focus on investigating the implicit bias originating from weight initialization. To this end, we examine the problem of solving underdetermined linear systems in various contexts, scrutinizing the impact of initialization on the implicit regularization when using deep networks to solve such systems. Our findings elucidate the role of initialization in the optimization and generalization paradoxes, contributing to a more comprehensive understanding of DL's performance characteristics.

## 1 Introduction

Deep Learning (DL) has revolutionized many fields and is poised to radically transform the modern world. DL is quickly becoming the best practice for many computer vision problems in commerce, finance, medicine, entertainment, and many more fields that shape our daily lives. This explosion in popularity in recent years, both in academia and in industry, is due to its practical success. Unfortunately, our understanding of DL and why it is so successful is lagging far behind. Simply put, we do not have satisfactory explanations for why it performs so well, or why it is even possible to optimize such non-convex models. These are two key examples of unanswered questions on this subject matter, among many others.

When studying neural networks, it is tempting to consider underdetermined linear systems as an exploratory model, as it retains many key characteristics that make Deep Neural Networks difficult to analyze (non-convexity, overparameterization) while also having the advantages of being a simple linear model (and thus theorems from linear algebra easily apply). Furthermore, there is a growing consensus that wide neural networks are approximately linear and operate in the so-called lazy regime (Liu et al., 2022). Highly overparameterized models that operate in the lazy regime are approximately Gaussian Processes, and thus can be linked to kernel methods, which are linear in the weights (but not the input). Lee et al. (2018) have established the link between wide networks and kernel machines. Thus, one can expect observations on linear models to apply, at least approximately, on non-linear deep networks, which further motivates us to explore overparametrized linear models and kernel methods.

A linear model attempts to find a weight vector $\mathbf{y}$ such that given an example-by-feature matrix $\mathbf{A}$ (or some non-linear transformation of an original example-by-feature matrix, in the case of kernel machines) and a target vector $\mathbf{b}$, we have $\mathbf{A}\mathbf{y} = \mathbf{b}$. In other words, a system of linear equations. In this work, we consider the problem of solving underdetermined systems of linear equations through the lens of DL, and specifically through fully connected linear neural networks.

A fully connected neural network is a model where we are given some example-by-feature matrix $\mathbf{A}$ and ground truth vector $\mathbf{b}$, and our goal is to find weights $\mathbf{W}_1, \mathbf{W}_2, \ldots, \mathbf{W}_h, \mathbf{x}$ such that

$$L_{\mathbf{A},\mathbf{b}}(\mathbf{W}_1, \mathbf{W}_2, \ldots, \mathbf{W}_h, \mathbf{x}) = \frac{1}{2}\|\sigma(\sigma(\ldots(\sigma(\mathbf{A}\mathbf{W}_1)\mathbf{W}_2)\ldots\mathbf{W}_h)\mathbf{x}) - \mathbf{b}\|_2^2$$

is minimized, where $\sigma$ is some activation function. Common choices are $\text{ReLU}(x) = \max(0, x)$ or $\text{Sigmoid}(x) = \frac{1}{1+e^{-x}}$. The success of these models in real life scenarios can not be overstated.

However, as previously stated, there are several open questions about this framework that require answers. Before we mention the most interesting questions and the connection to this work, a few key insights are given:

1. Deep networks are highly over-parametrized. Many possible minimizers, some better, some worse.

2. If $h > 1$ then $L_{\mathbf{A},\mathbf{b}}$ is non-convex, even if $\sigma(z) = z$

These properties alongside the empirical success of of deep networks go against prevailing common wisdoms in machine learning and statistical inference, that over-parametrized models tend to overfit, and that minimizing a non convex objective is difficult. However, when we consider the success of DL, this intuition seems incorrect. Deep networks are highly overparameterized, often having tens of billions of parameters, but, surprisingly, they often predict well (the *generalization paradox*). Deep networks are optimized by minimizing a non-convex function, yet, often a minimizer is found quickly (the *optimization paradox*). These two paradoxes have suscited great interest and a vast literature that explores them.

Much like this work, contemporary research frequently focuses explicitly on linear networks since it is an excellent model problem to comprehend for the reasons described above. In a linear model, the activation function $\sigma$ is simply $\sigma(x) = x$, and so

$$L_{\mathbf{A},\mathbf{b}}(\mathbf{W}_1, \mathbf{W}_2, \ldots, \mathbf{W}_h, \mathbf{x}) = \frac{1}{2}\|\mathbf{A}\mathbf{W}_1\mathbf{W}_2\ldots\mathbf{W}_h\mathbf{x} - \mathbf{b}\|_2^2$$

These linear models seem useless at first glance, as composition of linear functions is still linear. However, from an optimization perspective, they can behave quite differently. Indeed, the objective function is non-convex if $h > 1$ with many possible saddle points, including a trivial one ($\forall i : \mathbf{W}_i = 0$). The trivial saddle point ($\forall i : \mathbf{W}_i = 0$) proves non-convexity for $h > 1$ and highlights a key difference between the naive linear model and a deep linear model. Interestingly, this model can have advantages over the shallow model in certain scenarios. For example, Bah et al. (2021) show that optimizing a deep linear network is equivalent to Riemannian gradient flow on a manifold of low-rank matrices, with a suitable Riemannian metric. They show that this Riemannian optimization converges to a global optimum of $L^1$ loss with very high probability (given the rank constraint), and when the depth of the linear network is two, then with very high probability it minimizes $L^2$ loss.

In this work, we attempt to shed light on interesting properties that arise from initialization in several different linear network scenarios. To this end, we study ordinary linear regression in an overparameterized setting. We prove a condition for convergence to optimal solution with respect to the Euclidean norm (which is in line with the study in Bartlett et al. (2020)), provide an expression for the converged solution as a function of the initial gradient descent guess, and outline an algorithm that is capable of controlling to which solution gradient descent will converge (see Section 2.3). The reason we focus specifically on initialization is due to both industry experience that this seemingly innocent choice can have a drastic effect on generalization and theoretical results that used specific initialization schemes (Hu et al., 2020; Belkin et al., 2019; Bartlett et al., 2020).

Next, we prove similar results for an overparameterized linear model that has a single hidden layer. We show that it is possible to find a point where the solution gradient descent converges to is optimal, as well as having every weight be optimal with respect to the other weights. We provide algorithms that take advantage of this optimality to reduce the dimensionality of the problem (see Section 3). We then proceed to studying deep linear networks, proving a condition for when gradient descent converges to optimum, providing an argument

for why a balanced optimum point is unlikely to be found using our method, with more than two hidden layers. We then study the stability of deep linear networks and prove properties regarding weight norms (Section 4). Finally, we provide motivation and explanation for linear Riemannian models, study properties of such models, and perform experiments that emphasize the interesting traits of such models (these results are reported in Section 5).

Our results hint at the importance of initialization when designing deep learning solutions. Proper initialization of learning can be treated as another component controlled by a practitioner. By carefully selecting initialization, the practitioner can bias the result towards desired outcomes (e.g. perhaps using data specific initialization), reduce number of parameters, and accelerate convergence. However, additional research is required to determine how the aforementioned ideas can be translated into practice.

### 1.1 Related Work

Our work is closely related to, and inspired by, Bartlett et al.'s work on benign overfitting in a shallow, ordinary linear regression setting (Bartlett et al., 2020). It is shown there that in some cases, depending on the dimensionality of the problem and the spectrum of the noise covariance, the minimum norm interpolating solution generalizes well, in stark contradiction to common wisdom that says we should never interpolate, certainly in noisy settings. In this work, we focus on investigating the implicit bias that arises from initialization in deep linear networks, and it turns out that it is possible to easily bias the solution towards the minimum norm interpolant.

Our work is also related to the work of Belkin et al. (2019) which attempts to explain the disconnect between classical theory and the success of interpolating overparameterized solutions in practice, by suggesting a single unified "double descent" performance curve. We also investigate performance curves of interpolating overparametricized solutions but in a strictly linear setting. This is in contrast to Belkin et al. (2019) which considered random Fourier features, which are non-linear in the input.

Related is also a series of papers by Arora et al. (2018; 2019) on the effects of depth on generalization, optimization, and specifically on weight norms. They suggest that contrary to common wisdom, overparameterization accelerates the convergence of optimization, which is something we also explore, even managing to collapse a deep model into a shallow model, similarly to Ablin (2020) which shows that deep orthogonal linear networks are shallow. Furthermore, Arora et al. (2019) explores the effects of overparameterization via depth on which type of solution we converge to in matrix factorization, which is a continuation of the foundation that Gunasekar et al. (2017) laid out, and whether we are implicitly biased to minimize norm (and which type of norm), or rank.

Our work goes somewhat against Razin & Cohen (2020) and focuses exclusively on norms. That work has shown that in a matrix completion setting (which is different from our setting), there are natural problems where we can implicitly bias towards a solution that generalizes well, but that bias is not towards minimum norm, but rather it minimizes rank, even at the cost of pushing the nuclear norm towards infinity. This is another notion of implicit regularization, which does not apply in our setting (since we are dealing with linear regression where the solution is a one-dimensional column vector), and we do not focus on it at all.

## 2 Preliminaries

### 2.1 Notation

We denote scalars using Greek letters or $x, y, \ldots$. Vectors are denoted by $\mathbf{x}, \mathbf{y}, \ldots$ and matrices by $\mathbf{A}, \mathbf{B}, \ldots$. The $s \times s$ identity matrix is denoted by $\mathbf{I}_s$. If $\mathbf{A}$ is a $n \times d$ matrix then it has a singular value decomposition $\mathbf{A} = \mathbf{U}\mathbf{\Sigma}\mathbf{V}^{\mathrm{T}}$ where $\mathbf{U} \in \mathbb{R}^{n \times n}$ is orthogonal, $\mathbf{\Sigma} \in \mathbb{R}^{n \times d}$ is rectangular diagonal and $\mathbf{V} \in \mathbb{R}^{d \times d}$ orthogonal. For simplicity, we differentiate between column vectors and row vectors explicitly. A $s$ dimensional row vector is denoted as being in $\mathbb{R}^{1 \times s}$, and a $s$ dimensional column vector is denoted as being in $\mathbb{R}^{s \times 1}$. If $\mathbf{x}$ is a vector of any shape or dimension, we use $\|\mathbf{x}\|_2$ for the Euclidean norm. If $\mathbf{X}$ is a matrix of any shape or dimension, we use $\|\mathbf{X}\|$ for the operator (spectral) norm and $\|\mathbf{X}\|_F$ to mean the Frobenius norm

$(\|\mathbf{X}\|_F = \sqrt{\operatorname{trace}(\mathbf{X}\mathbf{X}^{\mathrm{T}})})$. The Moore-Penrose pseudoinverse of a matrix $\mathbf{M}$ with is denoted by $\mathbf{M}^+$. If the columns of $\mathbf{M}$ are independent, it is equal to $\mathbf{M}^+ = (\mathbf{M}^{\mathrm{T}}\mathbf{M})^{-1}\mathbf{M}^{\mathrm{T}}$.

When solving a linear system of equations, $\mathbf{A} \in \mathbb{R}^{n \times d}$ is our coefficient matrix where we assume $d > n$, and $\operatorname{rank}(\mathbf{A}) = n$ unless otherwise stated. We denote by $\mathbf{b} \neq 0 \in \mathbb{R}^{n \times 1}$ the target vector. We denote by $\alpha > 0$ the step size or the learning rate in the gradient descent iteration. We define $\boldsymbol{\theta}^\star$ to be the minimum norm solution

$$\boldsymbol{\theta}^\star := \underset{\mathbf{A}\mathbf{x}=\mathbf{b}}{\arg\min} \|\mathbf{x}\|_2 = \mathbf{A}^{\mathrm{T}}(\mathbf{A}\mathbf{A}^{\mathrm{T}})^{-1}\mathbf{b}$$

If $f$ is a function of two or more variables, we will explicitly write $\nabla_{\mathbf{x}} f$ to refer to the gradient with respect to the variable $\mathbf{x}$, and so on. In deep models with hidden weights, all hidden layer weights are assumed to be $d \times d$ unless otherwise stated. The subscripts will be used to denote gradient descent iterations, so $\mathbf{W}_k$ is the weight matrix $\mathbf{W}$ in iteration $k$, and we denote $\mathbf{W}_\infty := \lim_{k \to \infty} \mathbf{W}_k$ if such a limit exists. In Section 4 we make a slight change of notation where $\mathbf{W}_i^{(k)}$ stands for the value of the matrix $\mathbf{W}_i$ in iteration $k$, and $\mathbf{W}_i^{(\infty)} = \lim_{k \to \infty} \mathbf{W}_i^{(k)}$

## 2.2   Solving Underdetermined Least Squares using Gradient Descent

In this section we consider the classical task of finding a single vector $\mathbf{y} \in \mathbb{R}^{d \times 1}$ such that $\mathbf{A}\mathbf{y} = \mathbf{b}$, where we assume this is accomplished by defining a loss function

$$L_{\mathbf{A},\mathbf{b}}(\mathbf{y}) = \frac{1}{2}\|\mathbf{A}\mathbf{y} - \mathbf{b}\|_2^2$$

and applying gradient descent with fixed step size $\alpha > 0$. That is, an initial guess $\mathbf{y}_0$ is picked, and then in each iteration the algorithm moves in the direction directly opposite to the gradient

$$\nabla L_{\mathbf{A},\mathbf{b}}(\mathbf{y}) = \mathbf{A}^{\mathrm{T}}(\mathbf{A}\mathbf{y} - \mathbf{b})$$

with fixed step size $\alpha$. Thus, the iteration is

$$\mathbf{y}_{k+1} = \mathbf{y}_k - \alpha \mathbf{A}^{\mathrm{T}}(\mathbf{A}\mathbf{y}_k - \mathbf{b})$$

In many applications, it can occur that there is a single minimizer. However, since we are dealing with underdetermined systems $(d > n)$ and assume $\operatorname{rank}(\mathbf{A}) = n$, there is an infinite set of solutions: $\boldsymbol{\theta} = \boldsymbol{\theta}^\star + \mathbf{z}$ where $\mathbf{z}$ is any vector in $\ker(\mathbf{A})$.

Suppose $\mathbf{A}, \mathbf{b}$ were sampled from some population of features and targets for a problem for which we wish to build a predictive model. Common statistical wisdom is that complex prediction rules are inferior to simple ones. In this context, simplicity can refer to the solution vector's norm. Hence, not all solutions to $\mathbf{A}\mathbf{y} = \mathbf{b}$ will be as useful for predictive purposes. Our goal and objective in many scenarios is to find $\boldsymbol{\theta}^\star$.

If the iteration starts from an arbitrary $\mathbf{y}_0$, it will converge to an arbitrary solution, and we can expect poor generalization unless some explicit regularization is used. However, we now show that if we initialize smartly, we can ensure convergence to $\boldsymbol{\theta}^\star$, or any other predetermined solution.

The following lemma, which shows that the minimum norm solution $\boldsymbol{\theta}^\star$ is the *only* solution in row-space of $\mathbf{A}$, is a fundamental and classic result on underdetermined linear systems, which we make extensive use of throughout this work. We stress at the outset that this lemma is true regardless of the optimization algorithm used to reach a solution of the system.

**Lemma 1.** *If $\mathbf{A}\mathbf{y} = \mathbf{b}$ and $\mathbf{y} \in \mathbf{range}\left(\mathbf{A}^T\right)$ then $\mathbf{y} = \boldsymbol{\theta}^\star$.*

*Proof.* Since $\mathbf{y} \in \mathbf{range}\left(\mathbf{A}^T\right)$ there exists a $\mathbf{v} \in \mathbb{R}^{n \times 1}$ such that $\mathbf{y} = \mathbf{A}^{\mathrm{T}}\mathbf{v}$. So $\mathbf{A}\mathbf{A}^{\mathrm{T}}\mathbf{v} = \mathbf{b}$. Multiply the last equation on the left by $\mathbf{A}^{\mathrm{T}}(\mathbf{A}\mathbf{A}^{\mathrm{T}})^{-1}$ to get $\mathbf{y} = \boldsymbol{\theta}^\star$. $\qquad\square$

The next lemma is specific to gradient descent. It shows that being in the row-space of $\mathbf{A}$ is conserved throughout gradient descent iterations.

**Lemma 2.** *If* $\mathbf{y}_k \in \mathbf{range}\left(\mathbf{A}^T\right)$ *for some* $k$ *then* $\mathbf{y}_{k+1} \in \mathbf{range}\left(\mathbf{A}^T\right)$.

*Proof.* Since $\mathbf{y}_k \in \mathbf{range}\left(\mathbf{A}^{\mathrm{T}}\right)$ there exists a $\mathbf{v}_k \in \mathbb{R}^{n \times 1}$ such that $\mathbf{y}_k = \mathbf{A}^{\mathrm{T}}\mathbf{v}_k$. Then

$$\mathbf{y}_{k+1} = \mathbf{y}_k - \alpha\mathbf{A}^{\mathrm{T}}(\mathbf{A}\mathbf{y}_k - \mathbf{b}) = \mathbf{A}^{\mathrm{T}}(\mathbf{v}_k - \alpha(\mathbf{A}\mathbf{A}^{\mathrm{T}}\mathbf{v}_k - \mathbf{b})) \in \mathbf{range}\left(\mathbf{A}^{\mathrm{T}}\right).$$

$\square$

An important consequence of Lemmas 1 and 2, and the fact that $L_{\mathbf{A},\mathbf{b}}$ is convex, is the following corollary, which demonstrates a key aspect of our work: if we initialize in an intelligent way, we are guaranteed to converge to the optimal solution with respect to $l_2$ norm.

**Corollary 3.** *If* $\mathbf{y}_0 \in \mathbf{range}\left(\mathbf{A}^T\right)$ *and* $\alpha > 0$ *is such that the iteration converges to a stationary point, then* $\mathbf{y}_\infty = \boldsymbol{\theta}^\star$. *A trivial choice for* $\mathbf{y}_0$ *that ensures that* $\mathbf{y}_0 \in \mathbf{range}\left(\mathbf{A}^T\right)$ *is* $\mathbf{y}_0 = 0$.

*Proof.* Applying Lemma 2 in an inductive manner, we see that for all $k$ there exists a $\mathbf{v}_k$ such that $\mathbf{y}_k = \mathbf{A}^{\mathrm{T}}\mathbf{v}_k$. We also assumed convergence, so we know that $\mathbf{y}_\infty$ exists and is equal to some stationary point $\mathbf{y}$. Note that since $\mathbf{A}^{\mathrm{T}}$ has full rank, a stationary point must be a solution to $\mathbf{A}\mathbf{x} = \mathbf{b}$. Since $\mathbf{A}^{\mathrm{T}}$ has full column rank, we also have $\mathbf{v}_k = (\mathbf{A}^{\mathrm{T}})^+\mathbf{y}_k$, so $\mathbf{v}_\infty$ also exists. So,

$$\mathbf{y} = \mathbf{y}_\infty = \lim_{k \to \infty} \mathbf{A}^{\mathrm{T}}\mathbf{v}_k = \mathbf{A}^{\mathrm{T}}\mathbf{v}_\infty \in \mathbf{range}\left(\mathbf{A}^{\mathrm{T}}\right)$$

To sum up, $\mathbf{y}$ is a solution and $\mathbf{y} \in \mathbf{range}\left(\mathbf{A}^{\mathrm{T}}\right)$. Now apply Lemma 1 to get $\mathbf{y} = \boldsymbol{\theta}^\star$ $\square$

Though the last results are almost trivial, we mention them nonetheless as they lead to the central theme of this work: smart choices for initializations will bias us towards better solutions, and it is possible to determine properties of solutions we converge to simply by initializing in line with what we want to achieve. Notice that we have not added explicit regularization; this is not ridge regression. Using only a clever initialization, we have biased our solution to tend towards the minimal norm solution.

Furthermore, in this simple 0-depth case, it is possible to control exactly which solution the iteration will converge to. Theorem 4 proves this is possible and provides an algorithm to do so. This is another example of specific initializations yielding desirable solutions.

**Theorem 4.** *Suppose that* $\mathbf{A} \in \mathbb{R}^{n \times d}$ *is a matrix of any rank, where* $d \geq n$, *and let* $\mathbf{A} = \mathbf{U}\boldsymbol{\Sigma}\mathbf{V}^T$ *be a singular value decomposition of* $\mathbf{A}$. *Let* $\mathbf{V}_1$ *be the first* $n$ *columns of* $\mathbf{V}$, *and* $\mathbf{V}_2$ *the remaining columns. If* $\|\mathbf{A}\|^2 < \frac{2}{\alpha}$ *then for any given initial guess* $\mathbf{y}_0$ *we see that* $\mathbf{y}_\infty$ *exists and*

$$\mathbf{y}_\infty = \mathbf{V}_2\mathbf{V}_2^T\mathbf{y}_0 + \boldsymbol{\theta}^\star$$

*Proof.* Write

$$\boldsymbol{\Sigma} = \begin{bmatrix} \tilde{\boldsymbol{\Sigma}} & 0_{n \times (d-n)} \end{bmatrix}$$

where $\tilde{\boldsymbol{\Sigma}} \in \mathbb{R}^{n \times n}$ is diagonal. Notice that $\mathbf{A} = \mathbf{U}\tilde{\boldsymbol{\Sigma}}\mathbf{V}_1^{\mathrm{T}}$ and $\boldsymbol{\theta}^\star = \mathbf{V}_1\tilde{\boldsymbol{\Sigma}}^{-1}\mathbf{U}^{\mathrm{T}}\mathbf{b}$. At step $k$ of gradient descent we have

$$
\begin{aligned}
\mathbf{y}_k &= \mathbf{y}_{k-1} - \alpha\mathbf{A}^{\mathrm{T}}(\mathbf{A}\mathbf{y}_{k-1} - \mathbf{b}) \\
&= (\mathbf{I}_d - \alpha\mathbf{A}^{\mathrm{T}}\mathbf{A})\mathbf{y}_{k-1} + \alpha\mathbf{A}^{\mathrm{T}}\mathbf{b} \\
&= \ldots \\
&= (\mathbf{I}_d - \alpha\mathbf{A}^{\mathrm{T}}\mathbf{A})^k\mathbf{y}_0 + \alpha\sum_{j=0}^{k-1}(\mathbf{I}_d - \alpha\mathbf{A}^{\mathrm{T}}\mathbf{A})^j\mathbf{A}^{\mathrm{T}}\mathbf{b} \\
&= (\mathbf{V}(\mathbf{I}_d - \alpha\boldsymbol{\Sigma}^{\mathrm{T}}\boldsymbol{\Sigma})\mathbf{V}^{\mathrm{T}})^k\mathbf{y}_0 + \alpha\sum_{j=0}^{k-1}(\mathbf{V}(\mathbf{I}_d - \alpha\boldsymbol{\Sigma}^{\mathrm{T}}\boldsymbol{\Sigma})\mathbf{V}^{\mathrm{T}})^j\mathbf{V}\boldsymbol{\Sigma}^{\mathrm{T}}\mathbf{U}^{\mathrm{T}}\mathbf{b}
\end{aligned}
$$

Since $\mathbf{I}_d - \alpha\boldsymbol{\Sigma}^{\mathrm{T}}\boldsymbol{\Sigma}$ is diagonal and $\mathbf{V}$ is orthogonal, the last equation simplifies to

$$
\mathbf{y}_k = \mathbf{V}(\mathbf{I}_d - \alpha\boldsymbol{\Sigma}^{\mathrm{T}}\boldsymbol{\Sigma})^k\mathbf{V}^{\mathrm{T}}\mathbf{y}_0 + \alpha\sum_{j=0}^{k-1}\mathbf{V}(\mathbf{I}_d - \alpha\boldsymbol{\Sigma}^{\mathrm{T}}\boldsymbol{\Sigma})^j\boldsymbol{\Sigma}^{\mathrm{T}}\mathbf{U}^{\mathrm{T}}\mathbf{b}.
$$

Denote $\mathbf{z}_k = \mathbf{V}^{\mathrm{T}}\mathbf{y}_k$, and multiply the last equation by $\mathbf{V}^{\mathrm{T}}$ on the left to get

$$
\begin{aligned}
\mathbf{z}_k &= (\mathbf{I}_d - \alpha\boldsymbol{\Sigma}^{\mathrm{T}}\boldsymbol{\Sigma})^k\mathbf{z}_0 + \alpha\sum_{j=0}^{k-1}(\mathbf{I}_d - \alpha\boldsymbol{\Sigma}^{\mathrm{T}}\boldsymbol{\Sigma})^j\boldsymbol{\Sigma}^{\mathrm{T}}\mathbf{U}^{\mathrm{T}}\mathbf{b} \\
&= \begin{bmatrix} (\mathbf{I}_n - \alpha\tilde{\boldsymbol{\Sigma}}^2)^k & 0_{n \times (d-n)} \\ 0_{(d-n) \times n} & \mathbf{I}_{d-n} \end{bmatrix}\mathbf{z}_0 + \alpha\sum_{j=0}^{k-1}\begin{bmatrix} (\mathbf{I}_n - \alpha\tilde{\boldsymbol{\Sigma}}^2)^j\tilde{\boldsymbol{\Sigma}} \\ 0_{(d-n) \times n} \end{bmatrix}\mathbf{U}^{\mathrm{T}}\mathbf{b}
\end{aligned}
$$

The condition on $\alpha$ ensures that all eigenvalues of $\mathbf{I}_n - \alpha\tilde{\boldsymbol{\Sigma}}^2$ have absolute value strictly smaller than 1, which is a sufficient condition for $\lim_{k \to \infty}(\mathbf{I}_n - \alpha\tilde{\boldsymbol{\Sigma}}^2)^k = 0$, which is in turn equivalent to the convergence of the Neumann series $\sum_{j=0}^{\infty}(\mathbf{I}_n - \alpha\tilde{\boldsymbol{\Sigma}}^2)^j$ to $(\alpha\tilde{\boldsymbol{\Sigma}}^2)^{-1}$, so we have:

$$
\begin{aligned}
\lim_{k \to \infty}\mathbf{z}_k &= \begin{bmatrix} 0_n & 0_{n \times (d-n)} \\ 0_{(d-n) \times n} & \mathbf{I}_{d-n} \end{bmatrix}\mathbf{z}_0 + \alpha\sum_{j=0}^{\infty}\begin{bmatrix} (\mathbf{I}_n - \alpha\tilde{\boldsymbol{\Sigma}}^2)^j\tilde{\boldsymbol{\Sigma}} \\ 0_{(d-n) \times n} \end{bmatrix}\mathbf{U}^{\mathrm{T}}\mathbf{b} \\
&= \begin{bmatrix} 0_n & 0_{n \times (d-n)} \\ 0_{(d-n) \times n} & \mathbf{I}_{d-n} \end{bmatrix}\mathbf{z}_0 + \begin{bmatrix} \alpha(\alpha\tilde{\boldsymbol{\Sigma}}^2)^{-1}\tilde{\boldsymbol{\Sigma}} \\ 0_{(d-n) \times n} \end{bmatrix}\mathbf{U}^{\mathrm{T}}\mathbf{b} \\
&= \begin{bmatrix} 0_n & 0_{n \times (d-n)} \\ 0_{(d-n) \times n} & \mathbf{I}_{d-n} \end{bmatrix}\mathbf{z}_0 + \begin{bmatrix} \tilde{\boldsymbol{\Sigma}}^{-1} \\ 0_{(d-n) \times n} \end{bmatrix}\mathbf{U}^{\mathrm{T}}\mathbf{b}
\end{aligned}
$$

Since $\mathbf{y}_k = \mathbf{V}\mathbf{z}_k$ we have

$$
\begin{aligned}
\lim_{k \to \infty}\mathbf{y}_k &= \mathbf{V}(\lim_{k \to \infty}\mathbf{z}_k) \\
&= \begin{bmatrix} \mathbf{V}_1 & \mathbf{V}_2 \end{bmatrix}\begin{bmatrix} 0_n & 0_{n \times (d-n)} \\ 0_{(d-n) \times n} & \mathbf{I}_{d-n} \end{bmatrix}\begin{bmatrix} \mathbf{V}_1^{\mathrm{T}} \\ \mathbf{V}_2^{\mathrm{T}} \end{bmatrix}\mathbf{y}_0 + \begin{bmatrix} \mathbf{V}_1 & \mathbf{V}_2 \end{bmatrix}\begin{bmatrix} \tilde{\boldsymbol{\Sigma}}^{-1} \\ 0_{(d-n) \times n} \end{bmatrix}\mathbf{U}^{\mathrm{T}}\mathbf{b} \\
&= \mathbf{V}_2\mathbf{V}_2^{\mathrm{T}}\mathbf{y}_0 + \boldsymbol{\theta}^\star
\end{aligned}
$$

$\square$

As a somewhat esoteric use of the last theorem, we can not only tell in advance to which solution the iteration will converge to, we can also control to which solution. Theorem 4 tells us that if we want to reach a solution $\boldsymbol{\theta}$, to get a valid initial guess $\mathbf{y}_0$ that will lead to convergence to $\boldsymbol{\theta}$, we need to solve the system $\mathbf{V}_2\mathbf{V}_2^{\mathrm{T}}\mathbf{y}_0 = \boldsymbol{\theta} - \boldsymbol{\theta}^\star$. This is a $d \times d$ system of rank $d - n$. We claim that this system has infinitely many

solutions, because both $\boldsymbol{\theta}$ and $\boldsymbol{\theta}^\star$ are solutions to $\mathbf{A}\mathbf{y} = \mathbf{b}$, so $\boldsymbol{\theta} - \boldsymbol{\theta}^\star$ is a solution to $\mathbf{A}\mathbf{y} = 0$, and so the augmented matrix $\begin{bmatrix} \mathbf{V}_2\mathbf{V}_2^{\mathrm{T}} & \boldsymbol{\theta} - \boldsymbol{\theta}^\star \end{bmatrix}$ also has rank $d - n$, then the claim follows from the Rouche-Capelli Theorem.

To convince ourselves that the augmented matrix $\begin{bmatrix} \mathbf{V}_2\mathbf{V}_2^{\mathrm{T}} & \boldsymbol{\theta} - \boldsymbol{\theta}^\star \end{bmatrix}$ indeed has rank $d - n$, we first notice that since $\mathrm{rank}(\mathbf{X}\mathbf{X}^{\mathrm{T}}) = \mathrm{rank}(\mathbf{X})$ for any real matrix $\mathbf{X}$, we have $\mathrm{rank}(\mathbf{V}_2\mathbf{V}_2^{\mathrm{T}}) = \mathrm{rank}(\mathbf{V}_2) = d - n$, since the columns of $\mathbf{V}_2$ are orthogonal, so they are also independent. Furthermore, notice that

$$
\begin{aligned}
\mathbf{A}\mathbf{V}_2\mathbf{V}_2^{\mathrm{T}} &= \mathbf{U}\begin{bmatrix} \tilde{\boldsymbol{\Sigma}} & 0_{n\times(d-n)} \end{bmatrix}\begin{bmatrix} \mathbf{V}_1^{\mathrm{T}} \\ \mathbf{V}_2^{\mathrm{T}} \end{bmatrix}\mathbf{V}_2\mathbf{V}_2^{\mathrm{T}} \\
&= \mathbf{U}\begin{bmatrix} \tilde{\boldsymbol{\Sigma}} & 0_{n\times(d-n)} \end{bmatrix}\begin{bmatrix} 0_{n\times(d-n)} \\ \mathbf{I}_{d-n} \end{bmatrix}\mathbf{V}_2^{\mathrm{T}} \\
&= \mathbf{U}0_{n\times(d-n)}\mathbf{V}_2^{\mathrm{T}} \\
&= 0
\end{aligned}
$$

That is to say, the columns of $\mathbf{V}_2\mathbf{V}_2^{\mathrm{T}}$ span a $d - n$ dimensional subspace of vectors, where every vector in that subspace is a solution to $\mathbf{A}\mathbf{y} = 0$. By the rank-nullity theorem, we know that this subspace of homogeneous solutions is $d - n$ dimensional, so $\{\mathbf{y} : \mathbf{A}\mathbf{y} = 0\} = \mathrm{span}(\mathbf{V}_2\mathbf{V}_2^{\mathrm{T}})$, but $(\boldsymbol{\theta} - \boldsymbol{\theta}^\star) \in \{\mathbf{y} : \mathbf{A}\mathbf{y} = 0\}$ and so it does not add new information to $\mathbf{V}_2\mathbf{V}_2^{\mathrm{T}}$, hence the rank of the augmented matrix $\begin{bmatrix} \mathbf{V}_2\mathbf{V}_2^{\mathrm{T}} & \boldsymbol{\theta} - \boldsymbol{\theta}^\star \end{bmatrix}$ is $d - n$.

---

**Algorithm 1** Controlled ordinary linear regression.

Inputs: $\mathbf{A} \in \mathbb{R}^{n\times d}, \mathbf{b} \in \mathbb{R}^{n\times 1}, \alpha \in \mathbb{R}, \boldsymbol{\theta} \in \mathbb{R}^{d\times 1}$

$\text{---}, \quad \text{---}, \begin{bmatrix} \mathbf{V}_1^{\mathrm{T}} \\ \mathbf{V}_2^{\mathrm{T}} \end{bmatrix} \leftarrow \mathrm{SVD}(\mathbf{A})$

$\boldsymbol{\theta}^\star \leftarrow \mathbf{A}^{\mathrm{T}}(\mathbf{A}\mathbf{A}^{\mathrm{T}})^{-1}\mathbf{b}$

$\mathbf{z}_0 \leftarrow$ arbitrary

**for** iteration $k = 0, 1, \ldots$ until convergence **do**

$\quad \mathbf{z}_{k+1} \leftarrow \mathbf{z}_k - \alpha\mathbf{V}_2\mathbf{V}_2^{\mathrm{T}}(\mathbf{V}_2\mathbf{V}_2^{\mathrm{T}}\mathbf{z}_k - (\boldsymbol{\theta} - \boldsymbol{\theta}^\star))$

**end for**

$\mathbf{y}_0 \leftarrow \mathbf{z}_k$

**for** iteration $k = 0, 1, \ldots$ until convergence **do**

$\quad \mathbf{y}_{k+1} \leftarrow \mathbf{y}_k - \alpha\mathbf{A}^{\mathrm{T}}(\mathbf{A}\mathbf{y}_k - \mathbf{b})$

**end for**

output $\mathbf{y}_k$

---

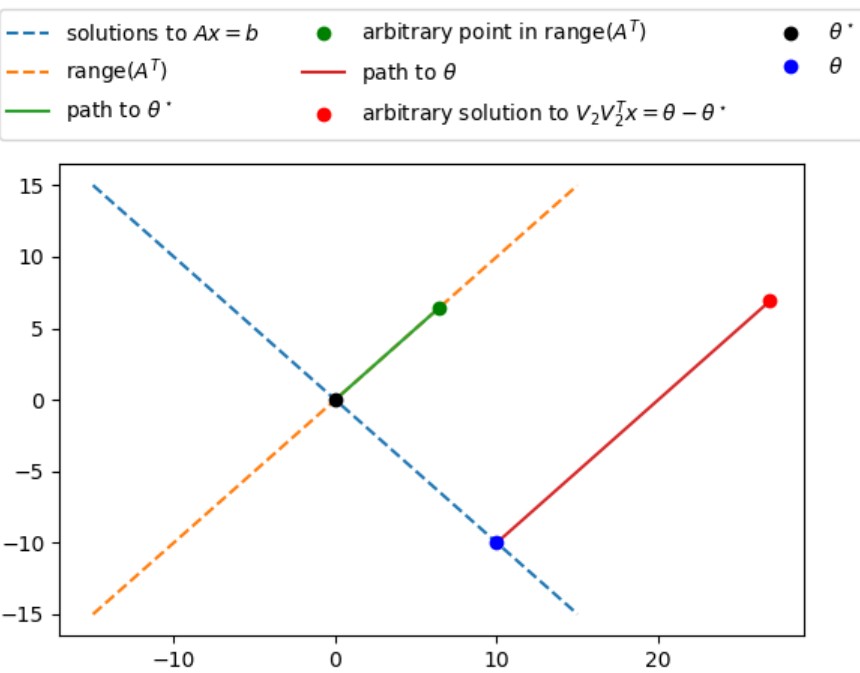

Figure 2.1: Illustration of solving $x + y = 0$ using two initializations. The desired solution was $(10, -10)$

This possibility of controlling which solution we converge to is summarized in Algorithm 1. In Figure 2.1 we illustrate it by solving the trivial system $x + y = 0$ with two initializations: one is a random point in $\mathbf{range}\left(\mathbf{A}^{\mathrm{T}}\right)$ and the other was the initial point suggested by Algorithm 1 when the desired solution was $(10, -10)$. We see that when initialized in $\mathbf{range}\left(\mathbf{A}^{\mathrm{T}}\right)$ the iteration converges to the minimum norm solution $(0, 0)$ and that Algorithm 1 works as intended.

The results of this subsection are striking, although simple, examples of the importance of initialization and the role it plays on regularization. Although we did not introduce an explicit regularization at any point, since we initialized in a clever way, we can reach the minimum norm solution. This is the central theme of this work. As a consequence of Corollary 3, we point out that $\mathbf{y}_0 = 0$ is always a good choice for an initial value if we want to converge to $\boldsymbol{\theta}^\star$ for depth-0 linear networks.

## 2.3 Deep linear networks

A deep linear network of depth $h$ is a machine learning model that attempts to fit $\mathbf{A}$ to $\mathbf{b}$ by finding $\mathbf{W}_1, \mathbf{W}_2, \ldots, \mathbf{W}_h \in \mathbb{R}^{d \times d}$ and $\mathbf{x} \in \mathbb{R}^{d \times 1}$ such that

$$\|\mathbf{A}\mathbf{W}_1\mathbf{W}_2\ldots\mathbf{W}_h\mathbf{x} - \mathbf{b}\|_2$$

is minimized. The minimization is done by gradient descent. The weights $\mathbf{W}_1, \mathbf{W}_2, \ldots, \mathbf{W}_h$ are called the hidden layer weights, and the amount of hidden layer weights is defined as the depth of the model. A deep linear network with depth $h$ has $hd^2 + d$ trainable weights to optimize.

In a non-linear network, the bigger $h$ is, the more expressive power the model has. In linear networks, the expressive power remains the same, but there is implicit acceleration at play (Arora et al., 2018), and the model can take a different path from the usual ordinary linear regression. Furthermore, the increase in $h$ over the ordinary linear regression model leads to the inclusion of saddle points (a trivial one is at $\mathbf{W}_i = 0, \mathbf{x} = 0$), which makes training more difficult in theory. In the following sections, we explore deep linear networks and how initialization affects their characteristics.

## 3 The Role of Initialization in Regularizing One Hidden Layer Linear Networks

In this section we consider the problem of learning both $\mathbf{W}$ and $\mathbf{x}$ such that

$$L_{\mathbf{A},\mathbf{b}}(\mathbf{W},\mathbf{x}) = \frac{1}{2}\|\mathbf{A}\mathbf{W}\mathbf{x} - \mathbf{b}\|_2^2 = \frac{1}{2}\|\mathbf{A}\mathbf{y} - \mathbf{b}\|_2^2$$

is minimized, where we define $\mathbf{y} := \mathbf{W}\mathbf{x}$. This is equivalent to linear neural network with a single hidden layer. Similar to the previous section, now the gradients are

$$
\begin{aligned}
\nabla_{\mathbf{x}} L_{\mathbf{A},\mathbf{b}}(\mathbf{W},\mathbf{x}) &= \mathbf{W}^{\mathrm{T}}\mathbf{A}^{\mathrm{T}}(\mathbf{A}\mathbf{W}\mathbf{x} - \mathbf{b}) \\
\nabla_{\mathbf{W}} L_{\mathbf{A},\mathbf{b}}(\mathbf{W},\mathbf{x}) &= \mathbf{A}^{\mathrm{T}}(\mathbf{A}\mathbf{W}\mathbf{x} - \mathbf{b})\mathbf{x}^{\mathrm{T}}
\end{aligned}
$$

and so the iteration step is

$$
\begin{aligned}
\mathbf{x}_{k+1} &= \mathbf{x}_k - \alpha\mathbf{W}_k^{\mathrm{T}}\mathbf{A}^{\mathrm{T}}(\mathbf{A}\mathbf{W}_k\mathbf{x}_k - \mathbf{b}) \\
\mathbf{W}_{k+1} &= \mathbf{W}_k - \alpha\mathbf{A}^{\mathrm{T}}(\mathbf{A}\mathbf{W}_k\mathbf{x}_k - \mathbf{b})\mathbf{x}_k^{\mathrm{T}}
\end{aligned}
$$

While the use of the hidden layer may seem redundant (since composition of linear functions is still linear), this model has highly non-trivial properties. One clear key difference is a dramatic increase in the level of overparameterization. Indeed, now there are $d^2 + d = O(d^2)$ trainable parameters. This is important, as modern machine learning theory is focused on the advantages in overparameterization (Arora et al. (2018); Bartlett et al. (2020); Belkin et al. (2019) and many others). Another important difference from the model in Section 2.3, is that similar to DNNs in any interesting setting, this loss function is non-convex. However, we see that this non-convexity does not harm deep networks too much, empirically (Sejnowski, 2020), as a good solution is often attained, and saddles and bad local minima are avoided.

We remark that single hidden layer networks have been studied extensively in the context of Neural Tangent Kernels (NTKs). These models operate in two different regimes, depending on the norm of initialization and how aggressive the overparameterization is in the hidden layer. If the deep model resembles it's linearization, then it operates in the kernel regime, also called lazy training. This shows that there is a connection between deep networks and linear networks, which provides additional motivation for our study.

Indeed, previous works on NTKs show that if the overparameterization (width of the network in this sense) is aggressive enough, and the weights have been initialized from a rotation-invariant distribution such as a standard normal distribution, then the law of large numbers assures us that the model is guaranteed to operate in this lazy regime and behave as a linear model, solving an under-determined system of equations, and has all the disadvantages of that model (like bad local minima, large norm solutions). Hence, while initialization does not matter for operating in this regime, it matters significantly from a generalization point of view when training the model. This is in line with our work, which emphasizes the importance of clever initializations. Furthermore, previous work on NTK also shows that one can choose to train an NTK by solving the linear system indirectly by kernel regression, which is a convex problem with a square matrix and a single solution, and there the initialization truly does not matter. Solving the kernel regression problem yields the minimum norm solution and is equivalent to solving the linear system with a row-space initialization. However, this does not preclude the possibility of other algorithms that will benefit from careful initialization.

The following lemma is an example of the similarities between the ordinary linear regression model of Section 2.3 and the one hidden layer model. It is the one hidden layer equivalent of Lemma 2, giving us an easy and good initialization for gradient descent to reach $\boldsymbol{\theta}^\star$.

**Lemma 5.** *If* $\mathbf{W}_k \in \mathbf{range}\left(\mathbf{A}^T\right)$ *for some* $k$ *then* $\mathbf{W}_{k+1} \in \mathbf{range}\left(\mathbf{A}^T\right)$.

*Proof.* In a similar way to Lemma 2, let $\mathbf{W}_k = \mathbf{A}^{\mathrm{T}}\mathbf{Z}$ for some $\mathbf{Z} \in \mathbb{R}^{n\times d}$, so

$$
\begin{aligned}
\mathbf{W}_{k+1} &= \mathbf{W}_k - \alpha\mathbf{A}^{\mathrm{T}}(\mathbf{A}\mathbf{W}_k\mathbf{x}_k - \mathbf{b})\mathbf{x}_k^{\mathrm{T}} \\
&= \mathbf{A}^{\mathrm{T}}(\mathbf{Z} - \alpha(\mathbf{A}\mathbf{A}^{\mathrm{T}}\mathbf{Z}\mathbf{x}_k - \mathbf{b})\mathbf{x}_k^{\mathrm{T}}) \in \mathbf{range}\left(\mathbf{A}^T\right)
\end{aligned}
$$

$\square$

As for the limit, suppose $\mathbf{W}_k = \mathbf{A}^{\mathrm{T}}\mathbf{Z}_k$ for all $k$, and that the limits $\mathbf{W}_\infty$ and $\mathbf{x}_\infty$ exist (which also means $\lim_{k\to\infty} \mathbf{Z}_k = \mathbf{Z}_\infty$ exists since $\mathbf{A}^{\mathrm{T}}$ has full column rank). This trivially gives us

$$
\begin{aligned}
\mathbf{W}_\infty &= \lim_{k\to\infty}[\mathbf{W}_k - \alpha\mathbf{A}^{\mathrm{T}}(\mathbf{A}\mathbf{W}_k\mathbf{x}_k - \mathbf{b})\mathbf{x}_k^{\mathrm{T}}] \\
&= \mathbf{A}^{\mathrm{T}}(\mathbf{Z}_\infty - \alpha(\mathbf{A}\mathbf{W}_\infty\mathbf{x}_\infty - \mathbf{b})\mathbf{x}_\infty^{\mathrm{T}})
\end{aligned}
$$

.

Since we also have $\mathbf{W}_\infty = \mathbf{A}^{\mathrm{T}}\mathbf{Z}_\infty$ we find that $(\mathbf{A}\mathbf{W}_\infty\mathbf{x}_\infty - \mathbf{b})\mathbf{x}_\infty^{\mathrm{T}} = 0$, so $\mathbf{x}_\infty \neq 0$ implies $\mathbf{A}\mathbf{W}_\infty\mathbf{x}_\infty = \mathbf{b}$. We lost convexity, so we are not sure we are converging to a solution (e.g. we could get stuck at a saddle point, a trivial example is $\mathbf{W}_0 = 0, \mathbf{x}_0 = 0$), but the above discussion shows that if we do converge to a solution, and $\mathbf{W}_0 \in \mathbf{range}\left(\mathbf{A}^{\mathrm{T}}\right)$, we are guaranteed to converge to $\boldsymbol{\theta}^\star$ regardless of what $\mathbf{x}_0$ was.

Having $\mathbf{W}_0 \in \mathbf{range}\left(\mathbf{A}^{\mathrm{T}}\right)$ and assuming that we converge to a solution assures us that $\lim_{k\to\infty} \mathbf{W}_k\mathbf{x}_k$ will be the minimal solution to the problem $\mathbf{A}\mathbf{x} = \mathbf{b}$, but each weight individually may not be optimal with respect to the other. That is, it is possible that $\mathbf{x}_\infty$ is not the optimal solution to problem $(\mathbf{A}\mathbf{W}_\infty)\mathbf{x} = \mathbf{b}$ and vice versa.

We refer to a solution pair $(\mathbf{W}, \mathbf{x})$ as *bi-optimal*, if $\mathbf{W}\mathbf{x}$ is a minimum norm solution of $\mathbf{A}\mathbf{z} = \mathbf{b}$, $\mathbf{W}$ is a minimum Frobenius norm solution of $\mathbf{A}\mathbf{Z}\mathbf{x} = \mathbf{b}$ where $\mathbf{Z}$ is the free parameter and $\mathbf{x}$ is a minimum norm solution to $\mathbf{A}\mathbf{W}\mathbf{z} = \mathbf{b}$. The following theorem provides us with a criterion on the initial $\mathbf{W}_0$ and $\mathbf{x}_0$ that ensures that $\mathbf{W}_\infty$ and $\mathbf{x}_\infty$, if they exist, are bi-optimal.

**Theorem 6** (bi-optimality). *Let $\mathbf{W}_0 = \mathbf{A}^{\mathrm{T}}\mathbf{v}_0\mathbf{x}_0^{T}$ for some $\mathbf{v}_0 \in \mathbb{R}^{n\times 1}$ and suppose $\mathbf{x}_k \neq 0$ for all $k$. Then for all $k$ there exists a $\mathbf{v}_k \in \mathbb{R}^{n\times 1}$ such that $\mathbf{W}_k = \mathbf{A}^{T}\mathbf{v}_k\mathbf{x}_k^{T}$.*

*Proof.* To make the proof more readable and less heavy on the use of subscripts, we make the following temporary change of notation:

$$
\begin{aligned}
\mathbf{W} &:= \mathbf{W}_0 \\
\mathbf{x} &:= \mathbf{x}_0 \\
\mathbf{Z} &:= \mathbf{W}_1 \\
\mathbf{y} &:= \mathbf{x}_1 \\
\mathbf{v} &:= \mathbf{v}_0 \\
\mathbf{r} &:= \mathbf{A}\mathbf{W}_0\mathbf{x}_0 - \mathbf{b}
\end{aligned}
$$

First, we write that the iteration step in the new notation is

$$
\begin{aligned}
\mathbf{Z} &= \mathbf{W} - \alpha\mathbf{A}^{\mathrm{T}}\mathbf{r}\mathbf{x}^{\mathrm{T}} \\
\mathbf{y} &= \mathbf{x} - \alpha\mathbf{W}^{\mathrm{T}}\mathbf{A}^{\mathrm{T}}\mathbf{r}
\end{aligned}
$$

then we claim that

$$
\mathbf{y}^{\mathrm{T}}\mathbf{y}\mathbf{Z} = \mathbf{Z}\mathbf{y}\mathbf{y}^{\mathrm{T}}
$$

To see this, we first write

$$
\begin{aligned}
\mathbf{y}^{\mathrm{T}}\mathbf{y}\mathbf{Z} &= (\mathbf{x}^{\mathrm{T}} - \alpha\mathbf{r}^{\mathrm{T}}\mathbf{A}\mathbf{W})(\mathbf{x} - \alpha\mathbf{W}^{\mathrm{T}}\mathbf{A}^{\mathrm{T}}\mathbf{r})(\mathbf{W} - \alpha\mathbf{A}^{\mathrm{T}}\mathbf{r}\mathbf{x}^{\mathrm{T}}) \\
&= (\mathbf{x}^{\mathrm{T}}\mathbf{x} - \alpha\mathbf{x}^{\mathrm{T}}\mathbf{W}^{\mathrm{T}}\mathbf{A}^{\mathrm{T}}\mathbf{r} - \alpha\mathbf{r}^{\mathrm{T}}\mathbf{A}\mathbf{W}\mathbf{x} + \alpha^2\mathbf{r}^{\mathrm{T}}\mathbf{A}\mathbf{W}\mathbf{W}^{\mathrm{T}}\mathbf{A}^{\mathrm{T}}\mathbf{r})(\mathbf{W} - \alpha\mathbf{A}^{\mathrm{T}}\mathbf{r}\mathbf{x}^{\mathrm{T}}) \\
&= (\mathbf{x}^{T}\mathbf{x})\cdot\mathbf{W} + (-\alpha\mathbf{x}^{T}\mathbf{x})\cdot\mathbf{A}^{\mathrm{T}}\mathbf{r}\mathbf{x}^{\mathrm{T}} + (-\alpha\mathbf{x}^{\mathrm{T}}\mathbf{W}^{\mathrm{T}}\mathbf{A}^{\mathrm{T}}\mathbf{r})\cdot\mathbf{W} + (\alpha^2\mathbf{x}^{\mathrm{T}}\mathbf{W}^{\mathrm{T}}\mathbf{A}^{\mathrm{T}}\mathbf{r})\cdot\mathbf{A}^{\mathrm{T}}\mathbf{r}\mathbf{x}^{\mathrm{T}} + \\
&\quad (-\alpha\mathbf{r}^{\mathrm{T}}\mathbf{A}\mathbf{W}\mathbf{x})\cdot\mathbf{W} + (\alpha^2\mathbf{r}^{\mathrm{T}}\mathbf{A}\mathbf{W}\mathbf{x})\cdot\mathbf{A}^{\mathrm{T}}\mathbf{r}\mathbf{x}^{\mathrm{T}} + (\alpha^2\mathbf{r}^{\mathrm{T}}\mathbf{A}\mathbf{W}\mathbf{W}^{\mathrm{T}}\mathbf{A}^{\mathrm{T}}\mathbf{r})\cdot\mathbf{W} + \\
&\quad (-\alpha^3\mathbf{r}^{\mathrm{T}}\mathbf{A}\mathbf{W}\mathbf{W}^{\mathrm{T}}\mathbf{A}^{\mathrm{T}}\mathbf{r})\cdot\mathbf{A}^{\mathrm{T}}\mathbf{r}\mathbf{x}^{\mathrm{T}}
\end{aligned}
$$

For reasons that will become clear, we will change the order of summation and instead write:

$$
\begin{aligned}
\mathbf{y}^{\mathrm{T}}\mathbf{y}\mathbf{Z} \;=\;& (\mathbf{x}^{\mathrm{T}}\mathbf{x}) \cdot \mathbf{W} + (-\alpha\mathbf{x}^{\mathrm{T}}\mathbf{W}^{\mathrm{T}}\mathbf{A}^{\mathrm{T}}\mathbf{r}) \cdot \mathbf{W} + (-\alpha\mathbf{r}^{\mathrm{T}}\mathbf{A}\mathbf{W}\mathbf{x}) \cdot \mathbf{W} + (\alpha^2\mathbf{r}^{\mathrm{T}}\mathbf{A}\mathbf{W}\mathbf{W}^{\mathrm{T}}\mathbf{A}^{\mathrm{T}}\mathbf{r}) \cdot \mathbf{W} \\
&+ (-\alpha\mathbf{x}^{\mathrm{T}}\mathbf{x}) \cdot \mathbf{A}^{\mathrm{T}}\mathbf{r}\mathbf{x}^{\mathrm{T}} + (\alpha^2\mathbf{x}^{\mathrm{T}}\mathbf{W}^{\mathrm{T}}\mathbf{A}^{\mathrm{T}}\mathbf{r}) \cdot \mathbf{A}^{\mathrm{T}}\mathbf{r}\mathbf{x}^{\mathrm{T}} + (\alpha^2\mathbf{r}^{\mathrm{T}}\mathbf{A}\mathbf{W}\mathbf{x}) \cdot \mathbf{A}^{\mathrm{T}}\mathbf{r}\mathbf{x}^{\mathrm{T}} + \\
&(-\alpha^3\mathbf{r}^{\mathrm{T}}\mathbf{A}\mathbf{W}\mathbf{W}^{\mathrm{T}}\mathbf{A}^{\mathrm{T}}\mathbf{r}) \cdot \mathbf{A}^{\mathrm{T}}\mathbf{r}\mathbf{x}^{\mathrm{T}}
\end{aligned}
$$

Similarly, write

$$
\begin{aligned}
\mathbf{Z}\mathbf{y}\mathbf{y}^{\mathrm{T}} \;=\;& (\mathbf{W} - \alpha\mathbf{A}^{\mathrm{T}}\mathbf{r}\mathbf{x}^{\mathrm{T}})(\mathbf{x} - \alpha\mathbf{W}^{\mathrm{T}}\mathbf{A}^{\mathrm{T}}\mathbf{r})(\mathbf{x}^{\mathrm{T}} - \alpha\mathbf{r}^{\mathrm{T}}\mathbf{A}\mathbf{W}) \\
=\;& (\mathbf{W} - \alpha\mathbf{A}^{\mathrm{T}}\mathbf{r}\mathbf{x}^{\mathrm{T}})(\mathbf{x}\mathbf{x}^{\mathrm{T}} - \alpha\mathbf{x}\mathbf{r}^{\mathrm{T}}\mathbf{A}\mathbf{W} - \alpha\mathbf{W}^{\mathrm{T}}\mathbf{A}^{\mathrm{T}}\mathbf{r}\mathbf{x}^{\mathrm{T}} + \alpha^2\mathbf{W}^{\mathrm{T}}\mathbf{A}^{\mathrm{T}}\mathbf{r}\mathbf{r}^{\mathrm{T}}\mathbf{A}\mathbf{W}) \\
=\;& \mathbf{W}\mathbf{x}\mathbf{x}^{\mathrm{T}} - \alpha\mathbf{W}\mathbf{x}\mathbf{r}^{\mathrm{T}}\mathbf{A}\mathbf{W} - \alpha\mathbf{W}\mathbf{W}^{\mathrm{T}}\mathbf{A}^{\mathrm{T}}\mathbf{r}\mathbf{x}^{\mathrm{T}} + \alpha^2\mathbf{W}\mathbf{W}^{\mathrm{T}}\mathbf{A}^{\mathrm{T}}\mathbf{r}\mathbf{r}^{\mathrm{T}}\mathbf{A}\mathbf{W} \\
&- \alpha\mathbf{A}^{\mathrm{T}}\mathbf{r}\mathbf{x}^{\mathrm{T}}\mathbf{x}\mathbf{x}^{\mathrm{T}} + \alpha^2\mathbf{A}^{\mathrm{T}}\mathbf{r}\mathbf{x}^{\mathrm{T}}\mathbf{x}\mathbf{r}^{\mathrm{T}}\mathbf{A}\mathbf{W} + \alpha^2\mathbf{A}^{\mathrm{T}}\mathbf{r}\mathbf{x}^{\mathrm{T}}\mathbf{W}^{\mathrm{T}}\mathbf{A}^{\mathrm{T}}\mathbf{r}\mathbf{x}^{\mathrm{T}} - \alpha^3\mathbf{A}^{\mathrm{T}}\mathbf{r}\mathbf{x}^{\mathrm{T}}\mathbf{W}^{\mathrm{T}}\mathbf{A}^{\mathrm{T}}\mathbf{r}\mathbf{r}^{\mathrm{T}}\mathbf{A}\mathbf{W}
\end{aligned}
$$

Both $\mathbf{y}^{\mathrm{T}}\mathbf{y}\mathbf{Z}$ and $\mathbf{Z}\mathbf{y}\mathbf{y}^{\mathrm{T}}$ have eight terms in their expressions. We claim that the equality is true, since each term is equal to its equivalent term (with respect to order) in the other expression.

1. First term:

$$
\begin{aligned}
\mathbf{W}\mathbf{x}\mathbf{x}^{\mathrm{T}} \;&=\; \mathbf{A}^{\mathrm{T}}\mathbf{v}\mathbf{x}^{\mathrm{T}}\mathbf{x}\mathbf{x}^{\mathrm{T}} \\
&=\; \mathbf{A}^{\mathrm{T}}\mathbf{v}(\mathbf{x}^{\mathrm{T}}\mathbf{x})\mathbf{x}^{\mathrm{T}} \\
&=\; (\mathbf{x}^{\mathrm{T}}\mathbf{x}) \cdot \mathbf{A}^{\mathrm{T}}\mathbf{v}\mathbf{x}^{\mathrm{T}} \\
&=\; (\mathbf{x}^{\mathrm{T}}\mathbf{x}) \cdot \mathbf{W}
\end{aligned}
$$

2. Second term:

$$
\begin{aligned}
-\alpha\mathbf{W}\mathbf{x}\mathbf{r}^{\mathrm{T}}\mathbf{A}\mathbf{W} \;&=\; -\alpha\mathbf{A}^{\mathrm{T}}\mathbf{v}\mathbf{x}^{\mathrm{T}}\mathbf{x}\mathbf{r}^{\mathrm{T}}\mathbf{A}\mathbf{A}^{\mathrm{T}}\mathbf{v}\mathbf{x}^{\mathrm{T}} \\
&=\; -\alpha\mathbf{A}^{\mathrm{T}}\mathbf{v}(\mathbf{x}^{\mathrm{T}}\mathbf{x})(\mathbf{r}^{\mathrm{T}}\mathbf{A}\mathbf{A}^{\mathrm{T}}\mathbf{v})\mathbf{x}^{\mathrm{T}} \\
&=\; -\alpha(\mathbf{x}^{\mathrm{T}}\mathbf{x})(\mathbf{r}^{\mathrm{T}}\mathbf{A}\mathbf{A}^{\mathrm{T}}\mathbf{v}) \cdot \mathbf{A}^{\mathrm{T}}\mathbf{v}\mathbf{x}^{\mathrm{T}} \\
&=\; -\alpha(\mathbf{x}^{\mathrm{T}}\mathbf{x})(\mathbf{r}^{\mathrm{T}}\mathbf{A}\mathbf{A}^{\mathrm{T}}\mathbf{v})^{\mathrm{T}} \cdot \mathbf{A}^{\mathrm{T}}\mathbf{v}\mathbf{x}^{\mathrm{T}} \\
&=\; -\alpha(\mathbf{x}^{\mathrm{T}}\mathbf{x})(\mathbf{v}^{\mathrm{T}}\mathbf{A}\mathbf{A}^{\mathrm{T}}\mathbf{r}) \cdot \mathbf{A}^{\mathrm{T}}\mathbf{v}\mathbf{x}^{\mathrm{T}} \\
&=\; (-\alpha\mathbf{x}^{\mathrm{T}})(\mathbf{x}\mathbf{v}^{\mathrm{T}}\mathbf{A})(\mathbf{A}^{\mathrm{T}}\mathbf{r}) \cdot \mathbf{A}^{\mathrm{T}}\mathbf{v}\mathbf{x}^{\mathrm{T}} \\
&=\; (-\alpha\mathbf{x}^{\mathrm{T}}\mathbf{W}^{\mathrm{T}}\mathbf{A}^{\mathrm{T}}\mathbf{r}) \cdot \mathbf{W}
\end{aligned}
$$

3. Third term:

$$
\begin{aligned}
-\alpha\mathbf{W}\mathbf{W}^{\mathrm{T}}\mathbf{A}^{\mathrm{T}}\mathbf{r}\mathbf{x}^{\mathrm{T}} \;&=\; -\alpha\mathbf{A}^{\mathrm{T}}\mathbf{v}\mathbf{x}^{\mathrm{T}}\mathbf{x}\mathbf{v}^{\mathrm{T}}\mathbf{A}\mathbf{A}^{\mathrm{T}}\mathbf{r}\mathbf{x}^{\mathrm{T}} \\
&=\; -\alpha\mathbf{A}^{\mathrm{T}}\mathbf{v}(\mathbf{x}^{\mathrm{T}}\mathbf{x})(\mathbf{v}^{\mathrm{T}}\mathbf{A}\mathbf{A}^{\mathrm{T}}\mathbf{r})\mathbf{x}^{\mathrm{T}} \\
&=\; -\alpha(\mathbf{v}^{\mathrm{T}}\mathbf{A}\mathbf{A}^{\mathrm{T}}\mathbf{r})(\mathbf{x}^{\mathrm{T}}\mathbf{x}) \cdot \mathbf{A}^{\mathrm{T}}\mathbf{v}\mathbf{x}^{\mathrm{T}} \\
&=\; -\alpha(\mathbf{v}^{\mathrm{T}}\mathbf{A}\mathbf{A}^{\mathrm{T}}\mathbf{r})^{\mathrm{T}}(\mathbf{x}^{\mathrm{T}}\mathbf{x}) \cdot \mathbf{A}^{\mathrm{T}}\mathbf{v}\mathbf{x}^{\mathrm{T}} \\
&=\; -\alpha(\mathbf{r}^{\mathrm{T}}\mathbf{A}\mathbf{A}^{\mathrm{T}}\mathbf{v})(\mathbf{x}^{\mathrm{T}}\mathbf{x}) \cdot \mathbf{A}^{\mathrm{T}}\mathbf{v}\mathbf{x}^{\mathrm{T}} \\
&=\; -\alpha(\mathbf{r}^{\mathrm{T}}\mathbf{A})(\mathbf{A}^{\mathrm{T}}\mathbf{v}\mathbf{x}^{\mathrm{T}})(\mathbf{x}) \cdot \mathbf{A}^{\mathrm{T}}\mathbf{v}\mathbf{x}^{\mathrm{T}} \\
&=\; (-\alpha\mathbf{r}^{\mathrm{T}}\mathbf{A}\mathbf{W}\mathbf{x}) \cdot \mathbf{W}
\end{aligned}
$$

4. Fourth term:

$$
\begin{aligned}
\alpha^2\mathbf{W}\mathbf{W}^{\mathrm{T}}\mathbf{A}^{\mathrm{T}}\mathbf{r}\mathbf{r}^{\mathrm{T}}\mathbf{A}\mathbf{W} \;&=\; \alpha^2\mathbf{A}^{\mathrm{T}}\mathbf{v}\mathbf{x}^{\mathrm{T}}\mathbf{x}\mathbf{v}^{\mathrm{T}}\mathbf{A}\mathbf{A}^{\mathrm{T}}\mathbf{r}\mathbf{r}^{\mathrm{T}}\mathbf{A}\mathbf{A}^{\mathrm{T}}\mathbf{v}\mathbf{x}^{\mathrm{T}} \\
&=\; \alpha^2\mathbf{A}^{\mathrm{T}}\mathbf{v}(\mathbf{x}^{\mathrm{T}}\mathbf{x})(\mathbf{v}^{\mathrm{T}}\mathbf{A}\mathbf{A}^{\mathrm{T}}\mathbf{r})(\mathbf{r}^{\mathrm{T}}\mathbf{A}\mathbf{A}^{\mathrm{T}}\mathbf{v})\mathbf{x}^{\mathrm{T}} \\
&=\; \alpha^2(\mathbf{r}^{\mathrm{T}}\mathbf{A}\mathbf{A}^{\mathrm{T}}\mathbf{v})(\mathbf{x}^{\mathrm{T}}\mathbf{x})(\mathbf{v}^{\mathrm{T}}\mathbf{A}\mathbf{A}^{\mathrm{T}}\mathbf{r}) \cdot \mathbf{A}^{\mathrm{T}}\mathbf{v}\mathbf{x}^{\mathrm{T}} \\
&=\; \alpha^2(\mathbf{r}^{\mathrm{T}}\mathbf{A})(\mathbf{A}^{\mathrm{T}}\mathbf{v}\mathbf{x}^{\mathrm{T}})(\mathbf{x}\mathbf{v}^{\mathrm{T}}\mathbf{A})(\mathbf{A}^{\mathrm{T}}\mathbf{r}) \cdot \mathbf{A}^{\mathrm{T}}\mathbf{v}\mathbf{x}^{\mathrm{T}} \\
&=\; (\alpha^2\mathbf{r}^{\mathrm{T}}\mathbf{A}\mathbf{W}\mathbf{W}^{\mathrm{T}}\mathbf{A}^{\mathrm{T}}\mathbf{r}) \cdot \mathbf{W}
\end{aligned}
$$

5. Fifth term:

$$
\begin{aligned}
-\alpha \mathbf{A}^{\mathrm{T}}\mathbf{r}\mathbf{x}^{\mathrm{T}}\mathbf{x}\mathbf{x}^{\mathrm{T}} &= -\alpha \mathbf{A}^{\mathrm{T}}\mathbf{r}(\mathbf{x}^{\mathrm{T}}\mathbf{x})\mathbf{x}^{\mathrm{T}} \\
&= (-\alpha \mathbf{x}^{\mathrm{T}}\mathbf{x}) \cdot \mathbf{A}^{\mathrm{T}}\mathbf{r}\mathbf{x}^{\mathrm{T}}
\end{aligned}
$$

6. Sixth term:

$$
\begin{aligned}
\alpha^2 \mathbf{A}^{\mathrm{T}}\mathbf{r}\mathbf{x}^{\mathrm{T}}\mathbf{x}\mathbf{r}^{\mathrm{T}}\mathbf{A}\mathbf{W} &= \alpha^2 \mathbf{A}^{\mathrm{T}}\mathbf{r}\mathbf{x}^{\mathrm{T}}\mathbf{x}\mathbf{r}^{\mathrm{T}}\mathbf{A}\mathbf{A}^{\mathrm{T}}\mathbf{v}\mathbf{x}^{\mathrm{T}} \\
&= \alpha^2 \mathbf{A}^{\mathrm{T}}\mathbf{r}(\mathbf{x}^{\mathrm{T}}\mathbf{x})(\mathbf{r}^{\mathrm{T}}\mathbf{A}\mathbf{A}^{\mathrm{T}}\mathbf{v})\mathbf{x}^{\mathrm{T}} \\
&= \alpha^2 (\mathbf{x}^{\mathrm{T}}\mathbf{x})(\mathbf{r}^{\mathrm{T}}\mathbf{A}\mathbf{A}^{\mathrm{T}}\mathbf{v}) \cdot \mathbf{A}^{\mathrm{T}}\mathbf{r}\mathbf{x}^{\mathrm{T}} \\
&= \alpha^2 (\mathbf{x}^{\mathrm{T}}\mathbf{x})(\mathbf{r}^{\mathrm{T}}\mathbf{A}\mathbf{A}^{\mathrm{T}}\mathbf{v})^{\mathrm{T}} \cdot \mathbf{A}^{\mathrm{T}}\mathbf{r}\mathbf{x}^{\mathrm{T}} \\
&= \alpha^2 (\mathbf{x}^{\mathrm{T}}\mathbf{x})(\mathbf{v}^{\mathrm{T}}\mathbf{A}\mathbf{A}^{\mathrm{T}}\mathbf{r}) \cdot \mathbf{A}^{\mathrm{T}}\mathbf{r}\mathbf{x}^{\mathrm{T}} \\
&= (\alpha^2 \mathbf{x}^{\mathrm{T}})(\mathbf{x}\mathbf{v}^{\mathrm{T}}\mathbf{A})(\mathbf{A}^{\mathrm{T}}\mathbf{r}) \cdot \mathbf{A}^{\mathrm{T}}\mathbf{r}\mathbf{x}^{\mathrm{T}} \\
&= (\alpha^2 \mathbf{x}^{\mathrm{T}}\mathbf{W}^{\mathrm{T}}\mathbf{A}^{\mathrm{T}}\mathbf{r}) \cdot \mathbf{A}^{\mathrm{T}}\mathbf{r}\mathbf{x}^{\mathrm{T}}
\end{aligned}
$$

7. Seventh term:

$$
\begin{aligned}
\alpha^2 \mathbf{A}^{\mathrm{T}}\mathbf{r}\mathbf{x}^{\mathrm{T}}\mathbf{W}^{\mathrm{T}}\mathbf{A}^{\mathrm{T}}\mathbf{r}\mathbf{x}^{\mathrm{T}} &= \alpha^2 \mathbf{A}^{\mathrm{T}}\mathbf{r}\mathbf{x}^{\mathrm{T}}\mathbf{x}\mathbf{v}^{\mathrm{T}}\mathbf{A}\mathbf{A}^{\mathrm{T}}\mathbf{r}\mathbf{x}^{\mathrm{T}} \\
&= \alpha^2 \mathbf{A}^{\mathrm{T}}\mathbf{r}(\mathbf{x}^{\mathrm{T}}\mathbf{x})(\mathbf{v}^{\mathrm{T}}\mathbf{A}\mathbf{A}^{\mathrm{T}}\mathbf{r})\mathbf{x}^{\mathrm{T}} \\
&= \alpha^2 (\mathbf{v}^{\mathrm{T}}\mathbf{A}\mathbf{A}^{\mathrm{T}}\mathbf{r})(\mathbf{x}^{\mathrm{T}}\mathbf{x}) \cdot \mathbf{A}^{\mathrm{T}}\mathbf{r}\mathbf{x}^{\mathrm{T}} \\
&= \alpha^2 (\mathbf{v}^{\mathrm{T}}\mathbf{A}\mathbf{A}^{\mathrm{T}}\mathbf{r})^{\mathrm{T}}(\mathbf{x}^{\mathrm{T}}\mathbf{x}) \cdot \mathbf{A}^{\mathrm{T}}\mathbf{r}\mathbf{x}^{\mathrm{T}} \\
&= \alpha^2 (\mathbf{r}^{\mathrm{T}}\mathbf{A}\mathbf{A}^{\mathrm{T}}\mathbf{v})(\mathbf{x}^{\mathrm{T}}\mathbf{x}) \cdot \mathbf{A}^{\mathrm{T}}\mathbf{r}\mathbf{x}^{\mathrm{T}} \\
&= \alpha^2 (\mathbf{r}^{\mathrm{T}}\mathbf{A})(\mathbf{A}^{\mathrm{T}}\mathbf{v}\mathbf{x}^{\mathrm{T}})(\mathbf{x}) \cdot \mathbf{A}^{\mathrm{T}}\mathbf{r}\mathbf{x}^{\mathrm{T}} \\
&= (\alpha^2 \mathbf{r}^{\mathrm{T}}\mathbf{A}\mathbf{W}\mathbf{x}) \cdot \mathbf{A}^{\mathrm{T}}\mathbf{r}\mathbf{x}^{\mathrm{T}}
\end{aligned}
$$

8. Eighth term:

$$
\begin{aligned}
-\alpha^3 \mathbf{A}^{\mathrm{T}}\mathbf{r}\mathbf{x}^{\mathrm{T}}\mathbf{W}^{\mathrm{T}}\mathbf{A}^{\mathrm{T}}\mathbf{r}\mathbf{r}^{\mathrm{T}}\mathbf{A}\mathbf{W} &= -\alpha^3 \mathbf{A}^{\mathrm{T}}\mathbf{r}\mathbf{x}^{\mathrm{T}}\mathbf{x}\mathbf{v}^{\mathrm{T}}\mathbf{A}\mathbf{A}^{\mathrm{T}}\mathbf{r}\mathbf{r}^{\mathrm{T}}\mathbf{A}\mathbf{A}^{\mathrm{T}}\mathbf{v}\mathbf{x}^{\mathrm{T}} \\
&= -\alpha^3 \mathbf{A}^{\mathrm{T}}\mathbf{r}(\mathbf{x}^{\mathrm{T}}\mathbf{x})(\mathbf{v}^{\mathrm{T}}\mathbf{A}\mathbf{A}^{\mathrm{T}}\mathbf{r})(\mathbf{r}^{\mathrm{T}}\mathbf{A}\mathbf{A}^{\mathrm{T}}\mathbf{v})\mathbf{x}^{\mathrm{T}} \\
&= -\alpha^3 (\mathbf{r}^{\mathrm{T}}\mathbf{A}\mathbf{A}^{\mathrm{T}}\mathbf{v})(\mathbf{x}^{\mathrm{T}}\mathbf{x})(\mathbf{v}^{\mathrm{T}}\mathbf{A}\mathbf{A}^{\mathrm{T}}\mathbf{r}) \cdot \mathbf{A}^{\mathrm{T}}\mathbf{r}\mathbf{x}^{\mathrm{T}} \\
&= -\alpha^3 (\mathbf{r}^{\mathrm{T}}\mathbf{A})(\mathbf{A}^{\mathrm{T}}\mathbf{v}\mathbf{x}^{\mathrm{T}})(\mathbf{x}\mathbf{v}^{\mathrm{T}}\mathbf{A})(\mathbf{A}^{\mathrm{T}}\mathbf{r}) \cdot \mathbf{A}^{\mathrm{T}}\mathbf{r}\mathbf{x}^{\mathrm{T}} \\
&= (-\alpha^3 \mathbf{r}^{\mathrm{T}}\mathbf{A}\mathbf{W}\mathbf{W}^{\mathrm{T}}\mathbf{A}^{\mathrm{T}}\mathbf{r}) \cdot \mathbf{A}^{\mathrm{T}}\mathbf{r}\mathbf{x}^{\mathrm{T}}
\end{aligned}
$$

This shows that $\mathbf{y}^{\mathrm{T}}\mathbf{y}\mathbf{Z} = \mathbf{Z}\mathbf{y}\mathbf{y}^{\mathrm{T}}$, which is another way of writing

$$
\mathbf{Z} = \frac{1}{\|\mathbf{y}\|_2^2}\mathbf{Z}\mathbf{y}\mathbf{y}^{\mathrm{T}}
$$

where we now used the assumption that $\mathbf{x}_k$ is never zero, and thus $\mathbf{y} \neq 0$ and this division is valid. To conclude the proof, let

$$
\mathbf{u} := \frac{1}{\|\mathbf{y}\|_2^2}(\mathbf{v} - \alpha \mathbf{r})\mathbf{x}^{\mathrm{T}}\mathbf{y}
$$

and check that

$$
\begin{aligned}
\mathbf{A}^{\mathrm{T}}\mathbf{u}\mathbf{y}^{\mathrm{T}} &= \frac{1}{\|\mathbf{y}\|_2^2}\mathbf{A}^{\mathrm{T}}(\mathbf{v} - \alpha \mathbf{r})\mathbf{x}^{\mathrm{T}}\mathbf{y}\mathbf{y}^{\mathrm{T}} \\
&= \frac{1}{\|\mathbf{y}\|_2^2}(\mathbf{W} - \alpha \mathbf{A}^{\mathrm{T}}\mathbf{r}\mathbf{x}^{\mathrm{T}})\mathbf{y}\mathbf{y}^{\mathrm{T}} \\
&= \frac{1}{\|\mathbf{y}\|_2^2}\mathbf{Z}\mathbf{y}\mathbf{y}^{\mathrm{T}} \\
&= \mathbf{Z}.
\end{aligned}
$$

Going back to our original notation, we have shown that if $\mathbf{W}_0 = \mathbf{A}^{\mathrm{T}} \mathbf{v}_0 \mathbf{x}_0^{\mathrm{T}}$ for some $\mathbf{v}_0$, then $\mathbf{W}_1 = \mathbf{A}^{\mathrm{T}} \mathbf{v}_1 \mathbf{x}_1^{\mathrm{T}}$ for some $\mathbf{v}_1$ which we called $\mathbf{u}$ and even gave an expression for. Applying this argument inductively exactly in the same way from iteration $k$ to iteration $k+1$, will give us the desired result. $\qquad\square$

Before proceeding, a short discussion on the case $\mathbf{x}_{k+1} = 0$, where an interesting phenomenon occurs, is in order. Suppose that at iteration $k$ the weights are the non-zero pair $(\mathbf{A}^{\mathrm{T}} \mathbf{v}_k \mathbf{x}_k^{\mathrm{T}}, \mathbf{x}_k^{\mathrm{T}})$. If $\mathbf{x}_{k+1} = 0$ then at iteration $k+1$ the weights are the pair $(\mathbf{A}^{\mathrm{T}} \mathbf{v}_k \mathbf{x}_k^{\mathrm{T}} - \alpha \mathbf{A}^{\mathrm{T}} \mathbf{r}_k \mathbf{x}_k^{\mathrm{T}}, 0)$, which shows that the theorem does not hold at this iteration. But we can notice that at iteration $k+2$ we have

$$
\begin{aligned}
\mathbf{x}_{k+2} &= \alpha(\mathbf{x}_k \mathbf{v}_k^{\mathrm{T}} \mathbf{A} - \alpha \mathbf{x}_k \mathbf{r}_k^{\mathrm{T}} \mathbf{A}) \mathbf{A}^{\mathrm{T}} \mathbf{b} \\
\mathbf{W}_{k+2} &= \mathbf{A}^{\mathrm{T}} \mathbf{v}_k \mathbf{x}_k^{\mathrm{T}} - \alpha \mathbf{A}^{\mathrm{T}} \mathbf{r}_k \mathbf{x}_k^{\mathrm{T}}
\end{aligned}
$$

and if we define $\mathbf{v}_{k+2} := \frac{1}{\|\mathbf{x}_{k+2}\|_2^2} \mathbf{A}^{\mathrm{T}^+} \mathbf{W}_{k+2} \mathbf{x}_{k+2}$ then

$$
\begin{aligned}
\mathbf{A}^{\mathrm{T}} \mathbf{v}_{k+2} \mathbf{x}_{k+2}^{\mathrm{T}} &= \frac{1}{\|\mathbf{x}_{k+2}\|_2^2} \mathbf{A}^{\mathrm{T}} \mathbf{A}^{\mathrm{T}^+} \mathbf{W}_{k+2} \mathbf{x}_{k+2} \mathbf{x}_{k+2}^{\mathrm{T}} \\
&= \frac{1}{\|\alpha(\mathbf{x}_k \mathbf{v}_k^{\mathrm{T}} \mathbf{A} - \alpha \mathbf{x}_k \mathbf{r}_k^{\mathrm{T}} \mathbf{A}) \mathbf{A}^{\mathrm{T}} \mathbf{b}\|_2^2} \mathbf{A}^{\mathrm{T}} \mathbf{A}^{\mathrm{T}^+} (\mathbf{A}^{\mathrm{T}} \mathbf{v}_k \mathbf{x}_k^{\mathrm{T}} - \alpha \mathbf{A}^{\mathrm{T}} \mathbf{r}_k \mathbf{x}_k^{\mathrm{T}}) \mathbf{x}_{k+2} \mathbf{x}_{k+2}^{\mathrm{T}} \\
&= \frac{1}{\|\alpha(\mathbf{x}_k \mathbf{v}_k^{\mathrm{T}} \mathbf{A} - \alpha \mathbf{x}_k \mathbf{r}_k^{\mathrm{T}} \mathbf{A}) \mathbf{A}^{\mathrm{T}} \mathbf{b}\|_2^2} \mathbf{A}^{\mathrm{T}} (\mathbf{v}_k \mathbf{x}_k^{\mathrm{T}} - \alpha \mathbf{r}_k \mathbf{x}_k^{\mathrm{T}}) \mathbf{x}_{k+2} \mathbf{x}_{k+2}^{\mathrm{T}} \\
&= \frac{\mathbf{A}^{\mathrm{T}} (\mathbf{v}_k \mathbf{x}_k^{\mathrm{T}} - \alpha \mathbf{r}_k \mathbf{x}_k^{\mathrm{T}}) \mathbf{x}_k (\mathbf{v}_k^{\mathrm{T}} \mathbf{A} - \alpha \mathbf{r}_k^{\mathrm{T}} \mathbf{A}) \mathbf{A}^{\mathrm{T}} \mathbf{b} \mathbf{b}^{\mathrm{T}} \mathbf{A} (\mathbf{A}^{\mathrm{T}} \mathbf{v}_k - \alpha \mathbf{A}^{\mathrm{T}} \mathbf{r}_k) \mathbf{x}_k^{\mathrm{T}}}{\|\mathbf{x}_k (\mathbf{v}_k^{\mathrm{T}} \mathbf{A} - \alpha \mathbf{r}_k^{\mathrm{T}} \mathbf{A}) \mathbf{A}^{\mathrm{T}} \mathbf{b}\|_2^2} \\
&= \frac{1}{\|\mathbf{x}_k\|_2^2} (\mathbf{A}^{\mathrm{T}} \mathbf{v}_k \mathbf{x}_k^{\mathrm{T}} - \alpha \mathbf{A}^{\mathrm{T}} \mathbf{r}_k \mathbf{x}_k^{\mathrm{T}}) \mathbf{x}_k \mathbf{x}_k^{\mathrm{T}} \\
&= \frac{1}{\|\mathbf{x}_k\|_2^2} (\mathbf{A}^{\mathrm{T}} \mathbf{v}_k - \alpha \mathbf{A}^{\mathrm{T}} \mathbf{r}_k)(\mathbf{x}_k^{\mathrm{T}} \mathbf{x}_k) \mathbf{x}_k^{\mathrm{T}} \\
&= \mathbf{A}^{\mathrm{T}} \mathbf{v}_k \mathbf{x}_k^{\mathrm{T}} - \alpha \mathbf{A}^{\mathrm{T}} \mathbf{r}_k \mathbf{x}_k^{\mathrm{T}} \\
&= \mathbf{W}_{k+2}
\end{aligned}
$$

which shows that in iteration $k+1$ we don't have the desired outcome, but one iteration later it course-corrects and we return to the pattern $\mathbf{W}_k = \mathbf{A}^{\mathrm{T}} \mathbf{v}_k \mathbf{x}_k^{\mathrm{T}}$.

The reason we refer to this theorem as bi-optimality is the following important corollary, which shows that if we initialize as required by Theorem 6, then bi-optimality is guaranteed.

**Corollary 7.** *If the conditions of Theorem 6 hold, and $\mathbf{W}_\infty, \mathbf{x}_\infty$ exist and are non-zero, then $\mathbf{A} \mathbf{W}_\infty \mathbf{x}_\infty = \mathbf{b}$ by the discussion following the proof of Lemma 5, and the following statements are true:*

1. *$\mathbf{W}_\infty \mathbf{x}_\infty$ is the minimum norm solution to the problem $\mathbf{A} \mathbf{z} = \mathbf{b}$*

2. *$\mathbf{x}_\infty$ is the minimum norm solution to the problem $(\mathbf{A} \mathbf{W}_\infty) \mathbf{z} = \mathbf{b}$*

3. *vec$(\mathbf{W}_\infty)$ is the minimum norm solution to the problem $(\mathbf{x}_\infty^{\mathrm{T}} \otimes \mathbf{A}) \mathbf{z} = \mathbf{b}$, (i.e. $\mathbf{W}_\infty$ is the minimum Frobenius norm solution to $\mathbf{A} \mathbf{Z} \mathbf{x}_\infty = \mathbf{b}$).*

*Proof.* 1. The conditions of Theorem 6 are assumed to hold, so for all $k$ $\mathbf{x}_k \neq 0$ and there exists $\mathbf{v}_k$ such that $\mathbf{W}_k = \mathbf{A}^{\mathrm{T}} \mathbf{v}_k \mathbf{x}_k^{\mathrm{T}}$. First, we mention that $\mathbf{v}_\infty$ must exist:

$$
\begin{aligned}
\mathbf{v}_\infty &= \lim_{k \to \infty} \frac{1}{\|\mathbf{x}_k\|_2^2} \mathbf{I}_n \mathbf{v}_k \mathbf{x}_k^{\mathrm{T}} \mathbf{x}_k \\
&= \lim_{k \to \infty} \frac{1}{\|\mathbf{x}_k\|_2^2} (\mathbf{A}^{\mathrm{T}^+} \mathbf{A}^{\mathrm{T}}) \mathbf{v}_k \mathbf{x}_k^{\mathrm{T}} \mathbf{x}_k \\
&= \lim_{k \to \infty} \frac{1}{\|\mathbf{x}_k\|_2^2} \mathbf{A}^{\mathrm{T}^+} (\mathbf{A}^{\mathrm{T}} \mathbf{v}_k \mathbf{x}_k^{\mathrm{T}}) \mathbf{x}_k \\
&= \lim_{k \to \infty} \frac{1}{\|\mathbf{x}_k\|_2^2} \mathbf{A}^{\mathrm{T}^+} \mathbf{W}_k \mathbf{x}_k \\
&= \frac{1}{\|\mathbf{x}\|_2^2} \mathbf{A}^{\mathrm{T}^+} \mathbf{W}_\infty \mathbf{x}_\infty .
\end{aligned}
$$

We emphasize that $\mathbf{A}$ has full rank and so $\mathbf{I}_n = \mathbf{A}^{\mathrm{T}+} \mathbf{A}^{\mathrm{T}}$. From the existence of the limits $\mathbf{W}_\infty, \mathbf{x}_\infty, \mathbf{v}_\infty$, we can now safely deduce by simple multiplication.

$$
\mathbf{W}_\infty = \lim_{k \to \infty} \mathbf{W}_k = \lim_{k \to \infty} \mathbf{A}^{\mathrm{T}} \mathbf{v}_k \mathbf{x}_k^{\mathrm{T}} = \mathbf{A}^{\mathrm{T}} \mathbf{v}_\infty \mathbf{x}_\infty^{\mathrm{T}}
$$

So $\mathbf{W}_\infty \mathbf{x}_\infty \in \mathbf{range}\left(\mathbf{A}^{\mathrm{T}}\right)$ and $\mathbf{A} \mathbf{W}_\infty \mathbf{x}_\infty = \mathbf{b}$, and due to Lemma 1 we have $\boldsymbol{\theta}^\star = \mathbf{W}_\infty \mathbf{x}_\infty$.

2. $\mathbf{x}_\infty$ is trivially a solution of $(\mathbf{A}\mathbf{W}_\infty)\mathbf{z} = \mathbf{b}$. To see that it is the minimum norm solution, we prove that $\mathbf{x}_\infty \in \mathbf{range}\left((\mathbf{A}\mathbf{W}_\infty)^{\mathrm{T}}\right)$ by exploiting the facts that $\mathbf{W} = \mathbf{A}^{\mathrm{T}} \mathbf{v}_\infty \mathbf{x}_\infty^{\mathrm{T}}$ and $\mathbf{v}_\infty \neq 0$, as that would imply $\mathbf{b} = 0$:

$$
\begin{aligned}
\mathbf{x}_\infty &= \mathbf{x}_\infty \frac{(\mathbf{v}_\infty^{\mathrm{T}} \mathbf{A}\mathbf{A}^{\mathrm{T}})(\mathbf{A}\mathbf{A}^{\mathrm{T}} \mathbf{v}_\infty)}{\|\mathbf{A}\mathbf{A}^{\mathrm{T}} \mathbf{v}_\infty\|_2^2} \\
&= \frac{(\mathbf{x}_\infty \mathbf{v}_\infty^{\mathrm{T}} \mathbf{A}) \mathbf{A}^{\mathrm{T}} \mathbf{A}\mathbf{A}^{\mathrm{T}} \mathbf{v}_\infty}{\|\mathbf{A}\mathbf{A}^{\mathrm{T}} \mathbf{v}_\infty\|_2^2} \\
&= \frac{\mathbf{W}_\infty^{\mathrm{T}} \mathbf{A}^{\mathrm{T}} \mathbf{A}\mathbf{A}^{\mathrm{T}} \mathbf{v}_\infty}{\|\mathbf{A}\mathbf{A}^{\mathrm{T}} \mathbf{v}_\infty\|_2^2} \\
&= (\mathbf{A}\mathbf{W}_\infty)^{\mathrm{T}} \left( \frac{1}{\|\mathbf{A}\mathbf{A}^{\mathrm{T}} \mathbf{v}_\infty\|_2^2} \mathbf{A}\mathbf{A}^{\mathrm{T}} \mathbf{v}_\infty \right) \in \mathbf{range}\left((\mathbf{A}\mathbf{W}_\infty)^{\mathrm{T}}\right)
\end{aligned}
$$

Now we apply Lemma 1.

3. $\mathbf{W}_\infty = \mathbf{A}^{\mathrm{T}} \mathbf{v}_\infty \mathbf{x}_\infty^{\mathrm{T}}$ implies by the properties of the Kronecker product that

$$
\begin{aligned}
\mathrm{vec}(\mathbf{W}_\infty) = \mathrm{vec}(\mathbf{A}^{\mathrm{T}} \mathbf{v}_\infty \mathbf{x}_\infty^{\mathrm{T}}) &= (\mathbf{x}_\infty \otimes \mathbf{A}^{\mathrm{T}}) \mathbf{v}_\infty \in \mathbf{range}\left(\mathbf{x}_\infty \otimes \mathbf{A}^{\mathrm{T}}\right) \\
&= \mathbf{range}\left((\mathbf{x}_\infty^{\mathrm{T}} \otimes \mathbf{A})^{\mathrm{T}}\right) .
\end{aligned}
$$

Combine this with the fact that

$$
\begin{aligned}
(\mathbf{x}_\infty^{\mathrm{T}} \otimes \mathbf{A}) \mathrm{vec}(\mathbf{W}_\infty) &= \mathrm{vec}(\mathbf{A} \mathbf{W}_\infty \mathbf{x}_\infty) \\
&= \mathbf{A} \mathbf{W}_\infty \mathbf{x}_\infty \\
&= \mathbf{b}
\end{aligned}
$$

and apply Lemma 1 to conclude the proof

$\square$

Even in this somewhat non-trivial model we see that initialization plays an important role, and if we initialize intelligently, not only can we reach the minimum norm solution $\boldsymbol{\theta}^\star$ that we desired, but we can factor it into

**Wx** such that each variable is the minimal solution with respect to the other, a sort of bi-optimality where every variable is optimal and no variable has any incentive to move. Of course, since we are still converging to $\boldsymbol{\theta}^\star$, this method does not offer better generalization than other methods that converge to $\boldsymbol{\theta}^\star$, though we hope that further research will yield useful properties of bi-optimal solutions.

An additional nice thing about Theorem 6 is that since $\mathbf{W}_k = \mathbf{A}^{\mathrm{T}}\mathbf{v}_k\mathbf{x}_k^{\mathrm{T}}$ in every iteration, instead of optimizing over $\mathbf{W}_k$, we can instead iterate over $\mathbf{v}_k$. The update step for $\mathbf{x}_k$ remains the same (except that we replace $\mathbf{W}_k$ with $\mathbf{A}^{\mathrm{T}}\mathbf{v}_k\mathbf{x}_k^{\mathrm{T}}$), and the update step for $\mathbf{v}_k$ is as denoted in the theorem. See Algorithm 2 for a pseudocode description.

---

**Algorithm 2** One hidden layer bi-optimal network

---

Inputs: $\mathbf{A} \in \mathbb{R}^{n \times d}, \mathbf{b} \in \mathbb{R}^{n \times 1}, \alpha \in \mathbb{R}$
$\mathbf{x}_0 \leftarrow$ arbitrary, not zero
$\mathbf{v}_0 \leftarrow$ arbitrary, not zero
**for** iteration $k = 0, 1, \dots$ until convergence **do**
$\quad \mathbf{x}_{k+1} \leftarrow \mathbf{x}_k - \alpha\mathbf{x}_k\mathbf{v}_k^{\mathrm{T}}\mathbf{A}\mathbf{A}^{\mathrm{T}}(\mathbf{A}\mathbf{A}^{\mathrm{T}}\mathbf{v}_k\mathbf{x}_k^{\mathrm{T}}\mathbf{x}_k - \mathbf{b})$
$\quad \mathbf{v}_{k+1} \leftarrow \frac{1}{\|\mathbf{x}_{k+1}\|_2^2}(\mathbf{v}_k - \alpha(\mathbf{A}\mathbf{A}^{\mathrm{T}}\mathbf{v}_k\mathbf{x}_k^{\mathrm{T}}\mathbf{x}_k - \mathbf{b}))\mathbf{x}_k^{\mathrm{T}}\mathbf{x}_{k+1}$
**end for**
output $\mathbf{A}^{\mathrm{T}}\mathbf{v}_k\mathbf{x}_k^{\mathrm{T}}, \mathbf{x}_k$

---

We reduced the dimensions of our variables from $d^2 + d$ to $d + n$, but we can improve further! Notice that

$$\gamma_k := \mathbf{v}_k^{\mathrm{T}}\mathbf{A}\mathbf{A}^{\mathrm{T}}(\|\mathbf{x}_k\|_2^2\mathbf{A}\mathbf{A}^{\mathrm{T}}\mathbf{v}_k - \mathbf{b})$$

is a scalar, so we can write

$$
\begin{aligned}
\mathbf{x}_{k+1} = (1 - \mathbf{v}_k^{\mathrm{T}}\mathbf{A}\mathbf{A}^{\mathrm{T}}(\|\mathbf{x}_k\|_2^2\mathbf{A}\mathbf{A}^{\mathrm{T}}\mathbf{v}_k - \mathbf{b}))\mathbf{x}_k &= (1 - \gamma_k)\mathbf{x}_k \\
&= \prod_{i=0}^{k}(1 - \gamma_i)\mathbf{x}_0 \\
&= \rho_{k+1}\mathbf{x}_0
\end{aligned}
$$

where $\rho_k := \prod_{i=0}^{k-1}(1 - \gamma_i)$. Now,

$$
\begin{aligned}
\mathbf{v}_{k+1} &= \frac{1}{\rho_{k+1}^2\|x_0\|_2^2}(\mathbf{v}_k - \alpha(\rho_k^2\|\mathbf{x}_0\|_2^2\mathbf{A}\mathbf{A}^{\mathrm{T}}\mathbf{v}_k - \mathbf{b}))\rho_k^2(1 - \gamma_k)\|\mathbf{x}_0\|_2^2 \\
&= \frac{1}{1 - \gamma_k}(\mathbf{v}_k - \alpha(\rho_k\|\mathbf{x}_0\|_2^2\mathbf{A}\mathbf{A}^{\mathrm{T}}\mathbf{v}_k - \mathbf{b}))\|\mathbf{x}_0\|_2^2.
\end{aligned}
$$

Since $\mathbf{x}_0$ is arbitrary, to simplify matters, we can sample $\mathbf{x}_0$ from the unit sphere, and arrive at the concise update step

$$\mathbf{v}_{k+1} = \frac{1}{1 - \gamma_k}(\mathbf{v}_k - \alpha(\rho_k\mathbf{A}\mathbf{A}^{\mathrm{T}}\mathbf{v}_k - \mathbf{b})).$$

Notice that we do not even need to iterate on $\mathbf{x}_k$, but rather only on $\gamma_k$, the product $\rho_k$, and $\mathbf{v}_k$. The number of parameters is now $n + 2$, which we remind the reader is possibly much lower than $d$, especially in the setting we consider as $d > n$, and probably $d \gg n$ in many real-world settings. This yields the following very interesting algorithm (Algorithm 3).

The time complexity of running Algorithm 3 for $t > 1$ iterations is $O(t \cdot \max(n, T_{\mathbf{A}}))$, where we abstract the time it takes to multiply by $\mathbf{A}$ or $\mathbf{A}^{\mathrm{T}}$ by $T_{\mathbf{A}}$. One final observation regarding this algorithm is that since

$$\lim_{k \to \infty} \rho_k\mathbf{A}^{\mathrm{T}}\mathbf{v}_k = \boldsymbol{\theta}^\star = \mathbf{A}^{\mathrm{T}}(\mathbf{A}\mathbf{A}^{\mathrm{T}})^{-1}\mathbf{b}$$

and $\mathbf{A}^{\mathrm{T}}$ has full column rank and thus has a left inverse, is that we know ahead of time that

$$\mathbf{v}_\infty = \lim_{k \to \infty} \frac{1}{\rho_k}(\mathbf{A}\mathbf{A}^{\mathrm{T}})^{-1}\mathbf{b}.$$

---

**Algorithm 3** Compact hidden layer iteration

---

$\mathbf{A} \in \mathbb{R}^{n \times d}, \mathbf{b} \in \mathbb{R}^{n \times 1}, \alpha \in \mathbb{R}$ inputs
$\mathbf{v}_0 \leftarrow$ arbitrary, not zero          $O(n)$
$\rho_0 \leftarrow 1$          $O(1)$
$\mathbf{z}_0 \leftarrow \mathbf{A}^\mathrm{T} \mathbf{v}_0$          $O(T_\mathbf{A})$
**for** iteration $k = 0, 1, \ldots$ until convergence **do**
     $\mathbf{y}_k \leftarrow \mathbf{A} \mathbf{z}_k$          $O(T_\mathbf{A})$
     $\mathbf{r}_k \leftarrow \alpha(\rho_k^2 \mathbf{y}_k - \mathbf{b})$          $O(n)$
     $\gamma_k \leftarrow \mathbf{y}_k^\mathrm{T} \mathbf{r}_k$          $O(n)$
     $\mathbf{v}_{k+1} \leftarrow \frac{1}{1-\gamma_k}(\mathbf{v}_k - \mathbf{r}_k)$          $O(n)$
     $\rho_{k+1} \leftarrow \rho_k(1 - \gamma_k)$          $O(1)$
     $\mathbf{z}_{k+1} \leftarrow \mathbf{A}^\mathrm{T} \mathbf{v}_{k+1}$          $O(T_\mathbf{A})$
**end for**
output $\rho_k^2 \mathbf{z}_k$

---

Of course $(\mathbf{A}\mathbf{A}^\mathrm{T})^{-1}\mathbf{b}$ is the unique solution to the problem $\mathbf{A}\mathbf{A}^\mathrm{T}\mathbf{z} = \mathbf{b}$, which is a highly ill-conditioned problem in most scenarios as $\kappa(\mathbf{A}\mathbf{A}^\mathrm{T}) = \kappa(\mathbf{A})^2$. So this algorithm aims to solve $\mathbf{A}\mathbf{x} = \mathbf{b}$ by way of solving $\mathbf{A}\mathbf{A}^\mathrm{T}\mathbf{x} = \mathbf{b}$ with a variable step size, but at the very low cost of optimizing a single layer, plus an additional scalar. This shows that given an intelligent initialization, not only can we converge to the best solution, we can collapse deep networks to have the same per iteration cost of shallow networks, up to a constant factor.

This is in the same spirit as the lottery ticket hypothesis (LTH, (Frankle & Carbin, 2019)), but there are two key differences. The crux of the LTH is that at initialization, a randomly initialized neural network contains a sub-network that if trained in isolation will reach the same or similar accuracy to the complete network after training, and propose an iterative method for finding this sub-network. The similarity is clear in that we reduce the cost of per iteration of tranining/testing, but the differences are that we propose an a priori method for reducing the number of parameters and faster iterations, rather than an a posteriori iterative one. Furthermore, our collapsed model is not composed of sub-weights of the original model, but rather constraining the weights to be of a particular low rank form and optimizing the respective vectors that make up this decomposition.

This algorithm is completely equivalent to a single hidden layer neural network, but does not give any advantages in generalization. Does it have any advantages when it comes to optimization? Recent work (Arora et al., 2018) suggests that overparameterization has advantages when it comes to optimization, and that depth preconditions the problem. However, to the best of our knowledge, they did not consider an underdetermined system, which is exactly our setting. We empirically test this idea in an underdetermined setting. See Section 4.1.

## 4 The Role of Initialization in Deep Linear Networks

### 4.1 Collapsing two hidden layers linear networks

In this section, we consider the task of finding $\mathbf{W}_1, \mathbf{W}_2, \ldots, \mathbf{W}_h, \mathbf{x}$ such that

$$L_{\mathbf{A},\mathbf{b}}(\mathbf{W}_1, \mathbf{W}_2, \ldots, \mathbf{W}_h, \mathbf{x}) = \frac{1}{2}\|\mathbf{A}\mathbf{W}_1\mathbf{W}_2 \ldots \mathbf{W}_h\mathbf{x} - \mathbf{b}\|_2^2 = \frac{1}{2}\|\mathbf{A}\mathbf{y} - \mathbf{b}\|_2^2$$

is minimized where $h > 1$ and we define $\mathbf{y} := \mathbf{W}_1\mathbf{W}_2 \ldots \mathbf{W}_h\mathbf{x}$. As one can expect, this model shares many properties with the previous two models. Gradients are

$$\nabla_{\mathbf{W}_j} L_{\mathbf{A},\mathbf{b}}(\mathbf{W}_1, \mathbf{W}_2, \ldots, \mathbf{W}_h, \mathbf{x}) = \begin{cases} \mathbf{W}_{j-1}^\mathrm{T} \ldots \mathbf{W}_1^\mathrm{T} \mathbf{A}^\mathrm{T}(\mathbf{A}\mathbf{W}_1 \ldots \mathbf{W}_h\mathbf{x} - \mathbf{b})\mathbf{x}^\mathrm{T}\mathbf{W}_h^\mathrm{T} \ldots \mathbf{W}_{j+1}^\mathrm{T}, & 1 < j < h \\ (\mathbf{A}\mathbf{W}_1 \ldots \mathbf{W}_h\mathbf{x} - \mathbf{b})\mathbf{x}^\mathrm{T}\mathbf{W}_h^\mathrm{T} \ldots \mathbf{W}_2^\mathrm{T}, & j = 1 \\ \mathbf{W}_{h-1}^\mathrm{T} \ldots \mathbf{W}_1^\mathrm{T} \mathbf{A}^\mathrm{T}(\mathbf{A}\mathbf{W}_1 \ldots \mathbf{W}_h\mathbf{x} - \mathbf{b}), & j = h \end{cases}$$

and

$$\nabla_{\mathbf{x}} L_{\mathbf{A},\mathbf{b}}(\mathbf{W}_1, \mathbf{W}_2, \ldots, \mathbf{W}_h, \mathbf{x}) = \mathbf{W}_h^{\mathrm{T}} \mathbf{W}_{h-1}^{\mathrm{T}} \ldots \mathbf{W}_1^{\mathrm{T}} \mathbf{A}^{\mathrm{T}} (\mathbf{A} \mathbf{W}_1 \mathbf{W}_2 \ldots \mathbf{W}_h \mathbf{x} - \mathbf{b}).$$

The iteration step is

$$\mathbf{W}_j^{(k+1)} = \mathbf{W}_j^{(k)} - \alpha \nabla_{\mathbf{W}_j} L_{\mathbf{A},\mathbf{b}}(\mathbf{W}_1^{(k)}, \mathbf{W}_2^{(k)}, \ldots, \mathbf{W}_h^{(k)}, \mathbf{x}_k)$$
$$\mathbf{x}_{k+1} = \mathbf{x}_k - \alpha \nabla_{\mathbf{x}} L_{\mathbf{A},\mathbf{b}}(\mathbf{W}_1^{(k)}, \mathbf{W}_2^{(k)}, \ldots, \mathbf{W}_h^{(k)}, \mathbf{x}_k)$$

which leads us to this next very familiar lemma, which generalizes the previous 0-layer and 1-layer results to arbitrary amount of layers. As one can expect, the property of the first weight being in the row-space of $\mathbf{A}$ is conserved throughout gradient descent iterations.

**Lemma 8.** *If* $\mathbf{W}_1^{(k)} \in \mathbf{range}\left(\mathbf{A}^T\right)$ *for some k, then* $\mathbf{W}_1^{(k+1)} \in \mathbf{range}\left(\mathbf{A}^T\right)$.

*Proof.* Identical to the proof of Lemma 5 with the respective change in matrices. □

Unsurprisingly, the critical importance of initialization remains for deep models and is even exacerbated. Although, naturally, many properties are shared with the model in Section 3, some things are also different.

The following series of results prove it is possible to collapse a linear network with $h = 2$ and outline how to do so. We start with Lemma 9, which describes how to initialize in a way that we will later use to achieve bi-optimality in this model. This lemma is the basis for an induction we later use to prove Theorem 12.

**Lemma 9.** *When* $h = 2$, *it is possible to find* $\mathbf{W}_1, \mathbf{W}_2, \mathbf{x}$ *all non-zero such that* $\mathbf{W}_1 = \mathbf{A}^T \mathbf{v} \mathbf{x}^T \mathbf{W}_2^T$ *and* $\mathbf{W}_2 = \mathbf{W}_1^T \mathbf{A}^T \mathbf{u} \mathbf{x}^T$ *and* $\mathbf{x} \in \mathbf{range}\left(\mathbf{W}_2^T \mathbf{W}_1^T \mathbf{A}^T\right)$ *for some* $\mathbf{v} \in \mathbb{R}^{n \times 1}$ *and* $\mathbf{u} \in \mathbb{R}^{n \times 1}$

*Proof.* Let $\mathbf{v}$ be any non-zero vector. Set $\mathbf{x} = \frac{\mathbf{A}^{\mathrm{T}} \mathbf{v}}{\|\mathbf{A}^{\mathrm{T}} \mathbf{v}\|_2^2}$, $\mathbf{u} = \frac{\mathbf{A} \mathbf{A}^{\mathrm{T}} \mathbf{v}}{\|\mathbf{A} \mathbf{A}^{\mathrm{T}} \mathbf{v}\|_2^2}$, and

$$\mathbf{W}_1 = \begin{pmatrix} | & 0 & 0 & \ldots & 0 \\ \mathbf{A}^{\mathrm{T}} \mathbf{v} & \vdots & \vdots & \ldots & \vdots \\ | & 0 & 0 & \ldots & 0 \end{pmatrix}.$$

Notice that $\mathbf{A}^{\mathrm{T}} \mathbf{v}, \mathbf{A} \mathbf{A}^{\mathrm{T}} \mathbf{v} \neq 0$ from the rank-nullity theorem. Finally set $\mathbf{W}_2 = \mathbf{W}_1^{\mathrm{T}} \mathbf{A}^{\mathrm{T}} \mathbf{u} \mathbf{x}^{\mathrm{T}}$. Let us verify the other conditions:

$$
\begin{aligned}
\mathbf{A}^{\mathrm{T}} \mathbf{v} \mathbf{x}^{\mathrm{T}} \mathbf{W}_2^{\mathrm{T}} &= \mathbf{A}^{\mathrm{T}} \mathbf{v} \mathbf{x}^{\mathrm{T}} \mathbf{x} \mathbf{u}^{\mathrm{T}} \mathbf{A} \mathbf{W}_1 \\
&= \mathbf{A}^{\mathrm{T}} \mathbf{v} \mathbf{u}^{\mathrm{T}} \mathbf{A} \mathbf{W}_1 \\
&= ((\mathbf{A}^{\mathrm{T}} \mathbf{v})(\mathbf{A}^{\mathrm{T}} \mathbf{u})^{\mathrm{T}}) \mathbf{W}_1 \\
&= \begin{pmatrix} | & 0 & 0 & \ldots & 0 \\ ((\mathbf{A}^{\mathrm{T}} \mathbf{v})^{\mathrm{T}} (\mathbf{A}^{\mathrm{T}} \mathbf{u})) \mathbf{A}^{\mathrm{T}} \mathbf{v} & \vdots & \vdots & \ldots & \vdots \\ | & 0 & 0 & \ldots & 0 \end{pmatrix} \\
&= \begin{pmatrix} | & 0 & 0 & \ldots & 0 \\ \frac{1}{\|\mathbf{A} \mathbf{A}^{\mathrm{T}} \mathbf{v}\|_2^2} ((\mathbf{A}^{\mathrm{T}} \mathbf{v})^{\mathrm{T}} (\mathbf{A}^{\mathrm{T}} \mathbf{A} \mathbf{A}^{\mathrm{T}} \mathbf{v})) \mathbf{A}^{\mathrm{T}} \mathbf{v} & \vdots & \vdots & \ldots & \vdots \\ | & 0 & 0 & \ldots & 0 \end{pmatrix} \\
&= \begin{pmatrix} | & 0 & 0 & \ldots & 0 \\ \mathbf{A}^{\mathrm{T}} \mathbf{v} & \vdots & \vdots & \ldots & \vdots \\ | & 0 & 0 & \ldots & 0 \end{pmatrix} \\
&= \mathbf{W}_1
\end{aligned}
$$

The matrix $(\mathbf{A}^{\mathrm{T}}\mathbf{v})(\mathbf{A}^{\mathrm{T}}\mathbf{u})^{\mathrm{T}}$ has eigenvector $\mathbf{A}^{\mathrm{T}}\mathbf{v}$ with eigenvalue $(\mathbf{A}^{\mathrm{T}}\mathbf{v})^{\mathrm{T}}(\mathbf{A}^{\mathrm{T}}\mathbf{u}) = 1$. It is useful to mention that if the matrices and vectors were constructed this way, we also have

$$
\begin{aligned}
\mathbf{W}_2 &= \mathbf{W}_1^{\mathrm{T}}\mathbf{A}^{\mathrm{T}}\mathbf{u}\mathbf{x}^{\mathrm{T}} \\
&= \frac{1}{\|\mathbf{A}\mathbf{A}^{\mathrm{T}}\mathbf{v}\|_2^2}
\begin{pmatrix} - & \mathbf{v}^{\mathrm{T}}\mathbf{A} & - \\ 0 & \cdots & 0 \\ \vdots & \cdots & \vdots \\ 0 & \cdots & 0 \end{pmatrix}
\mathbf{A}^{\mathrm{T}}\mathbf{A}\mathbf{A}^{\mathrm{T}}\mathbf{v}\mathbf{x}^{\mathrm{T}} \\
&= \frac{1}{\|\mathbf{A}\mathbf{A}^{\mathrm{T}}\mathbf{v}\|_2^2}
\begin{pmatrix} (\mathbf{v}^{\mathrm{T}}\mathbf{A}\mathbf{A}^{\mathrm{T}})(\mathbf{A}\mathbf{A}^{\mathrm{T}}\mathbf{v}) \\ 0 \\ \vdots \\ 0 \end{pmatrix}
\mathbf{x}^{\mathrm{T}} \\
&= \begin{pmatrix} 1 \\ 0 \\ \vdots \\ 0 \end{pmatrix}\mathbf{x}^{\mathrm{T}} \\
&= \begin{pmatrix} - & \mathbf{x}^{\mathrm{T}} & - \\ 0 & \cdots & 0 \\ \vdots & \cdots & \vdots \\ 0 & \cdots & 0 \end{pmatrix}
\end{aligned}
$$

To finish the proof, note that

$$
\begin{aligned}
\mathbf{W}_2^{\mathrm{T}}\mathbf{W}_1^{\mathrm{T}}\mathbf{A}^{\mathrm{T}}\mathbf{u} &=
\begin{pmatrix} | & 0 & \cdots & 0 \\ \mathbf{x} & \vdots & \cdots & \vdots \\ | & 0 & \cdots & 0 \end{pmatrix}
\begin{pmatrix} - & \mathbf{v}^{\mathrm{T}}\mathbf{A} & - \\ 0 & \cdots & 0 \\ \vdots & \cdots & \vdots \\ 0 & \cdots & 0 \end{pmatrix}
\mathbf{A}^{\mathrm{T}}\frac{\mathbf{A}\mathbf{A}^{\mathrm{T}}\mathbf{v}}{\|\mathbf{A}\mathbf{A}^{\mathrm{T}}\mathbf{v}\|_2^2} \\
&= \begin{pmatrix} | & 0 & \cdots & 0 \\ \mathbf{x} & \vdots & \cdots & \vdots \\ | & 0 & \cdots & 0 \end{pmatrix}
\begin{pmatrix} - & \mathbf{v}^{\mathrm{T}}\mathbf{A}\mathbf{A}^{\mathrm{T}} & - \\ 0 & \cdots & 0 \\ \vdots & \cdots & \vdots \\ 0 & \cdots & 0 \end{pmatrix}
\frac{\mathbf{A}\mathbf{A}^{\mathrm{T}}\mathbf{v}}{\|\mathbf{A}\mathbf{A}^{\mathrm{T}}\mathbf{v}\|_2^2} \\
&= \begin{pmatrix} | & 0 & \cdots & 0 \\ \mathbf{x} & \vdots & \cdots & \vdots \\ | & 0 & \cdots & 0 \end{pmatrix}
\begin{pmatrix} 1 \\ 0 \\ \vdots \\ 0 \end{pmatrix} \\
&= \mathbf{x}.
\end{aligned}
$$

so $\mathbf{x} \in \mathbf{range}\left(\mathbf{W}_2^{\mathrm{T}}\mathbf{W}_1^{\mathrm{T}}\mathbf{A}^{\mathrm{T}}\right)$ $\qquad\square$

We now state and prove two technical lemmas, which are needed later only to prove the much more insightful Theorem 12.

**Lemma 10.** *For $h = 2$, if at any iteration $k$ we have $\mathbf{W}_1^{(k)}$ be all zeros except first column and $\mathbf{W}_2^{(k)}$ be all zeros except first row, then $\mathbf{W}_1^{(k+1)}$ is all zeros except first column and $\mathbf{W}_2^{(k+1)}$ is all zeros except first row.*

*Proof.* Follows immediately from the iteration update steps:

$$
\begin{aligned}
\mathbf{W}_1^{(k+1)} &= \mathbf{W}_1^{(k)} - \alpha\mathbf{A}^{\mathrm{T}}(\mathbf{A}\mathbf{W}_1^{(k)}\mathbf{W}_2^{(k)}\mathbf{x}_k - \mathbf{b})\mathbf{x}_k^{\mathrm{T}}\mathbf{W}_2^{(k)^T} \\
\mathbf{W}_2^{(k+1)} &= \mathbf{W}_2^{(k)} - \alpha\mathbf{W}_1^{(k)^T}\mathbf{A}^{\mathrm{T}}(\mathbf{A}\mathbf{W}_1^{(k)}\mathbf{W}_2^{(k)}\mathbf{x}_k - \mathbf{b})\mathbf{x}_k^{\mathrm{T}}
\end{aligned}
$$

$\qquad\square$

**Lemma 11.** *For $h = 2$, if in any iteration $k$ we have $\mathbf{W}_1^{(k)}$ all zeros except for the first column, and*

$$\mathbf{W}_2^{(k)} = \begin{pmatrix} - & \mathbf{x}_k^T & - \\ 0 & \cdots & 0 \\ \vdots & \cdots & \vdots \\ 0 & \cdots & 0 \end{pmatrix}$$

*Then,*

$$\mathbf{W}_2^{(k+1)} = \begin{pmatrix} - & \mathbf{x}_{k+1}^T & - \\ 0 & \cdots & 0 \\ \vdots & \cdots & \vdots \\ 0 & \cdots & 0 \end{pmatrix}$$

*Proof.* $\mathbf{W}_1^{(k)\,\mathrm{T}}\mathbf{A}^{\mathrm{T}}(\mathbf{A}\mathbf{W}_1^{(k)}\mathbf{W}_2^{(k)}\mathbf{x}_k - \mathbf{b})$ is a column vector of length $d$ with a single non-zero entry in the first index. Denote that non-zero value as $\beta$. Therefore, the first row of the matrix is

$$\mathbf{W}_2^{(k+1)} \;=\; \mathbf{W}_2^{(k)} - \alpha \mathbf{W}_1^{(k)^T}\mathbf{A}^{\mathrm{T}}(\mathbf{A}\mathbf{W}_1^{(k)}\mathbf{W}_2^{(k)}\mathbf{x}_k - \mathbf{b})\mathbf{x}_k^T$$

is $\mathbf{x}_k^{\mathrm{T}} - \alpha\beta\mathbf{x}_k^{\mathrm{T}}$. Similarly, $(\mathbf{A}\mathbf{W}_1^{(k)}\mathbf{W}_2^{(k)}\mathbf{x}_k - \mathbf{b})^{\mathrm{T}}\mathbf{A}\mathbf{W}_1^{(k)}$ is a row vector of length $d$ with a single non-zero entry $\beta$ in the first index. Now the update step for $\mathbf{x}^{\mathrm{T}}$ is

$$\begin{aligned}
\mathbf{x}_{k+1}^{\mathrm{T}} &= \mathbf{x}_k^{\mathrm{T}} - \alpha(\mathbf{A}\mathbf{W}_1^{(k)}\mathbf{W}_2^{(k)}\mathbf{x}_k - b)^{\mathrm{T}}\mathbf{A}\mathbf{W}_1^{(k)}\mathbf{W}_2^{(k)} \\
&= \mathbf{x}_k^{\mathrm{T}} - \alpha \begin{pmatrix} \beta & 0 & \cdots & 0 \end{pmatrix} \begin{pmatrix} - & \mathbf{x}_k^T & - \\ 0 & \cdots & 0 \\ \vdots & \cdots & \vdots \\ 0 & \cdots & 0 \end{pmatrix} \\
&= \mathbf{x}_k^{\mathrm{T}} - \alpha\beta\mathbf{x}_k^{\mathrm{T}}
\end{aligned}$$

Both terms have the same update step, hence they are equal. $\qquad\square$

The next theorem is a key result, which builds on the previous lemmas, and shows that gradient descent conserves during training the very special form of the weights described in Lemma 9. We then use this theorem to prove the bi-optimality equivalent of this model in Corollary 14.

**Theorem 12.** *For $h = 2$, suppose that $\mathbf{W}_1^{(0)}$ and $\mathbf{W}_2^{(0)}$ and $\mathbf{x}_0$ were constructed as described in Lemma 9 and assume that $\mathbf{x}_k$ is never zero. Then, for all $k$ there exist $\mathbf{v}_k \in \mathbb{R}^{n \times 1}$ and $\mathbf{u}_k \in \mathbb{R}^{n \times 1}$ such that $\mathbf{W}_1^{(k)} = \mathbf{A}^T\mathbf{v}_k\mathbf{x}_k^T\mathbf{W}_2^{(k)^T}$ and $\mathbf{W}_2^{(k)} = \mathbf{W}_1^{(k)^T}\mathbf{A}^T\mathbf{u}_k\mathbf{x}_k^T$ and $\mathbf{x}_k \in \mathbf{range}\left(\mathbf{W}_2^{(k)^T}\mathbf{W}_1^{(k)^T}\mathbf{A}^T\right)$*

*Proof.* The proof is inductive, just as in the $h = 1$ case. The basis of our induction is given by Lemma 9. Now suppose that the hypothesis is true up to $k$. By Lemma 10, we know that $\mathbf{W}_1^{(k+1)}$ are all zeros except the first column $\mathbf{w}_1^{(k+1)}$, $\mathbf{W}_2^{(k+1)}$ is all zeros except first row, which is equal to $\mathbf{x}_{k+1}^{\mathrm{T}}$ by Lemma 11. Define

$$\mathbf{v}_{k+1} \;:=\; \frac{1}{\|\mathbf{x}_{k+1}\|_2^4}\mathbf{A}^{\mathrm{T}^+}\mathbf{W}_1^{(k+1)}\mathbf{W}_2^{(k+1)}\mathbf{x}_{k+1}$$

and verify that

$$
\begin{aligned}
\mathbf{A}^{\mathrm{T}} \mathbf{v}_{k+1} \mathbf{x}_{k+1}^{\mathrm{T}} \mathbf{W}_2^{(k+1)^{\mathrm{T}}} &= \frac{1}{\|\mathbf{x}_{k+1}\|_2^4} \mathbf{A}^{\mathrm{T}} \mathbf{A}^{\mathrm{T}^+} \mathbf{W}_1^{(k+1)} \mathbf{W}_2^{(k+1)} \mathbf{x}_{k+1} \mathbf{x}_{k+1}^{\mathrm{T}} \mathbf{W}_2^{(k+1)^{\mathrm{T}}} \\
&= \frac{1}{\|\mathbf{x}_{k+1}\|_2^4} \mathbf{A}^{\mathrm{T}} \mathbf{A}^{\mathrm{T}^+} \mathbf{W}_1^{(k+1)} \begin{pmatrix} \|\mathbf{x}_{k+1}\|_2^2 \\ 0 \\ \vdots \\ 0 \end{pmatrix} \begin{pmatrix} \|\mathbf{x}_{k+1}\|_2^2 & 0 & \dots & 0 \end{pmatrix} \\
&= \mathbf{A}^{\mathrm{T}} \mathbf{A}^{\mathrm{T}^+} \begin{pmatrix} | & 0 & \dots & 0 \\ \mathbf{w}_1^{(k+1)} & \vdots & \dots & 0 \\ | & 0 & \dots & 0 \end{pmatrix} \begin{pmatrix} 1 & 0 & \dots & 0 \\ 0 & 0 & \dots & 0 \\ \vdots & \vdots & \dots & 0 \\ 0 & 0 & \dots & 0 \end{pmatrix} \\
&= \mathbf{A}^{\mathrm{T}} \mathbf{A}^{\mathrm{T}^+} \begin{pmatrix} | & 0 & \dots & 0 \\ \mathbf{w}_1^{(k+1)} & \vdots & \dots & 0 \\ | & 0 & \dots & 0 \end{pmatrix} \\
&= \mathbf{A}^{\mathrm{T}} \mathbf{A}^{\mathrm{T}^+} \mathbf{W}_1^{(k+1)} \\
&= \mathbf{A}^{\mathrm{T}} \mathbf{A}^{\mathrm{T}^+} (\mathbf{W}_1^{(k)} - \alpha \mathbf{A}^{\mathrm{T}} (\mathbf{A} \mathbf{W}_1^{(k)} \mathbf{W}_2^{(k)} \mathbf{x}_k - \mathbf{b}) \mathbf{x}_k^{\mathrm{T}} \mathbf{W}_2^{(k)^{\mathrm{T}}}) \\
&= \mathbf{A}^{\mathrm{T}} \mathbf{A}^{\mathrm{T}^+} (\mathbf{A}^{\mathrm{T}} \mathbf{v}_k \mathbf{x}_k^{\mathrm{T}} \mathbf{W}_2^{(k)^{\mathrm{T}}} - \alpha \mathbf{A}^{\mathrm{T}} (\mathbf{A} \mathbf{W}_1^{(k)} \mathbf{W}_2^{(k)} \mathbf{x}_k - \mathbf{b}) \mathbf{x}_k^{\mathrm{T}} \mathbf{W}_2^{(k)^{\mathrm{T}}}) \\
&= \mathbf{A}^{\mathrm{T}} \mathbf{v}_k \mathbf{x}_k^{\mathrm{T}} \mathbf{W}_2^{(k)^{\mathrm{T}}} - \alpha \mathbf{A}^{\mathrm{T}} (\mathbf{A} \mathbf{W}_1^{(k)} \mathbf{W}_2^{(k)} \mathbf{x}_k - \mathbf{b}) \mathbf{x}_k^{\mathrm{T}} \mathbf{W}_2^{(k)^{\mathrm{T}}} \\
&= \mathbf{W}_1^{(k)} - \alpha \mathbf{A}^{\mathrm{T}} (\mathbf{A} \mathbf{W}_1^{(k)} \mathbf{W}_2^{(k)} \mathbf{x}_k - \mathbf{b}) \mathbf{x}_k^{\mathrm{T}} \mathbf{W}_2^{(k)^{\mathrm{T}}} \\
&= \mathbf{W}_1^{(k+1)}
\end{aligned}
$$

Recall that $\mathbf{A}$ has full rank and less rows than columns, so we used the fact that $\mathbf{A}^{\mathrm{T}^+} \mathbf{A}^{\mathrm{T}} = \mathbf{I}_n$. As for $\mathbf{W}_2^{(k+1)}$, the proof is similar. Denote:

$$
\mathbf{u}_{k+1} = \frac{1}{\|\mathbf{x}_{k+1}\|_2^2 \cdot \|\mathbf{A} \mathbf{W}_1^{(k+1)}\|_F^2} \mathbf{A} \mathbf{W}_1^{(k+1)} \mathbf{W}_2^{(k+1)} \mathbf{x}_{k+1}
$$

remember that $\mathbf{x}_k$ is never zero and notice that $\mathbf{A} \mathbf{W}_1^{(k+1)} = (\mathbf{A} \mathbf{A}^{\mathrm{T}})(\mathbf{v}_{k+1} \mathbf{x}_{k+1}^{\mathrm{T}} \mathbf{W}_2^{(k+1)^{\mathrm{T}}})$ can't be the zero matrix because $\mathbf{A} \mathbf{A}^{\mathrm{T}}$ is full rank and only has the trivial solution. Now following a similar logic as before:

$$
\begin{aligned}
\mathbf{W}_1^{(k+1)^{\mathrm{T}}} \mathbf{A}^{\mathrm{T}} \mathbf{u}_{k+1} \mathbf{x}_{k+1}^{\mathrm{T}} &= \frac{1}{\|\mathbf{x}_{k+1}\|_2^2 \cdot \|\mathbf{A} \mathbf{W}_1^{(k+1)}\|_F^2} \mathbf{W}_1^{(k+1)^{\mathrm{T}}} \mathbf{A}^{\mathrm{T}} \mathbf{A} \mathbf{W}_1^{(k+1)} \mathbf{W}_2^{(k+1)} \mathbf{x}_{k+1} \mathbf{x}_{k+1}^{\mathrm{T}} \\
&= \frac{1}{\|\mathbf{x}_{k+1}\|_2^2} \begin{pmatrix} 1 & 0 & \dots & 0 \\ 0 & 0 & \dots & 0 \\ \vdots & \vdots & \dots & \vdots \\ 0 & 0 & \dots & 0 \end{pmatrix} \mathbf{W}_2^{(k+1)} \mathbf{x}_{k+1} \mathbf{x}_{k+1}^{\mathrm{T}} \\
&= \frac{1}{\|\mathbf{x}_{k+1}\|_2^2} (\mathbf{W}_2^{(k+1)} \mathbf{x}_{k+1}) \mathbf{x}_{k+1}^{\mathrm{T}} \\
&= \begin{pmatrix} 1 \\ 0 \\ \vdots \\ 0 \end{pmatrix} \mathbf{x}_{k+1}^{\mathrm{T}} \\
&= \mathbf{W}_2^{(k+1)}
\end{aligned}
$$

This concludes the proofs for $\mathbf{W}_1$ and $\mathbf{W}_2$. The claim of $\mathbf{x}_{k+1} \in \mathbf{range}\left(\mathbf{W}_2^{(k+1)^{\mathrm{T}}} \mathbf{W}_1^{(k+1)^{\mathrm{T}}} \mathbf{A}^{\mathrm{T}}\right)$ follows the same steps as in the proof of Lemma 9. $\qquad \square$

Notice that we assumed in the previous theorem that $\mathbf{x}_k \neq 0$, and one reason for that assumption is that if $\mathbf{x}_k = 0$ then by Lemma 11 we have $\mathbf{W}_2^{(k)} = 0$ as well, which leads to a saddle point (all gradients are zero) and the iteration stops.

The following lemma extends Theorem 12 when $k \to \infty$.

**Lemma 13.** *For $h = 2$, consider the sequences $\{\mathbf{W}_1^{(0)}, \mathbf{W}_1^{(1)}, \dots\}, \{\mathbf{W}_2^{(0)}, \mathbf{W}_2^{(1)}, \dots\}, \{\mathbf{x}_0, \mathbf{x}_1, \dots\}$. If the conditions of Theorem 12 are met and the sequences converge to $\mathbf{W}_1^{(\infty)}, \mathbf{W}_2^{(\infty)}, \mathbf{x}_\infty \neq 0$ respectively, then the sequences $\{\mathbf{v}_1, \mathbf{v}_2, \dots\}, \{\mathbf{u}_1, \mathbf{u}_2, \dots\}$ defined by the result of Theorem 12 converge to $\mathbf{v}_\infty$ and $\mathbf{u}_\infty$ respectively, and*

$$\begin{aligned} \mathbf{W}_1^{(\infty)} &= \mathbf{A}^T \mathbf{v}_\infty \mathbf{x}_\infty^T \mathbf{W}_2^{(\infty)T} \\ \mathbf{W}_2^{(\infty)} &= \mathbf{W}_1^{(\infty)T} \mathbf{A}^T \mathbf{u}_\infty \mathbf{x}_\infty^T \end{aligned}$$

*and $\mathbf{x}_\infty \in \mathbf{range}\left( \mathbf{W}_2^{(\infty)T} \mathbf{W}_1^{(\infty)T} \mathbf{A}^T \right)$*

*Proof.* Going by the definitions outlined in Theorem 12 we have

$$\mathbf{v}_\infty = \lim_{k \to \infty} \mathbf{v}_k = \lim_{k \to \infty} \frac{1}{\|\mathbf{x}_k\|_2^4} \mathbf{A}^{T^+} \mathbf{W}_1^{(k)} \mathbf{W}_2^{(k)} \mathbf{x}_k = \frac{1}{\|\mathbf{x}_\infty\|_2^4} \mathbf{A}^{T^+} \mathbf{W}_1^{(\infty)} \mathbf{W}_2^{(\infty)} \mathbf{x}_\infty$$

and a similar logic for

$$\mathbf{u}_\infty = \lim_{k \to \infty} \mathbf{u}_k = \lim_{k \to \infty} \frac{1}{\|\mathbf{x}_k\|_2^2 \cdot \|\mathbf{A}\mathbf{W}_1^{(k)}\|_F^2} \mathbf{A}\mathbf{W}_1^{(k)} \mathbf{W}_2^{(k)} \mathbf{x}_k = \frac{1}{\|\mathbf{x}_\infty\|_2^2 \cdot \|\mathbf{A}\mathbf{W}_1^{(\infty)}\|_F^2} \mathbf{A}\mathbf{W}_1^{(\infty)} \mathbf{W}_2^{(\infty)} \mathbf{x}_\infty$$

Now we can simply multiply and see that

$$\begin{aligned} \mathbf{W}_1^{(\infty)} &= \mathbf{A}^T \mathbf{v}_\infty \mathbf{x}_\infty^T \mathbf{W}_2^{(\infty)T} \\ \mathbf{W}_2^{(\infty)} &= \mathbf{W}_1^{(\infty)T} \mathbf{A}^T \mathbf{u}_\infty \mathbf{x}_\infty^T \end{aligned}$$

as required. The proof for $\mathbf{x}_\infty$ follows the same steps as Lemma 9. $\qquad\square$

As expected, this initialization admits properties similar to those outlined in Corollary 7. The next corollary describes the results of initializing in this special way and is the goal we built towards in this section.

**Corollary 14.** *Denote $\mathbf{W}_1^{(\infty)} := \lim_{k \to \infty} \mathbf{W}_1^{(k)}, \mathbf{W}_2^{(\infty)} := \lim_{k \to \infty} \mathbf{W}_2^{(k)}, \mathbf{x}_\infty := \lim_{k \to \infty} \mathbf{x}_k$. If the conditions of Theorem 12 hold, the limits exist and are non-zero, and finally $\mathbf{A}\mathbf{W}_1^{(\infty)} \mathbf{W}_2^{(\infty)} \mathbf{x}_\infty = \mathbf{b}$, then the following statements are true:*

1. *$\mathbf{W}_1^{(\infty)} \mathbf{W}_2^{(\infty)} \mathbf{x}_\infty$ is the minimum norm solution to the problem $\mathbf{A}\mathbf{z} = \mathbf{b}$*

2. *$\mathbf{x}_\infty$ is the minimum norm solution to the problem $(\mathbf{A}\mathbf{W}_1^{(\infty)} \mathbf{W}_2^{(\infty)})\mathbf{z} = \mathbf{b}$*

3. *$vec(\mathbf{W}_1^{(\infty)})$ is the minimum norm solution to the problem $(\mathbf{x}_\infty^T \mathbf{W}_2^{(\infty)T} \otimes \mathbf{A})\mathbf{z} = \mathbf{b}$*

4. *$vec(\mathbf{W}_2^{(\infty)})$ is the minimum norm solution to the problem $(\mathbf{x}_\infty^T \otimes \mathbf{A}\mathbf{W}_1^{(\infty)})\mathbf{z} = \mathbf{b}$*

*Proof.* Use Lemma 13 and follow the same logic as Corollary 7 $\qquad\square$

Another similarity to the $h = 1$ linear model is that this can be collapsed to a more compact algorithm. We do not need to iterate over $\mathbf{W}_1$ and $\mathbf{W}_2$. Suppose that we know $\mathbf{x}_k$ and $\mathbf{v}_k$ at some iteration $k$. Then we can construct

$$\mathbf{W}_2^{(k)} = \begin{pmatrix} - & \mathbf{x}_k^T & - \\ 0 & \dots & 0 \\ \vdots & \dots & \vdots \\ 0 & \dots & 0 \end{pmatrix}$$

trivially as we have shown from Lemma 11. This now allows us to compute $\mathbf{W}_1^{(k)} = \mathbf{A}^{\mathrm{T}} \mathbf{v}_k \mathbf{x}_k^{\mathrm{T}} \mathbf{W}_2^{(k)^{\mathrm{T}}}$.

Thus, if we wanted to stop at this iteration and produce a result, knowing $\mathbf{x}_k$ and $\mathbf{v}_k$ is all the information we need. It is also all we need for the iteration step. We can write $\mathbf{x}_{k+1}$ as follows:

$$
\begin{aligned}
\mathbf{x}_{k+1} &= \mathbf{x}_k - \alpha \mathbf{W}_2^{(k)^{\mathrm{T}}} \mathbf{W}_1^{(k)^{\mathrm{T}}} \mathbf{A}^{\mathrm{T}} (\mathbf{A} \mathbf{W}_1^{(k)} \mathbf{W}_2^{(k)} \mathbf{x}_k - \mathbf{b}) \\
&= \mathbf{x}_k - \alpha \mathbf{W}_2^{(k)^{\mathrm{T}}} \mathbf{W}_2^{(k)} \mathbf{x}_k \mathbf{v}_k^{\mathrm{T}} \mathbf{A} \mathbf{A}^{\mathrm{T}} (\mathbf{A} \mathbf{A}^{\mathrm{T}} \mathbf{v}_k \mathbf{x}_k^{\mathrm{T}} \mathbf{W}_2^{(k)^{\mathrm{T}}} \mathbf{W}_2^{(k)} \mathbf{x}_k - \mathbf{b}) \\
&= \mathbf{x}_k - \alpha \begin{pmatrix} | & 0 & \cdots & 0 \\ \mathbf{x}_k & \vdots & \cdots & \vdots \\ | & 0 & \cdots & 0 \end{pmatrix} \begin{pmatrix} \|\mathbf{x}_k\|_2^2 \\ 0 \\ \vdots \\ 0 \end{pmatrix} \mathbf{v}_k^{\mathrm{T}} \mathbf{A} \mathbf{A}^{\mathrm{T}} (\|\mathbf{x}_k\|_2^4 \mathbf{A} \mathbf{A}^{\mathrm{T}} \mathbf{v}_k - \mathbf{b}) \\
&= (1 - \alpha \|\mathbf{x}_k\|_2^2 \mathbf{v}_k^{\mathrm{T}} \mathbf{A} \mathbf{A}^{\mathrm{T}} (\|\mathbf{x}_k\|_2^4 \mathbf{A} \mathbf{A}^{\mathrm{T}} \mathbf{v}_k - \mathbf{b})) \mathbf{x}_k
\end{aligned}
$$

and by using the iteration step for $\mathbf{v}_{k+1}$ written in the proof of Theorem 12, we can write:

$$
\begin{aligned}
\mathbf{v}_{k+1} &= \frac{1}{\|\mathbf{x}_{k+1}\|_2^4} \mathbf{A}^{\mathrm{T}^+} \mathbf{W}_1^{(k+1)} \mathbf{W}_2^{(k+1)} \mathbf{x}_{k+1} \\
&= \frac{1}{\|\mathbf{x}_{k+1}\|_2^4} \mathbf{A}^{\mathrm{T}^+} (\mathbf{W}_1^{(k)} - \alpha \mathbf{A}^{\mathrm{T}} (\mathbf{A} \mathbf{W}_1^{(k)} \mathbf{W}_2^{(k)} \mathbf{x}_k - \mathbf{b}) \mathbf{x}_k^{\mathrm{T}} \mathbf{W}_2^{(k)^{\mathrm{T}}}) \mathbf{W}_2^{(k+1)} \mathbf{x}_{k+1} \\
&= \frac{1}{\|\mathbf{x}_{k+1}\|_2^4} \mathbf{A}^{\mathrm{T}^+} (\mathbf{A}^{\mathrm{T}} \mathbf{v}_k \begin{pmatrix} \|\mathbf{x}_k\|_2^2 & 0 & \cdots & 0 \end{pmatrix} \\
&\qquad - \alpha \mathbf{A}^{\mathrm{T}} (\|\mathbf{x}_k\|_2^4 \mathbf{A} \mathbf{A}^{\mathrm{T}} \mathbf{v}_k - \mathbf{b}) \begin{pmatrix} \|\mathbf{x}_k\|_2^2 & 0 & \cdots & 0 \end{pmatrix}) \begin{pmatrix} \|\mathbf{x}_{k+1}\|_2^2 \\ 0 \\ \vdots \\ 0 \end{pmatrix} \\
&= \frac{1}{\|\mathbf{x}_{k+1}\|_2^2} (\mathbf{v}_k - \alpha(\|\mathbf{x}_k\|_2^4 \mathbf{A} \mathbf{A}^{\mathrm{T}} \mathbf{v}_k - \mathbf{b})) \begin{pmatrix} \|\mathbf{x}_k\|_2^2 & 0 & \cdots & 0 \end{pmatrix} \begin{pmatrix} 1 \\ 0 \\ \vdots \\ 0 \end{pmatrix} \\
&= \frac{\|\mathbf{x}_k\|_2^2}{\|\mathbf{x}_{k+1}\|_2^2} (\mathbf{v}_k - \alpha(\|\mathbf{x}_k\|_2^4 \mathbf{A} \mathbf{A}^{\mathrm{T}} \mathbf{v}_k - \mathbf{b}))
\end{aligned}
$$

So even for the iteration we just need $\mathbf{x}_k$ and $\mathbf{v}_k$, and can iterate over them only, reducing the number of parameters from $2d^2 + d$ to $d + n$, but just as before we can do better.

Notice that the iteration step for $\mathbf{x}_k$ again looks like

$$
\begin{aligned}
\mathbf{x}_{k+1} &= (1 - \gamma_k) \mathbf{x}_k \\
&= \prod_{i=0}^{k} (1 - \gamma_i) x_0 \\
&= \rho_{k+1} x_0
\end{aligned}
$$

where $\gamma_k = \alpha \|\mathbf{x}_k\|_2^2 \mathbf{v}_k^{\mathrm{T}} \mathbf{A} \mathbf{A}^{\mathrm{T}} (\|\mathbf{x}_k\|_2^4 \mathbf{A} \mathbf{A}^{\mathrm{T}} \mathbf{v}_k - \mathbf{b})$ and $\rho_k = \prod_{i=0}^{k-1} (1 - \gamma_i)$. We can use that $\|\mathbf{x}_0\|_2 = 1$ to rewrite $\gamma_k$ as

$$
\gamma_k = \alpha \rho_k^2 \mathbf{v}_k^{\mathrm{T}} \mathbf{A} \mathbf{A}^{\mathrm{T}} (\rho_k^4 \mathbf{A} \mathbf{A}^{\mathrm{T}} \mathbf{v}_k - \mathbf{b})
$$

We can use $\gamma_k$ and $\rho_k$ to get a succinct and simple update step for $\mathbf{v}_k$:

$$
\begin{aligned}
\mathbf{v}_{k+1} &= \frac{\|\mathbf{x}_k\|_2^2}{\|x_{k+1}\|_2^2} (\mathbf{v}_k - \alpha(\|\mathbf{x}_k\|_2^4 \mathbf{A} \mathbf{A}^{\mathrm{T}} \mathbf{v}_k - \mathbf{b})) \\
&= \frac{\rho_k^2}{\rho_{k+1}^2} (\mathbf{v}_k - \alpha(\rho_k^4 \mathbf{A} \mathbf{A}^{\mathrm{T}} \mathbf{v}_k - \mathbf{b})) \\
&= \frac{1}{(1 - \gamma_k)^2} (\mathbf{v}_k - \alpha(\rho_k^4 \mathbf{A} \mathbf{A}^{\mathrm{T}} \mathbf{v}_k - \mathbf{b}))
\end{aligned}
$$

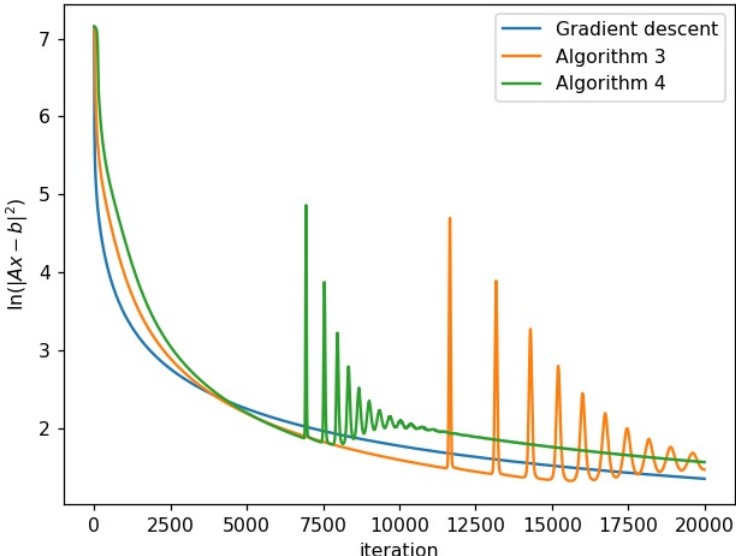

Figure 4.1: On rcv1 multiclass test set, each method with its largest learning rate in exponents of 10 (Algorithm 3 with $\alpha = 10^{-2}$, Algorithm 4 with $\alpha = 10^{-3}$, gradient descent with $\alpha = 10$)

This allows us to effectively collapse a two hidden layers linear network to $O(n)$ variables, much like we did in the one hidden layer model. The algorithm is outlined below (Algorithm 4).

---

**Algorithm 4** Compact two hidden layers iteration

$\mathbf{A} \in \mathbb{R}^{n \times d}, \mathbf{b} \in \mathbb{R}^{n \times 1}, \alpha \in \mathbb{R}$ inputs
$\mathbf{v}_0 \leftarrow$ arbitrary, not zero $\qquad\qquad O(n)$
$\rho_0 \leftarrow 1 \qquad\qquad\qquad\qquad\qquad O(1)$
$\mathbf{z}_0 \leftarrow \mathbf{A}^{\mathrm{T}} \mathbf{v}_0 \qquad\qquad\qquad\qquad O(T_{\mathbf{A}})$
**for** iteration $k = 0, 1, \dots$ until convergence **do**
$\qquad \mathbf{y}_k \leftarrow \mathbf{A} \mathbf{z}_k \qquad\qquad\qquad\qquad O(T_{\mathbf{A}})$
$\qquad \mathbf{e}_k \leftarrow \alpha(\rho_k^4 \mathbf{y}_k - \mathbf{b}) \qquad\qquad\quad O(n)$
$\qquad \gamma_k \leftarrow \rho_k^2 (\mathbf{y}_k^{\mathrm{T}} \mathbf{e}_k) \qquad\qquad\qquad O(n)$
$\qquad \mathbf{v}_{k+1} \leftarrow \frac{1}{(1-\gamma_k)^2}(\mathbf{v}_k - \mathbf{e}_k) \qquad O(n)$
$\qquad \rho_{k+1} \leftarrow \rho_k(1 - \gamma_k) \qquad\qquad\quad O(1)$
$\qquad \mathbf{z}_{k+1} \leftarrow \mathbf{A}^{\mathrm{T}} \mathbf{v}_{k+1} \qquad\qquad\quad O(T_{\mathbf{A}})$
**end for**
output $\rho_k^4 \mathbf{z}_k$

---

The time complexity of running Algorithm 4 for $t > 1$ iterations is $O(t \cdot \max(n, T_{\mathbf{A}}))$. The similarities between Algorithm 3 and Algorithm 4 are striking, but not entirely surprising.

We tested both these algorithms against the baseline gradient descent algorithm to answer two questions. Can these two new algorithms outperform gradient descent and take different paths to $\boldsymbol{\theta}^{\star}$? To answer the first question, we used the rcv1 multiclass test set, removed zero columns, and divided the feature matrix $\mathbf{A}$ and the target vector $\mathbf{b}$ by 52. We then trained three models using the methods mentioned above, and the results in Figure 4.1 show that the new methods we propose are competitive and even beat gradient descent, but begin to zigzag wildly after a certain amount of iterations.

Looking further into the matter, we see that Algorithm 3 begins to zigzag as soon as $\frac{1}{1-\gamma_k} > 1$ and Algorithm 4 begins to zigzag as soon as $\frac{1}{(1-\gamma_k)^2} > 1$, which both zigzag back and forth between a bit more than 1 and a bit less than 1. An illustration of this is shown in Figure 4.2.

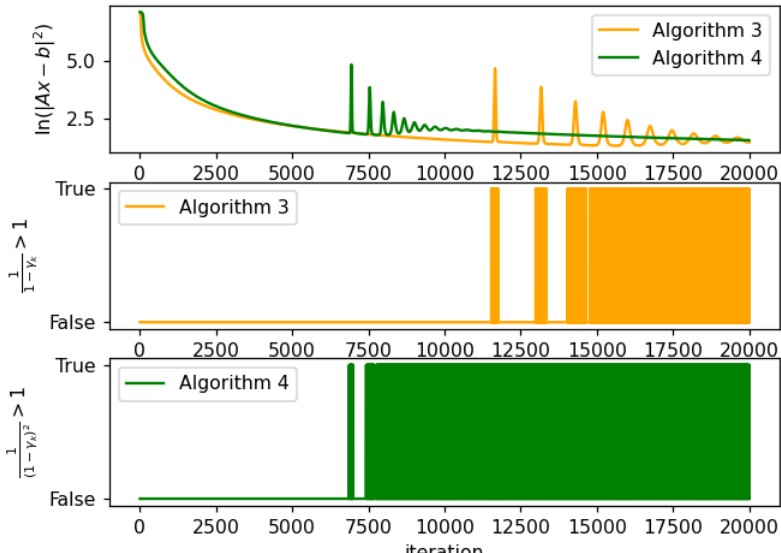

Figure 4.2: The new algorithms begin to zigzag when the corresponding coefficients zigzag between a bit more and a bit less than 1.

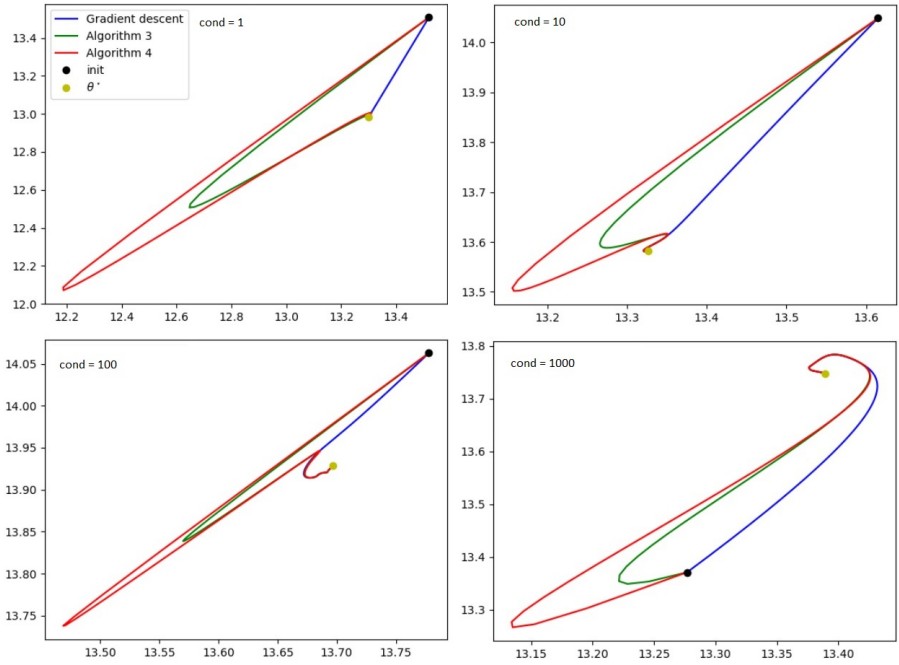

Figure 4.3: Random projection of path. $n = 100, d = 1000,$ condition number varies

As for the second question, the answer is a definite "No", as can be clearly seen in figure 4.3 where we solved random 100 by 1000 problems with specified condition numbers, and then projected the iteration path unto a 2d plane with a random projection to see if the two methods take the same path. They don't take the same path, Algorithm 4 seems to take a longer path, but it steps through that path more quickly as can be seen empirically by the constraint on $\alpha$ in the experiments on the rcv1 dataset.

Is it possible to "collapse" deep models for $h > 2$? We conjecture that no. We do not have a formal proof but a heuristic argument. Consider, for example, the model when $h = 3$, $\mathbf{A}\mathbf{W}_1\mathbf{W}_2\mathbf{W}_3\mathbf{x} = \mathbf{b}$. As before, we

would like $\mathbf{W}_1 = \mathbf{A}^{\mathrm{T}}\mathbf{v}\mathbf{x}^{\mathrm{T}}\mathbf{W}_3^{\mathrm{T}}\mathbf{W}_2^{\mathrm{T}}, \mathbf{W}_2 = \mathbf{W}_1^{\mathrm{T}}\mathbf{A}^{\mathrm{T}}\mathbf{u}\mathbf{x}^{\mathrm{T}}\mathbf{W}_3^{\mathrm{T}}, \mathbf{W}_3 = \mathbf{W}_2^{\mathrm{T}}\mathbf{W}_1^{\mathrm{T}}\mathbf{A}^{\mathrm{T}}\mathbf{s}\mathbf{x}^{\mathrm{T}}$ for some $u, v, s$. Using the same strategy, we would have $\mathbf{W}_1 = (\mathbf{A}^{\mathrm{T}}\mathbf{v}\mathbf{x}^{\mathrm{T}}\mathbf{W}_3^{\mathrm{T}})(\mathbf{A}^{\mathrm{T}}\mathbf{u}\mathbf{x}^{\mathrm{T}}\mathbf{W}_3^{\mathrm{T}})^{\mathrm{T}}\mathbf{W}_1$ and $\mathbf{W}_2 = (\mathbf{W}_1^{\mathrm{T}}\mathbf{A}^{\mathrm{T}}\mathbf{u})(\mathbf{W}_1^{\mathrm{T}}\mathbf{A}^{\mathrm{T}}\mathbf{s})^{\mathrm{T}}\mathbf{W}_2$ which means we need to choose $\mathbf{u}, \mathbf{v}, \mathbf{s}$ such that $(\mathbf{A}^{\mathrm{T}}\mathbf{v}\mathbf{x}^{\mathrm{T}}\mathbf{W}_3^{\mathrm{T}})^{\mathrm{T}}(\mathbf{A}^{\mathrm{T}}\mathbf{u}\mathbf{x}^{\mathrm{T}}\mathbf{W}_3^{\mathrm{T}}) = (\mathbf{W}_1^{\mathrm{T}}\mathbf{A}^{\mathrm{T}}\mathbf{u})^{\mathrm{T}}(\mathbf{W}_1^{\mathrm{T}}\mathbf{A}^{\mathrm{T}}\mathbf{s}) = 1$. This does not seem feasible for weight matrices that are strongly coupled like that. It only worked in the case $h = 2$ because the term for $\mathbf{W}_1$ was without any mention of $\mathbf{W}_2$, but here there is seemingly no way to decouple the weight matrices from each other. The key to solving the problem in the $h = 2$ case does not work in the $h > 2$ case, and there is no clear way of overcoming this problem. We do not claim the statement is true, we leave it as an open problem. We only claim that the previous strategy does not work.

A final question is how our initialization methods compare to industry standard popular initializations. Unfortunately, there is little relation, as our initializations, while random, are supported on a set of zero measure. In contrast, most popular methods today sample scalar entries individually, and the support has a positive measure (possibly even the entire space). Two prominent examples of industry standard initialization are Xavier (Glorot & Bengio, 2010) and He Initializations (He et al., 2015).

In a Xavier Initialization we generate all entries from a uniform distribution on $-\frac{1}{\sqrt{s}}$ and $\frac{1}{\sqrt{s}}$, where $s$ is the number of neurons in the previous layer. The goal of this initialization, which is widely used for the activation functions $\frac{1}{1+e^{-x}}$ and $\tanh(x)$, is to have constant variance across all layers. This prevents the gradients from vanishing or exploding. He Initialization was invented to solve the problem that Xavier does not work well when the activation function is ReLU. When performing He initialization, we generate numbers from a normal distribution with mean 0 and variance $\frac{2}{s}$.

Both of these initializations, and indeed most initialization techniques today, sample entries individually, and so they miss the big picture of possible dependency on the data given and how to use it. They are designed with optimization in mind, rather than generalization, and are very different from the methods we propose. Initializing with these methods will almost surely not yield $\boldsymbol{\theta}^{\star}$ and will not take advantage of the collapsing property we have outlined. It is possible, however, that these initialization schemes avoid possible exploding/vanishing gradient phenomena better than our proposed methods.

### 4.2   Stability analysis of deep linear networks

We have shown in Lemma 8 that if $\mathbf{W}_1^{(0)} \in \mathbf{range}\left(\mathbf{A}^{\mathrm{T}}\right)$ and $\mathbf{A}\mathbf{W}_1^{(\infty)}\dots\mathbf{W}_h^{(\infty)}\mathbf{x}_\infty = \mathbf{b}$ then the limit $\mathbf{W}_1^{(\infty)}\dots\mathbf{W}_h^{(\infty)}\mathbf{x}_\infty$ exists and equals $\boldsymbol{\theta}^{\star}$. However, it is not always easy to achieve this perfectly, and due to machine precision or other reasons we might have $\mathbf{W}_1^{(0)} \notin \mathbf{range}\left(\mathbf{A}^{\mathrm{T}}\right)$. Thus, a natural question to ask is what would happen if $\mathbf{W}_1^{(0)} \notin \mathbf{range}\left(\mathbf{A}^{\mathrm{T}}\right)$, but is close to $\mathbf{range}\left(\mathbf{A}^{\mathrm{T}}\right)$ in some sense. We formalize this question by first writing

$$\mathbf{W}_1^{(k)} = \mathbf{A}^{\mathrm{T}}\mathbf{P}_k + \mathbf{C}_k$$

where

$$\mathbf{P}_k = (\mathbf{A}^{\mathrm{T}})^{+}\mathbf{W}_1^{(k)}$$
$$\mathbf{C}_k = \mathbf{W}_1^{(k)} - \mathbf{A}^{\mathrm{T}}\mathbf{P}_k.$$

and we assume $\mathbf{C}_0 \neq 0$. First, notice that $\mathbf{A}\mathbf{C}_k = 0$ is retained throughout our iterations. This is because

$$\mathbf{A}\mathbf{C}_k = \mathbf{A}\mathbf{W}_1^{(k)} - \mathbf{A}\mathbf{A}^{\mathrm{T}}(\mathbf{A}\mathbf{A}^{\mathrm{T}})^{-1}\mathbf{A}\mathbf{W}_1^{(k)}$$
$$= \mathbf{A}\mathbf{W}_1^{(k)} - \mathbf{A}\mathbf{W}_1^{(k)}$$
$$= 0$$

We can use this to arrive at the conclusion that $\mathbf{C}_k$ never changes, as

$$
\begin{aligned}
\mathbf{C}_{k+1} &= \mathbf{W}_1^{(k+1)} - \mathbf{A}^{\mathrm{T}}\mathbf{P}_{k+1} \\
&= \mathbf{W}_1^{(k+1)} - \mathbf{A}^{\mathrm{T}}\mathbf{A}^{\mathrm{T}^+}\mathbf{W}_1^{(k+1)} \\
&= \mathbf{W}_1^{(k)} - \mathbf{A}^{\mathrm{T}}(\mathbf{A}\mathbf{W}_1^{(k)}\mathbf{x}_k - \mathbf{b})\mathbf{x}_k^{\mathrm{T}} - \mathbf{A}^{\mathrm{T}}\mathbf{A}^{\mathrm{T}^+}(\mathbf{W}_1^{(k)} - \mathbf{A}^{\mathrm{T}}(\mathbf{A}\mathbf{W}_1^{(k)}\mathbf{x}_k - \mathbf{b})\mathbf{x}_k^{\mathrm{T}}) \\
&= \mathbf{W}_1^{(k)} - \mathbf{A}^{\mathrm{T}}(\mathbf{A}\mathbf{W}_1^{(k)}\mathbf{x}_k - \mathbf{b})\mathbf{x}_k^{\mathrm{T}} - \mathbf{A}^{\mathrm{T}}\mathbf{A}^{\mathrm{T}^+}\mathbf{W}_1^{(k)} + \mathbf{A}^{\mathrm{T}}(\mathbf{A}\mathbf{W}_1^{(k)}\mathbf{x}_k - \mathbf{b})\mathbf{x}_k^{\mathrm{T}} \\
&= \mathbf{W}_1^{(k)} - \mathbf{A}^{\mathrm{T}}\mathbf{A}^{\mathrm{T}^+}\mathbf{W}_1^{(k)} \\
&= \mathbf{A}^{\mathrm{T}}\mathbf{P}_k + \mathbf{C}_k - \mathbf{A}^{\mathrm{T}}\mathbf{A}^{\mathrm{T}^+}(\mathbf{A}^{\mathrm{T}}\mathbf{P}_k + \mathbf{C}_k) \\
&= \mathbf{C}_k - \mathbf{A}^{\mathrm{T}}\mathbf{A}^{\mathrm{T}^+}\mathbf{C}_k \\
&= \mathbf{C}_k - \mathbf{A}^{\mathrm{T}}(\mathbf{A}\mathbf{A}^{\mathrm{T}})^{-1}\mathbf{A}\mathbf{C}_k \\
&= \mathbf{C}_k - \mathbf{A}^{\mathrm{T}}(\mathbf{A}\mathbf{A}^{\mathrm{T}})^{-1}\cdot 0 \\
&= \mathbf{C}_k
\end{aligned}
$$

Thus, we instead write $\mathbf{W}_1^{(k)} = \mathbf{A}^{\mathrm{T}}\mathbf{P}_k + \mathbf{C}$ where $\mathbf{C}$ is constant and only $\mathbf{P}_k$ is being iterated upon.

A second observation is that if all limits are assumed to exist and $\mathbf{A}\mathbf{W}_1^{(\infty)}\mathbf{W}_2^{(\infty)}\ldots\mathbf{W}_h^{(\infty)}\mathbf{x}_\infty = \mathbf{b}$, then $\mathbf{A}^{\mathrm{T}}\mathbf{P}_\infty\mathbf{W}_2^{(\infty)}\ldots\mathbf{W}_h^{(\infty)}\mathbf{x}_\infty = \boldsymbol{\theta}^\star$. An easy way to see this is that

$$
\begin{aligned}
\mathbf{A}\mathbf{W}_1^{(\infty)}\mathbf{W}_2^{(\infty)}\ldots\mathbf{W}_h^{(\infty)}\mathbf{x}_\infty &= \mathbf{A}\mathbf{A}^{\mathrm{T}}\mathbf{P}_\infty\mathbf{W}_2^{(\infty)}\ldots\mathbf{W}_h^{(\infty)}\mathbf{x}_\infty + 0 \\
&= \mathbf{b}
\end{aligned}
$$

so $\mathbf{A}^{\mathrm{T}}\mathbf{P}_\infty\mathbf{W}_2^{(\infty)}\ldots\mathbf{W}_h^{(\infty)}\mathbf{x}_\infty$ is a solution and it is trivially in $\mathbf{range}\left(\mathbf{A}^{\mathrm{T}}\right)$ so it is equal to $\boldsymbol{\theta}^\star$ by Lemma 1.

Now observe that,

$$
\begin{aligned}
\|\mathbf{W}_1^{(\infty)}\mathbf{W}_2^{(\infty)}\ldots\mathbf{W}_h^{(\infty)}\mathbf{x}_\infty - \boldsymbol{\theta}^\star\|_2 &= \|\mathbf{A}^{\mathrm{T}}\mathbf{P}_\infty\mathbf{W}_2^{(\infty)}\ldots\mathbf{W}_h^{(\infty)}\mathbf{x}_\infty + \mathbf{C}\mathbf{W}_2^{(\infty)}\ldots\mathbf{W}_h^{(\infty)}\mathbf{x}_\infty - \boldsymbol{\theta}^\star\|_2 \\
&= \|\mathbf{C}\mathbf{W}_2^{(\infty)}\ldots\mathbf{W}_h^{(\infty)}\mathbf{x}_\infty\|_2 \\
&\leq \|\mathbf{W}_2^{(\infty)}\|\ldots\|\mathbf{W}_h^{(\infty)}\|\cdot\|\mathbf{x}_\infty\|_2\cdot\|\mathbf{C}\|
\end{aligned}
$$

We again see the importance of initialization on the constant $\|\mathbf{C}\|$. Can depth fix this constant however? The inequality suggests that if $h$ is large and the weight norms are smaller than 1 at convergence, then this fixes large $\|\mathbf{C}\|$. Conversely, if the norms are greater than 1, the bound explodes and a small perturbation during initialization can result in radically different solutions. We tested this empirically on randomly generated problems to see if depth helps. We created a linear neural network of varying depth, with $\mathbf{W}_1 = \mathbf{A}^{\mathrm{T}}\mathbf{P}_0 + \mathbf{C}$ where $\|\mathbf{C}\| = 1$ and tested whether depth helps or harms the distance to $\boldsymbol{\theta}^\star$. The other initial weights were all $\mathbf{I}_d$ except $\mathbf{x}_0 = \mathtt{random}(\mathbb{S}^{d-1})$.

Quite surprisingly, we see that the product $\|\mathbf{W}_2\|\ldots\|\mathbf{W}_h\|\|\mathbf{x}\|_2$ increases as the depth increases, but the distance to $\boldsymbol{\theta}^\star$ could decrease nonetheless. It could increase, decrease, or be non-monotonic (see Figure 4.4). In every experiment, the norm product always increased with depth. In the vast majority of experiments, the distance to $\boldsymbol{\theta}^\star$ increased monotonically with depth, signaling that depth causes the error to explode and does not help with generalization.

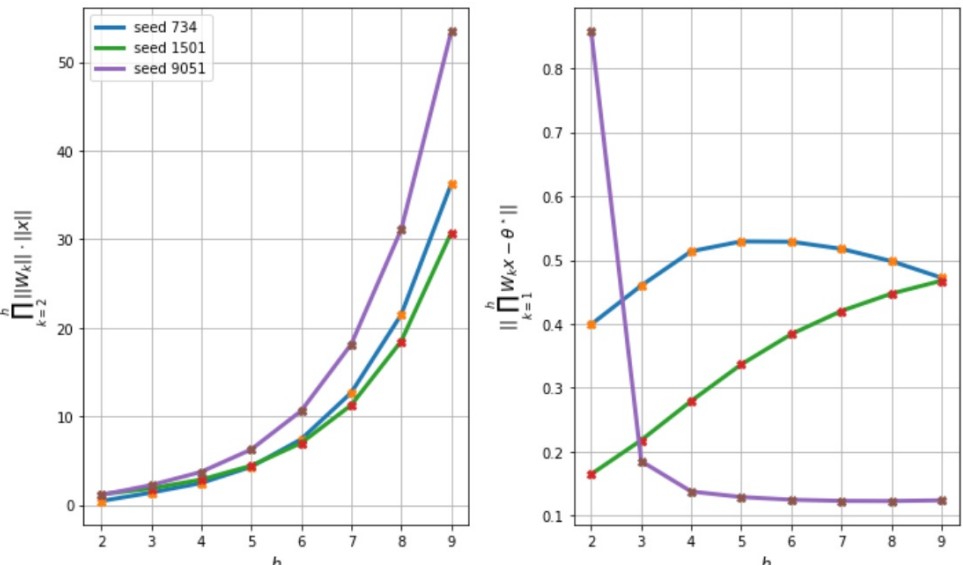

Figure 4.4: Norm product and distance to $\boldsymbol{\theta}^\star$ at the end of training, with varying depths and seeds, exhibiting different properties regarding distance to $\boldsymbol{\theta}^\star$

# 5 Riemannian Linear Neural Networks

In this section, we consider a deep linear model $\mathbf{Ay} = \mathbf{b}$ where $\mathbf{y}$ is parameterized as $\mathbf{y} := \mathbf{W}_1 \mathbf{W}_2 \ldots \mathbf{W}_h \mathbf{x}$ where

$$\mathbf{W}_1, \mathbf{W}_2, \ldots, \mathbf{W}_h \in \text{Stiefel}(d, d) := \{\mathbf{W} \in \mathbb{R}^{d \times d} : \mathbf{W}\mathbf{W}^{\mathrm{T}} = \mathbf{I}_d\}$$

and $\mathbf{x}$ remains unconstrained. The motivation for this model is clear from the previous section. The inequality in Section 4.2 tempts us to enforce that $\|\mathbf{W}_k\| = 1$ and then $\|\mathbf{W}_1 \mathbf{W}_2 \ldots \mathbf{W}_h \mathbf{x} - \boldsymbol{\theta}^\star\|_2 \leq \|\mathbf{x}\|_2 \cdot \|\mathbf{C}\|$, which if $\|\mathbf{x}\|_2$ is not large, hopefully fixes the damage by a poor initialization, or at the very least does not harm it like deep linear networks might. This model makes it so that adding more layers does not increase the upper bound on the error, which can often happen in regular deep linear networks, as shown in the figures in Section 4.2, where in every model adding layers increased the product of norms (an upper bound). However, we shall see that while the product of hidden weight norms is constant, depth in a Riemannian model can have both a positive and negative effect, and results are inconclusive.

## 5.1 Brief Informal Background on Riemannian Optimization

This explanation, while simplistic and informal, is meant to convey the essential notion rather than to provide a detailed and formal account of Riemannian optimization. Additional, formalized and detailed information is provided by Absil et al. (2008); Boumal (2022).

Suppose we wish to find a vector $\mathbf{x} \in \mathbb{S}^{d-1}$ that minimizes the function $\|\mathbf{Az} - \mathbf{b}\|_2^2$ where $\|\mathbf{z}\|_2 = 1$, like we would encounter in Lagrange Multipliers for instance. Neural networks (whether linear or not) do not allow us to specify which domain we want our weights to be in. It does not allow us to constrain them. But in real-world applications, we often want to constrain the parameters. For instance, we might have a problem where we are looking for the correct orientation of an object in space, thus our search domain is only rotation matrices, which is not a linear space, but it is a smooth manifold that is locally linearizable at every point.

Back to our problem of minimizing $f(z) = \|\mathbf{Az} - \mathbf{b}\|_2^2$ over the unit sphere. The unit sphere is not a linear space, so we cannot define an inner product on it, and as such there is no notion of gradient. However, it is locally linearizable at every point. We can find the tangent space $T_\mathbf{z}\mathbb{S}^{d-1}$ at every point $\mathbf{z}$, choose an inner product for it (there are many choices; conceptually, this is not far from preconditioning); an obvious choice

is the standard inner product inherited from the Euclidean space $\mathbb{R}^d$. This tangent space is now a linear space endowed with an inner product, so we can now have a clear notion about the gradients in it.

The gradient of $f(\mathbf{z})$ will not, in general, be in $T_{\mathbf{z}}\mathbb{S}^{d-1}$, so we will define the Riemannian gradient as the vector $\mathrm{rgrad}f(\mathbf{z})$ which is the unique vector in $T_{\mathbf{z}}\mathbb{S}^{d-1}$ such that $\langle \mathrm{rgrad}f(\mathbf{z}), \mathbf{v} \rangle = \mathbf{D}_f(\mathbf{x})\mathbf{v}$ for all $\mathbf{v}$ in $T_{\mathbf{z}}\mathbb{S}^{d-1}$, where $\mathbf{D}_f(\mathbf{x})\mathbf{v} := \lim_{\delta \to 0} \frac{f(\mathbf{x}+\delta\mathbf{v})-f(\mathbf{x})}{\delta}$. As a consequence of this definition, we can easily calculate it with $\mathrm{rgrad}f(\mathbf{z}) = \mathrm{Proj}_{\mathbf{z}}(\nabla f(\mathbf{z}))$ where $\mathrm{Proj}_{\mathbf{z}}$ is the orthogonal projection operator from $\mathbb{R}^d$ to the tangent space $T_{\mathbf{z}}\mathbb{S}^{d-1}$.

We now have $\mathbf{z} - \mathrm{rgrad}f(\mathbf{z})$ be in $T_{\mathbf{z}}\mathbb{S}^{d-1}$, but it is not on $\mathbb{S}^{d-1}$. What we need is a mapping from the tanget space onto the manifold. Such a mapping is called a retraction, and for this case an example is the normalizing function. Now we can define a Riemannian version of gradient descent: move in the direction opposite the Riemannian gradient and retract back to the manifold. This procedure allows us to optimize functions over smooth non-linear manifolds, and not all $\mathbb{R}^d$. This is also a form of regularization, as we can choose "simple" manifolds and, we hope, get "simple" solutions.

This procedure for optimizing over the manifold $\mathbb{S}^{d-1}$ can be extended to any manifold we wish. All we need is the tangent space at every point on the manifold, an inner product on that tangent space, the orthogonal projection operator onto that tangent space, and a retraction. In Section 5.2 we consider Riemannian optimization where our target manifold is the product of Stiefel manifolds (orthogonal matrices).

## 5.2 The Role of Initialization in Riemannian Linear Neural Networks

In this section we consider the problem of solving $\mathbf{A}\mathbf{W}\mathbf{x} = \mathbf{b}$ where $\mathbf{W}$ is either on the Stiefel manifold, or overparametrized as a product of such matrices, and the effects of initialization on this problem. We begin with a definition. The Frobenius distance of an orthogonal $d \times d$ matrix $\mathbf{W}$ from the range of a $d \times n$ full-rank matrix $\mathbf{M}$ is $d_{\mathbf{range(M)}}(\mathbf{W}) := \|\mathbf{M}\mathbf{M}^+\mathbf{W} - \mathbf{W}\|_F$. This definition is sensible because, indeed, $\mathbf{M}^+\mathbf{W}$ minimizes $\|\mathbf{M}\mathbf{X} - \mathbf{W}\|_F$ from the properties of Moore-Penrose pseudoinverse. This definition motivates the following theorem. This theorem is not specifically related to our use cases and models, but we use it to show that we cannot initialize like in the previous sections, which is a key difference to the previous models.

**Theorem 15.** *Let $\mathbf{W} \in \mathbb{R}^{d \times d}$ be an orthogonal matrix and $\mathbf{M} \in \mathbb{R}^{d \times n}$ be of full rank. Then $d_{\mathbf{range(M)}}(\mathbf{W}) = \sqrt{d-n}$.*

*Proof.* The closest matrix to $\mathbf{W}$ in $\mathbf{range}(\mathbf{M})$ is

$$\begin{aligned} \mathbf{Z} &= \mathbf{M}\mathbf{M}^+\mathbf{W} \\ &= \mathbf{M}(\mathbf{M}^\mathrm{T}\mathbf{M})^{-1}\mathbf{M}^\mathrm{T}\mathbf{W}. \end{aligned}$$

All we need to do is calculate the distance between $\mathbf{W}$ and $\mathbf{Z}$.

$$\|\mathbf{Z} - \mathbf{W}\|_F^2 = \|\mathbf{M}(\mathbf{M}^\mathrm{T}\mathbf{M})^{-1}\mathbf{M}^\mathrm{T}\mathbf{W} - \mathbf{W}\|_F^2 = \|(\mathbf{M}(\mathbf{M}^\mathrm{T}\mathbf{M})^{-1}\mathbf{M}^\mathrm{T} - \mathbf{I}_d)\mathbf{W}\|_F^2 = \|\mathbf{M}(\mathbf{M}^\mathrm{T}\mathbf{M})^{-1}\mathbf{M}^\mathrm{T} - \mathbf{I}_d\|_F^2$$

since $\mathbf{W}$ is orthogonal.

Now

$$\begin{aligned} \|\mathbf{M}(\mathbf{M}^\mathrm{T}\mathbf{M})^{-1}\mathbf{M}^\mathrm{T} - \mathbf{I}_d\|_F^2 &= \mathrm{trace}[(\mathbf{M}(\mathbf{M}^\mathrm{T}\mathbf{M})^{-1}\mathbf{M}^\mathrm{T} - \mathbf{I}_d)^\mathrm{T}(\mathbf{M}(\mathbf{M}^\mathrm{T}\mathbf{M})^{-1}\mathbf{M}^\mathrm{T} - \mathbf{I}_d)] \\ &= \mathrm{trace}[\mathbf{M}(\mathbf{M}^\mathrm{T}\mathbf{M})^{-1}\mathbf{M}^\mathrm{T}\mathbf{M}(\mathbf{M}^\mathrm{T}\mathbf{M})^{-1}\mathbf{M}^\mathrm{T} - 2\mathbf{M}(\mathbf{M}^\mathrm{T}\mathbf{M})^{-1}\mathbf{M}^\mathrm{T} + \mathbf{I}_d] \\ &= \mathrm{trace}[\mathbf{M}(\mathbf{M}^\mathrm{T}\mathbf{M})^{-1}\mathbf{M}^\mathrm{T} - 2\mathbf{M}(\mathbf{M}^\mathrm{T}\mathbf{M})^{-1}\mathbf{M}^\mathrm{T}] + d \\ &= d - \mathrm{trace}[\mathbf{M}(\mathbf{M}^\mathrm{T}\mathbf{M})^{-1}\mathbf{M}^\mathrm{T}] = d - \mathrm{trace}[\mathbf{M}^\mathrm{T}\mathbf{M}(\mathbf{M}^\mathrm{T}\mathbf{M})^{-1}] = d - \mathrm{trace}[\mathbf{I}_n] \\ &= d - n \end{aligned}$$

$\square$

A consequence of the previous theorem is that it is impossible for us to have $\mathbf{W}_1^{(k)} \in \mathbf{range}\left(\mathbf{A}^\mathrm{T}\right)$ in the orthogonal network case. The theorem shows that if $\mathbf{W}_1^{(k)}$ is orthogonal and $\mathbf{W}_1^{(k)} = \mathbf{A}^{\mathrm{T}^+}\mathbf{P}_k + \mathbf{C}_k$, then we

always have $\|\mathbf{C}_k\|_F = \sqrt{d - n}$, Hence $\mathbf{C}_k \neq 0$ for all $k$. While we have not shown that $\mathbf{C}_k$ is conserved like in Section 4.2 (in fact, it is not conserved), but this proves that $\|\mathbf{C}_k\| = \sqrt{d - n}$ is conserved across iterations. It also shows that we cannot ever have $\mathbf{W}_1^{(k)} \in \mathbf{range}\left(\mathbf{A}^{\mathrm{T}}\right)$. Thus, we do not necessarily find the minimum norm solution!

$\mathbf{W}_1^{(k)}$ may never be in $\mathbf{range}\left(\mathbf{A}^{\mathrm{T}}\right)$ but we could still have $\mathbf{W}_1^{(\infty)}\mathbf{W}_2^{(\infty)} \dots \mathbf{W}_h^{(\infty)}\mathbf{x}_\infty \in \mathbf{range}\left(\mathbf{A}^{\mathrm{T}}\right)$. We wanted to check whether this happens empirically in orthogonal linear networks, so we tested several problems with pymanopt (Townsend et al., 2016), with the default QR retraction and no line search to keep things as simple as possible (although this did not seem to have an effect regardless).

Figures 5.1, 5.2 are two examples of such experiments, where we solved the same problem (outlined below) using deep orthogonal linear networks with two different initializations, and Figure 5.3 was entirely a different problem. The goal of these experiments was to check whether depth helps us or not in orthogonal linear networks and whether we converge to the minimum norm solution, perhaps regardless of initialization. We clarify that in Figures 5.1, 5.2 both experiments solved the same problem

$$\mathbf{A} = \begin{pmatrix} 5 & -3 & 1 \\ 3 & 1 & -1 \end{pmatrix} \quad \mathbf{b} = \begin{pmatrix} 6 \\ 4 \end{pmatrix}.$$

In all experiments the hidden weights were optimized on the $d \times d$ Stiefel manifold while the outmost layer was unconstrained. The only difference between Figures 5.1, 5.2 was the seed that governed the initialization.

We immediately see that the paths may diverge with depth. This disagrees with (Ablin, 2020), which states that deep orthogonal networks are shallow and that depth has no effect. Ablin (2020)'s result holds only in the matrix factorization case (that is, trying to decompose a given (square) matrix as a product of orthogonal matrices, and to do so, they initialize the weights, all square matrices, to be orthogonal and strictly optimize on Stiefel manifolds). This is unlike our setting, which is not matrix factorization, but rather regression, where the outermost layer is a vector optimized on $\mathbb{R}^d$ and is unconstrained, only the inner layers have the orthogonality constraint. These are two separate problems which are related in the sense that they are both linear models where the weights are simply multiplied together but are in essence distinct in dimension of the objective and the constraints on the parameters. Hence, the trajectories and biases are different as well, as empirically shown in this work.

We also see in Figure 5.2 that even though we had a random initialization, as we have no choice on that with orthogonal networks because of Theorem 15, we still converged to $\boldsymbol{\theta}^\star$. This is very interesting. In a regular deep linear network with random initialization, the odds of converging to $\boldsymbol{\theta}^\star$ are very low, we have never encountered that happening randomly, but for orthogonal networks it happens quite frequently. This is mysterious and we haven't managed yet to find a convincing argument as to why this is the case.

We also see that depth can have both a positive and a negative effect, as seen in Figure 5.1 showing that depth brings us closer to $\boldsymbol{\theta}^\star$ while in Figure 5.3 depths displace us further from $\boldsymbol{\theta}^\star$

The experiments disprove the idea that the distance to $\boldsymbol{\theta}^\star$ is related to the norms of the individual weights, as indicated in Section 4.2. The inequality is very lenient, and depth does not fix a bad initialization, not even in the orthogonal case, as illustrated in Figure 5.3, where depth even makes us farther away from $\boldsymbol{\theta}^\star$.

To further assess the behavior of Riemannian networks optimized on the product of Stiefel manifolds, we have conducted 10000 trials with random initializations on the above linear system of equations, with the goal of exploring whether statistically depth helps or harms the distance to the minimum norm solution. Convergence to $\boldsymbol{\theta}^\star$ is not guaranteed, and while it is interesting to explore when it convergence to $\boldsymbol{\theta}^\star$ happens, it is also useful to ask whether depth helps when it does not happen? Figures 5.4 and 5.5 aim to answer this question.

In Figure 5.4 we draw the histogram of the distance from $\boldsymbol{\theta}^\star$ for depths $h = 1, 3, 6$ and see that as depth increases, the histograms become more centered to the left (smaller error) and also more tightly clustered (smaller variance). This indicates that while all options are possible, statistically when solving the above problem, depth helps us. In Figure 5.5 we plot the 25th, 50th and 75th percentile distances for each respective

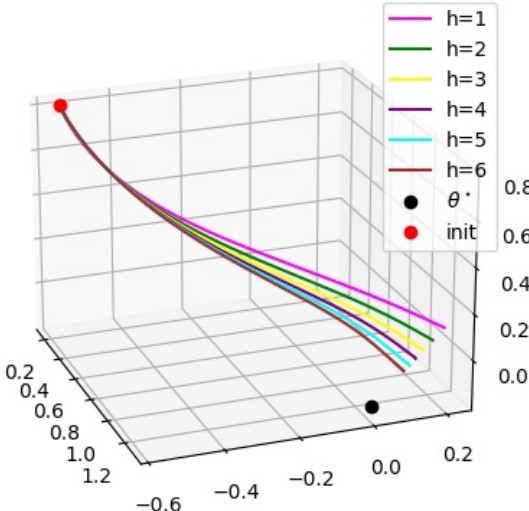

Figure 5.1: Deep orthogonal linear network, seed = 5. Solving $\begin{pmatrix} 5 & -3 & 1 \\ 3 & 1 & -1 \end{pmatrix} \mathbf{x} = \begin{pmatrix} 6 \\ 4 \end{pmatrix}$

$h$, and the shaded area represents one variance. We observe that percentile distances decay and the shaded areas become thinner as $h$ increases, indicating that statistically, depth is beneficial.

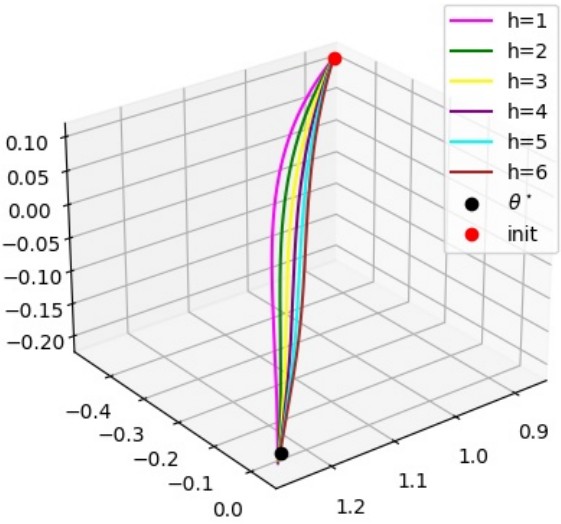

Figure 5.2: Deep orthogonal linear network, seed $= 351$. Solving $\begin{pmatrix} 5 & -3 & 1 \\ 3 & 1 & -1 \end{pmatrix} \mathbf{x} = \begin{pmatrix} 6 \\ 4 \end{pmatrix}$

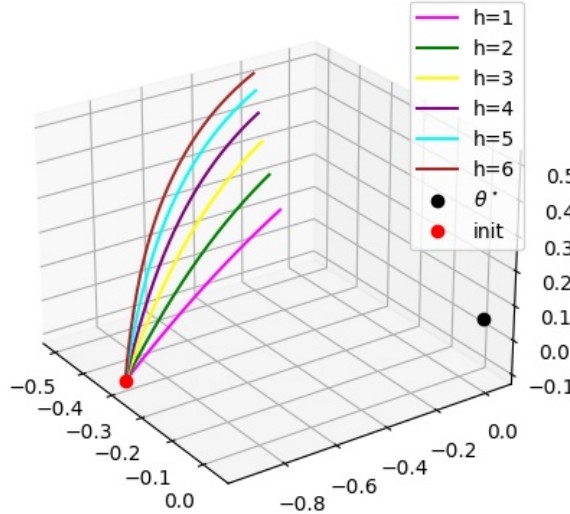

Figure 5.3: Deep orthogonal linear network, seed $= 12$. Solving a randomly generated problem.

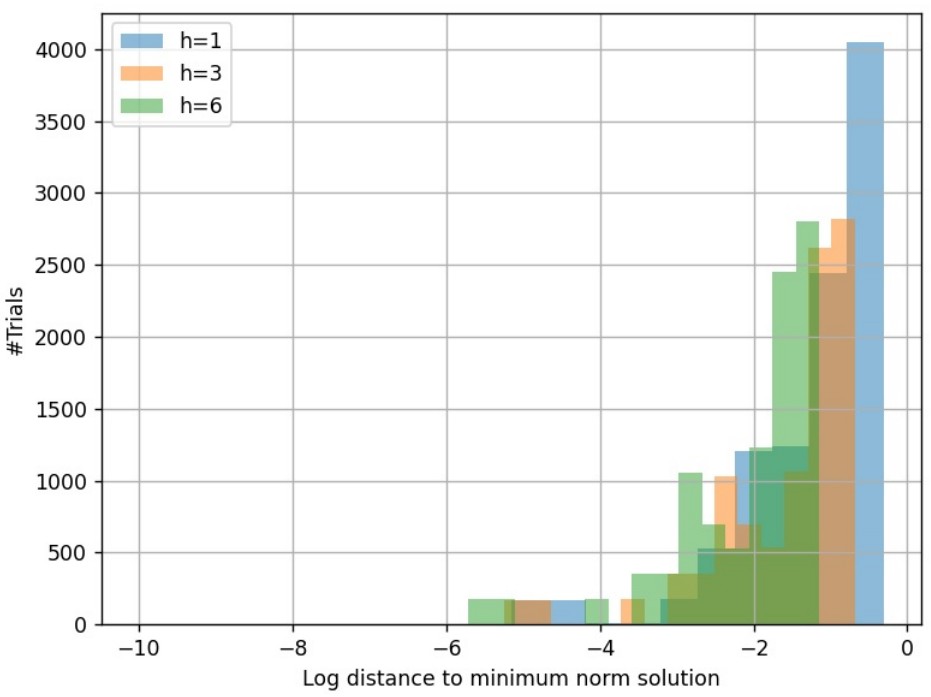

Figure 5.4: Histogram depicting amount of experiments vs distance to $\boldsymbol{\theta}^\star$ observed for different $h$ values in Riemannian setting

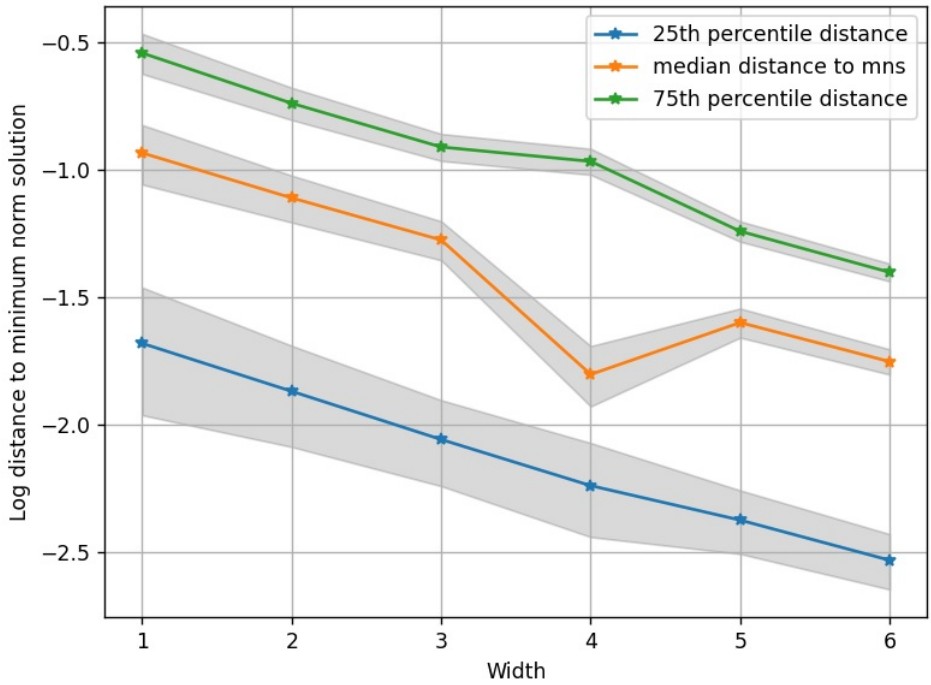

Figure 5.5: Different percentile log distance to $\boldsymbol{\theta}^\star$ for different $h$ values, shaded area is possible values seen in experiments.

## 6  Conclusions

We hope that this work clearly illustrates the pivotal role of initialization in deep learning. For linear networks, when we can control the initialization freely, we have clear advantages of choosing where to converge (Theorem 4, Corollary 7, Corollary 14), and we can collapse the problem from a high-dimensional problem to an equivalent problem with low dimensions (Algorithms 3 and 4). We can ensure convergence to an optimal solution (since we converge to a solution rather than a saddle point) and give a very rough error bound if we can not initialize exactly where we wish (Section 4.2). We saw that where we cannot control the initialization (Section 5), the best we can do is hope to converge to a good solution, and depth often will not fix bad initializations.

The implicit bias determined by initialization is a key question to solve in deep neural networks, and in our work, we attempted to convey the importance of this seemingly innocent part of any parametric method, but there is more work to be done. The new algorithms we propose (Algorithms 3 and 4) need to be looked at further and given bounds on rate of convergence, and any other advantages these methods may have that we hope will come to light. Specifically, we believe that there may be advantages to data-based initializations and possibly other initializations apart from ours that take advantage of the data given to reach a desired bias, but that is something that needs to be carefully and thoroughly researched further, as the industry standard currently is simple random initialization that does not depend on the data. It is also very tempting to show under which circumstances an orthogonal linear network will converge to the minimum-norm solution, as we saw that it happens quite frequently, which is very surprising. A natural next step will be to try generalize our work to the nonlinear case and prove a criterion that will assure convergence to the least norm solution (or a low norm solution) in ordinary deep networks. The issue of extending our work to linear networks of depth greater than $h = 2$ is another matter that requires resolution - a general method for collapsing deep linear networks, or proof that such a method does not exist when $h > 2$. Finally, we hope to find an explanation and perhaps a fix for the zigzag phenomenon that we see in Figure 4.1 that would make the new algorithms even better, and test that solution on non-linear networks, which is the main motivation since it is unlikely the new algorithms will be better than modern methods for linear regression like Krylov subspace solutions.

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
