# OpenReview forum: "On the Role of Initialization on the Implicit Bias in Deep Learning"
_TMLR — Rejected by TMLR_

### Review · Reviewer_eaQ3 · 2023-08-15

**Summary Of Contributions:**

This work analyzes the role of weight initialization in deep linear networks. Several theoretical results are proved, algorithms proposed, and experiments conducted.

The works begins with an observation about solving underdetermined linear systems with gradient descent: choice of initialization determines to which solution the iterates converge. For example, if the initial iterate is in the rowspace of A, then the solution converges to the minimum norm solution.

Building on this observation, the authors proceed to analyze several cases including (1) one hidden layer linear networks (2) two hidden layer linear networks and (3) Riemannian linear networks. Their theoretical results concern notions like (i) if iterates begin in the rowspace of A, then they remain so and (ii) when it is possible to "collapse" the problem into a lower dimensions.

**Audience:**

Yes

**Broader Impact Concerns:**

No concerns about broader impact or ethical implications.


**Claims And Evidence:**

No

**Requested Changes:**

**critical**

The use of "deep learning" in the title is too broad. I suggest changing it to "deep linear networks" to reflect what this paper is about.

The authors should dedicate more space to describing the results *in words* and how they relate to the overall argument. Specifically they should describe each lemma, corollary and theorem prior to stating it, and how it connects to the main argument and/or following results.

For the experiment in section 5.2, more random trials are necessary to assess the behavior of the problem. How about plotting the distributions of distances. one for each $h$, from $\theta^*$ for many random initializations (e.g. O(1000)).  Also only one linear system $A,b$ is analyzed, where $A$ is size 2x3. Analyzing a greater collection of linear systems would strength the results. Could the results you observe be due to numerical error?

Why can Lemma 1 be applied in Corollary 7? Lemma 1 was proven in the context of linear regression.

Need more discussion of why section 5.2 doesn't directly contradict Ablin 2020.

Authors should explain notions before using them. For instance, on page 9, the authors use the term "bi-optimality" before defining or describing it.

**strengthen**

It would be helpful if the authors explicitly prove that linear networks are non-convex for $h > 1$. I know this claim is echoed in a number of other works in the literature, however it would increase clarity to include an explicit proof, e.g. in the appendix.

Figures 4.8, 4.9, and 4.10 could be combined into one plot to reduce space.

Figures 4.3, 4.4, 4.5, 4.6, 4.7 could be combined into a grid of plots to reduce space and facilitate comparison.

Typos
* page 5: please change "thesis" to "paper/article/work".
* page 14: "psuedocode"

Unclear:
* page 14: not clear why the line "conjecture that bi-optimal solution can be useful in meta-learning" is included in this paper.

Page 27: "In the vast majority of experiments the distance to θ ⋆ increased monotonically with depth, signaling that depth
makes the error explodes and does not help with generalization, contrary to some recent works that suggest otherwise."

^ The authors need to cite what works they are referring to. Also explain how exactly their results contradict others in the literature.


**Strengths And Weaknesses:**

**Strengths**

Analysis of initialization for deep linear networks is a reasonable problem. A careful and clear analysis of this would be of interest to the community.

**Weaknesses**

This submission is long, technical and not very clear. I did not find any technical errors, but neither did I reach great clarity on the results. Therefore overall I am not fully confident that all claims are supported with clear evidence, which is a key evaluation criteria of this journal.

Despite lots of technical details, not enough detail is given to the interpretation of the results, which makes the argument difficult to follow.

Section 5.1 on Riemannian linear networks is very unclear. The authors already write that the "results are inconclusive".  Furthermore, comparing results from two different seeds is not sufficient to conclude that "The experiments disprove the idea that the distance to θ⋆ is related to the norms of the individual weights".

The plots in this work are not good.

---

> ### Author Response · Authors · 2023-12-01
> **Responses**
>
> **Comment:** The use of "deep learning" in the title is too broad. I suggest changing it to "deep linear networks" to reflect what this paper is about.
>
> **Answer:** Done.
>
> **Comment:** The authors should dedicate more space to describing the results in words and how they relate to the overall argument. Specifically they should describe each lemma, corollary and theorem prior to stating it, and how it connects to the main argument and/or following results.
>
> **Answer:** Done.
>
> **Comment:** For the experiment in section 5.2, more random trials are necessary to assess the behavior of the problem. How about plotting the distributions of distances. One for each $h$, from $\theta^\star$ for many random initializations (e.g. O(1000)).
>
> **Answer:** Thank you for the idea! We have done as you suggested and reflected the result in the work in the final two paragraphs of the Riemannian section, and Figures 5.4 and 5.5.
>
> *Comment:** Also only one linear system $A$,$b$ is analyzed, where $A$ is size $2x3$. Analyzing a greater collection of linear systems would strength the results.
>
> **Answer:** The experiments on $2x3$ matrices were meant for illustration purposes, and the size was select so that we can effectively visualize the results. We do not believe that experiments with additional matrices will help convey the points in this section.
>
> **Comment:** Could the results you observe be due to numerical error?
>
> **Answer:** We do not think this is the case. The distances are too large for that.
>
> **Comment:** Why can Lemma 1 be applied to Corollary 7? Lemma 1 was proven in the context of linear regression.
>
> **Answer:** Lemma 1 is a general lemma on underdetermined linear systems, and is applicable regardless of the optimization method. We added a paragraph before Lemma 1 that clarifies this.
>
> **Comment:** Need more discussion of why section 5.2 does not directly contradict Ablin 2020.
>
> **Answer:** We immediately see that the paths may diverge with depth. This disagrees with \citep{ablin2020deep}, which states that deep orthogonal networks are shallow and that depth has no effect. \citet{ablin2020deep}\'s result holds only in the matrix factorization case (i.e., attempting to decompose a given (square) matrix as a product of orthogonal matrices, and to do so, they initialize the weights, all square matrices, to be orthogonal and strictly optimize on Stiefel manifolds). This is unlike our setting, which is not matrix factorization, but rather regression where the outermost layer is a vector optimized on $\mathbb{R}^d$ and is unconstrained, only the inner layers have the orthogonality constraint. These are two separate problems which are related in the sense that they are both linear models where the weights are simply multiplied together but are in essence distinct in dimension of the objective and the constraints on the parameters. hence the trajectories and biases are different as well, as empirically shown in this work.
>
> We added this discussion to the article (see Section 5.2).
>
> **Comment:** Authors should explain notions before using them. For instance, on page 9, the authors use the term "bi-optimality" before defining or describing it.
>
> **Answer:** We added a definition of bi-optimality before its first use. Are there additional terms that the reviewer believes are undefined?
>
> **Comment:** It would be helpful if the authors explicitly prove that linear networks are non-convex for $h > 1$. I know this claim is echoed in a number of other works in the literature, however it would increase clarity to include an explicit proof, e.g. in the appendix.
>
> **Answer:** Non-convexity for $h>1$ is almost trivial, One can easily see that for $h>1$ any solution for which $(\forall i: W_i = 0)$ is a saddle point. Since there is a saddle point, the problem cannot be convex. We added a sentence that says this in the introduction, right after we first mention non-convexity of $h>1$ and the trivial saddle points.
>
> **Comment:** Figures 4.8, 4.9, and 4.10 could be combined into one plot to reduce space.
>
> **Answer:** Done.
>
> **Comment:** Figures 4.3, 4.4, 4.5, 4.6, 4.7 could be combined into a grid of plots to reduce space and facilitate comparison.
>
> **Answer:** Done
>
> **Comment:** Typos
>
> **Answer:** All typos were fixed. Thank you for spotting them.
>
> **Comment:** Unclear: page 14: not clear why the line "conjecture that bi-optimal solution can be useful in meta-learning" is included in this paper.
>
> **Answer:** We removed that line.
>
> **Comment:** Page 27: "In the vast majority of experiments the distance to theta star increased monotonically with depth, signaling that depth makes the error explodes and does not help with generalization, contrary to some recent works that suggest otherwise."  The authors need to cite what works they are referring to. Also explain how exactly their results contradict others in the literature.
>
> *Answer:** We removed the part starting with "contrary".

---

### Review · Reviewer_T5w8 · 2023-08-17

**Summary Of Contributions:**

The paper studies the optimization of deep linear neural networks. Particularly, the paper studies an underdetermined system and shows that different initializations lead to solutions with different qualities. The paper scrutinizes the optimization process and proposes algorithms that optimize variables with reduced dimensionalities. Empirically, the paper shows some interesting examples to demonstrate the importance of initial solutions for underdetermined linear neural network settings.

**Audience:**

Yes

**Claims And Evidence:**

Yes

**Requested Changes:**

1. The connections to neural tangent kernel and infinite width networks are mentioned. However, I think there require more discussions. The general ideas are contradictory from a certain perspective: this work suggests that initialization points are critical while the previous work often suggests that when the networks are wide enough, the initializations do not matter.

2. This may also relate to the lottery ticket hypothesis and more discussions on this seems necessary.

3. The paper also does not relate to popular weight initialization methods widely used in practice, and why/why not those methods should work well under the paper's settings.


**Strengths And Weaknesses:**

Strengths:
The paper takes a novel and interesting perspective to study the optimization of deep linear neural networks, and it may inspire future research in more general settings.


Weaknesses:
1. The findings are limited to deep linear neural networks. There are no theoretical or empirical implications on how such results can generalize to practical deep networks.

2. The proposed algorithms exhibit zigzag behaviors during optimization, and it is not understood very clearly why this happens and how to avoid that.

3. Even under the deep linear network setting, it is not clear how we utilize the observations. I.e., if we want to find weights with certain properties other than the minimal normed solution, how do we incorporate such properties as objectives for the optimization? It seems that after all, we still retreat back to regularizations.

---

> ### Author Response · Authors · 2023-12-01
> **Responses**
>
> **Comment:** Weaknesses pointed out by the reviewer ((1), (2), and (3)).
>
> **Answers:** We agree with the reviewer's assessment that it is unclear how the results translate to concrete implications for practical deep neural networks, and that the algorithms are mostly theoretical as they exhibit peculiar behaviors.
>
> Our work is mostly theoretical, exploring the role of initialization in deep learning. Although it is intended to provide insight into deep learning, it is not meant yet to guide concrete modifications for practical deep learning. However, also at the request of Reviewer EJET, we added a paragraph at the end of the introduction that discusses possible research ideas for improved learning suggested by our work.
>
> **Comment:** The connections to neural tangent kernel and infinite width networks are mentioned. However, I think there require more discussions. The general ideas are contradictory from a certain perspective: this work suggests that initialization points are critical while the previous work often suggests that when the networks are wide enough, the initializations do not matter.
>
> **Answer:**This is a good point.  However, we believe that the NTK literature's result should be interpreted slightly differently than stated by the reviewer. While previous works on NTKs show that if the overparameterization (width of the network in this sense) is aggressive enough, and the weights have been initialized from a rotation-invariant distribution such as a standard normal distribution, then the law of large numbers assures us that the model is guaranteed to operate in this lazy regime and behave as a linear model, solving an underdetermined system of equations, and has all the disadvantages of that model (like bad local minima, large norm solutions). Hence, while initialization does not matter for operating in this regime, it matters significantly from a generalization point of view when training the model. This is in line with our work, which emphasizes the importance of clever initializations. Furthermore, previous work on NTK also show that one can choose the train an NTK by solving the linear system indirectly by kernel regression, which is a convex problem with a square matrix and a single solution, and there the initialization truly does not matter. Solving the kernel regression problem yields the minimum norm solution and is equivalent to solving the linear system with a row-space initialization. However, this does not exclude the possibility of another algorithm that will benefit from careful initialization.
>
> We added this discussion to Section 3 of the revised manuscript.
>
> **Comment:**
> This may also relate to the lottery ticket hypothesis and more discussions on this seems necessary.
>
> **Answers:**
> Indeed our results have the same "spirit" as the lottery ticket hypothesis (LTH) results, but there are two key differences. The crux of the LTH is that at initialization, a randomly initialized neural network contains a subnetwork that, if trained in isolation, will reach the same or similar accuracy to the complete network after training, and propose an iterative method for finding this subnetwork. The similarity is clear in that we reduce the cost of per iteration of tranining/testing, but the differences are that we propose an a-priori method for reducing the number of parameters and faster iterations, rather than an a posteriori iterative one. Furthermore, our collapsed model is not composed of subweights of the original model, but rather constraining the weights to be of a particular low rank form and optimizing the respective vectors that make up this decomposition.
>
> We added this discussion to the article near the end of Section 3.

---

> ### Author Response · Authors · 2023-12-01
> **More Responses**
>
> **Comment:**
> The paper also does not relate to popular weight initialization methods widely used in practice, and why/why not those methods should work well under the paper's settings.
>
> **Answers:**
> Unfortunately, there is little relation, as our initializations, while random, are are supported on a set of zero measure. In contrast, most popular methods today sample scalar entries individually, and the support has a positive measure (if not the entire space). Two prominent examples of industry standard initialization are Xavier and He Initializations.
>
> In a Xavier Initialization we generate all entries from a uniform distribution on $-\frac{1}{\sqrt{s}}$ and $\frac{1}{\sqrt{s}}$, where $s$ is the number of neurons in the previous layer. The goal of this initialization, which is widely used for the activation functions $\frac{1}{1+e^{-x}}$ and $\tanh(x)$, is to have constant variance across all layers. This prevents the gradients from vanishing or exploding.
> He Initialization was invented to solve the problem that Xavier does not work well when the activation function is ReLU. When performing He initialization, we generate numbers from a normal distribution with mean $0$ and variance $\frac{2}{s}$.
>
>  Both of these initializations, and indeed most initialization techniques today, sample entries individually, and so they miss the big picture of possible dependency on the data given and how to use it. They are designed with optimization in mind, rather than generalization, and are very different from the methods we propose. Initializing with these methods will almost surely not yield $\theta^\star$ and will not take advantage of the collapsing property we have outlined. It is possible, however, that these initialization schemes avoid possible exploding/vanishing gradient phenomena better than our proposed methods.
>
> We have added the above discussion to the paper, at the end of Section 4.1

---

### Review · Reviewer_EJET · 2023-09-04

**Summary Of Contributions:**

The authors show initialization plays a key role in determining the solution deep linear networks converge to. With smart initialization, deep linear networks can be biased to converge to the minimum norm solution. The authors demonstrate the role of initialization in three settings:

--For ordinary linear regression (no hidden layers), initializing the weights to 0 ensures convergence to the minimum norm solution

--For a single hidden layer, initializing W0 in the range of AT and x0 nonzero biases the solution to be "bi-optimal" - i.e. W and x are each optimal given the other (W is the minimum Frobenius norm, x is the minimum 2-norm vector respectively)

--For two hidden layers, initializing W1 and W2 using a specifically structured initialization given in Lemma 9 and Theorem 12 again biases the solution to be "bi-optimal."

**Audience:**

Yes

**Broader Impact Concerns:**

No obvious concerns about ethical implications.

**Claims And Evidence:**

Yes

**Requested Changes:**

If the authors provide more clarity on the specific insights described at the theory level, this could lead to downstream at the level of practice. Not necessarily at the level of how to make use of the conclusions about initialization, but something that gives less theory-oriented readers an idea of what questions this creates for current best practices in models.

Correct me if I'm wrong, but this work still supports the general guidelines of making random initialization as small as possible while still remaining trainable. Is there something the authors expect to see based on their theoretical analysis of initialization that would change what the community is already doing?

**Strengths And Weaknesses:**

Note: As a reviewer who does not specialize in theory, I do not feel qualified to comment on particular weaknesses of this work. Therefore, I will focus on the strengths:

--Illustrates through theoretical analysis and empirical examples how initialization plays a crucial role in determining the solutions learned by deep linear networks. Proper initialization can bias the model towards better solutions like the minimum norm solution.

--Shows how a single hidden layer linear network can be collapsed to optimize only d+n parameters rather than d^2+d original parameters with proper initialization.

Questions:

Does this work imply that there are regularization benefits to data-dependent initialization? Since some of the theorems involve reaching the optimal solution based on W being initialized in range(A), does it make sense to perform data-based initialization rather than random initialization to create an implicit bias towards the minimum norm solution?

The authors mention that collapsing the single hidden layer network to optimize a smaller set of parameters can lead to faster optimization in their specific case of an underdetermined system. Is this an empirical claim shown by Figure 4.1, or does theory justify this claim as well?

---

> ### Author Response · Authors · 2023-12-01
> **Responses**
>
> **Comment:** Does this work imply that there are regularization benefits to data-dependent initialization? Since some of the theorems involve reaching the optimal solution based on $W$ being initialized in $range(A)$, does it make sense to perform data-based initialization rather than random initialization to create an implicit bias towards the minimum norm solution?
>
> **Answer:** This is an interesting proposition. We believe that there can be likely benefits for data-dependent initialization, though the topic needs to be researched thoroughly to determine whether this is the case. We now mention this as possible future research in the conclusions section of the revised paper.
>
> **Comment:** The authors mention that collapsing the single hidden layer network to optimize a smaller set of parameters can lead to faster optimization in their specific case of an underdetermined system. Is this an empirical claim shown in Figure 4.1, or does theory justify this claim as well?
>
> **Answer:** We can definitely say (theory-wise) that our algorithms have smaller per-iteration costs. Indeed, the original single hidden layer model before collapsing, the update step for $W_k$ requires subtracting two $d\times d$ matrices. This operation alone is $O(d^2)$, while every step in the collapsed algorithm is $\max(n, T_{A})$, so this claim is also supported by theory, especially for sparse data matrices. However, this does not necessarily mean that optimization is faster because convergence might be slower. Empricially, we do not observe this, but we do not have supporting theory as well. We made sure that the text only claims a faster per-iteration cost.
>
> **Comment:** If the authors provide more clarity on the specific insights described at the theory level, this could lead to downstream at the level of practice. Not necessarily at the level of how to make use of the conclusions about initialization, but something that gives less theory-oriented readers an idea of what questions this creates for current best practices in models.
>
> **Answer:** Our work is predominantly theoretical. Nevertheless, we added a paragraph at the end of the introduction stating possible avenues in which it can benefit practitioners (although more research is needed to actually realize these potential benefits).
>
> **Comment:** Correct me if I'm wrong, but this work still supports the general guidelines of making random initialization as small as possible while still remaining trainable. Is there something the authors expect to see based on their theoretical analysis of initialization that would change what the community is already doing?
>
> **Answer:** As mentioned in the previous response, this work is predominantly theoretical. It does not offer concrete changes to what the community is already doing, although it suggests possible avenues for improved learning, as discussed in the additional paragraph in the Introduction that we mention in the response to the previous comment.

---

### Author Response · Authors · 2023-09-23
**Extension asked**

On September 5th I sent the action editor the following mail:

"Dear Mr. Mishra,

I hope this email finds you well. I am writing to request an extension for the current step of the review process for submission 1273.

I fully understand the importance of meeting deadlines, but unfortunately I am going on a family trip in two days, and I won't return until after the deadline has already passed. I'm afraid I won't be able to dedicate the proper effort that this important process deserves under these time constraints. We are committed to delivering a high quality paper, and we would not want to rush this step.

Since it is holiday season where me and the other author are from, it is likely that he will have limited availability as well. Thus, I am asking you whether it would be possible to get an extension of 2 more weeks to converse with the reviewers and correct any issues.

I understand that extensions are not always possible, and I deeply appreciate your understanding.

Thank you very much, I am awaiting your response."

He approved the extension for 2 weeks and asked me to post my request here for bookkeeping purposes. Thank you.

---

### Decision · Action_Editor_Eeoy · 2023-12-04

**Recommendation:** Reject

**Comment:**

As I mentioned earlier, there are major concerns about the writing in the paper. It looks more like a seminar or study report without a coherent story for the readers. For example, a reader does not know what the paper is trying to convince. Does regularization play a role? Is it something that is not known? The paper does show some nice toy examples on how to make things work but I do not understand what the concrete contributions (at least not super clear from a reader point of view) are. There are just too many sections/subsections and without a clarity on what to expect when.

I would encourage the authors to consider a major revision by improving the readability of the paper.

**Audience:**

Yes, it would be interesting to researchers to get some new insights.

**Claims And Evidence:**

The paper studies the role of regularization in deep linear neural networks. It nicely goes through the developments and shows some interesting insights. As the paper is positioned more of a study, the experiments and the related developments are okay but not sufficient. However, I have major concerns on the writing of the paper (one of the reviewers already pointed it out and I agree with them).

**Resubmission Of Major Revision:**

The authors may consider submitting a major revision at a later time.